# Precise Information Control in Long-Form Text Generation

Jacqueline He[ω]    Howard Yen[π]*    Margaret Li[ω]*    Shuyue Stella Li[ω]    Zhiyuan Zeng[ω]
Weijia Shi[ω]    Yulia Tsvetkov[ω]    Danqi Chen[π]    Pang Wei Koh[ω,α]    Luke Zettlemoyer[ω]

[ω]Paul G. Allen School of Computing Science & Engineering, University of Washington
[π]Princeton Language and Intelligence (PLI), Princeton University    [α]Allen Institute for AI
{jyyh, pangwei, lsz}@cs.washington.edu

## Abstract

A central challenge in language models (LMs) is *faithfulness hallucination*: the generation of information unsubstantiated by input context. To study this problem, we propose **Precise Information Control (PIC)**, a new task formulation that requires models to generate long-form outputs grounded in a provided set of short self-contained statements, without adding any unsupported ones. PIC includes a full setting that tests a model's ability to include exactly all input claims, and a partial setting that requires the model to selectively incorporate only relevant claims. We present PIC-Bench, a benchmark of eight long-form generation tasks (e.g., summarization, biography generation) adapted to the PIC setting, where LMs are supplied with well-formed, verifiable input claims. Our evaluation of a range of open and proprietary LMs on PIC-Bench reveals that, surprisingly, state-of-the-art LMs still hallucinate against user-provided input in over 70% of generations. To alleviate this lack of faithfulness, we introduce a post-training framework that uses a weakly supervised preference data construction method to train an 8B PIC-LM with stronger PIC ability—improving from 69.1% to 91.0% $F_1$ in the full PIC setting. When integrated into end-to-end factual generation pipelines, PIC-LM improves exact match recall by 17.1% on ambiguous QA with retrieval, and factual precision by 30.5% on a birthplace fact-checking task, underscoring the potential of precisely grounded generation.

## 1  Introduction

Exact control over what content language models (LMs) should produce is an open challenge in long-form generation; LMs may miss key details, or generate superfluous or false claims. This is true not only for unconstrained tasks that depend on parametric knowledge, but even when the prompt states exactly what information the model ought to include. Omissions, distortions, and fabrications collectively constitute a particular breakdown of information control, more generally referred to as hallucinations [43, 41, 42].

In language modeling parlance, *factuality hallucinations* are generations that cannot be verified by external real-world knowledge, while *faithfulness hallucinations* contradict supplied input context knowledge [75, 82, 41]. Faithfulness hallucination is problematic in instances where the LM must reliably transform or synthesize source text, such as in scientific literature synthesis [5], abstractive summarization [75], or arbitrary style transfer [97]. The elimination thereof is just as critical in fast-moving or high-stakes real-world applications in which updated, reliable in-context evidence must override the model's stale parametric memory [114]. Examples include gathering the latest scientific consensus on COVID-19, or generating customized medical guidance based on a patient's

---

*Equal contribution.

39th Conference on Neural Information Processing Systems (NeurIPS 2025).

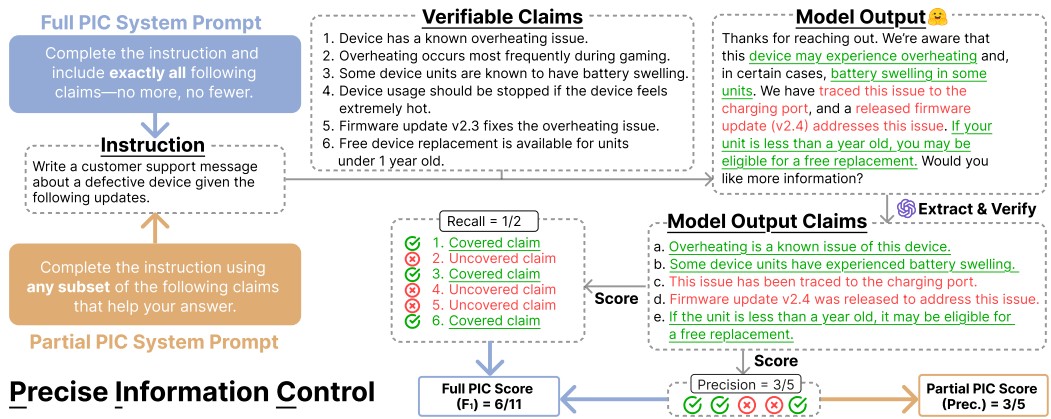

Figure 1: Precise Information Control (PIC) evaluates faithfulness hallucination at the level of verifiable claims. It has two settings: full PIC requires including all input claims and no others, while partial PIC requires a subset of input claims (exact system prompts in Table 10).

approved treatment list. Furthermore, even efforts to reduce *factual* hallucination may require faithful grounding on retrieved or filtered context [7], making intrinsic control a critical building block for trustworthy generation.

To this end, we propose **P**recise **I**nformation **C**ontrol (PIC), a task formulation for assessing faithfulness hallucination in long-form generation. PIC raises a fundamental question: *can LMs produce long-form responses that are strictly grounded on a set of explicit statements, without introducing any unsupported claims?*

Instead of forming binary judgments of faithfulness hallucination [78, 42], we frame this problem at the level of *verifiable claims*—short, standalone information units that can be unambiguously validated [105]. To support more flexible, user-defined control, we consider two task settings specified in natural language. *Full PIC* reflects scenarios that require complete information coverage (e.g., text rewriting), where the LM should incorporate all given input claims (evaluated with $F_1$). Meanwhile, *partial PIC* applies when the context includes irrelevant or extraneous claims, requiring selective grounding (e.g., summarization). Therefore, in this case, the LM ought to choose only a relevant subset (evaluated with precision) for its response. As an example, Fig. 1 illustrates for a customer-support message about a defective device, following updated guidelines. Full PIC requires the LM to integrate all six claims into its answer, while under partial PIC, the LM may choose any helpful subset. Both PIC modes require precise control: every output claim must be corroborated by input claims.

To evaluate PIC across long-form generation, we introduce PIC-Bench, a 1.2K-example benchmark covering six full and two partial PIC task settings. In PIC-Bench evaluation, the verification search space is restricted to input context. Therefore, unlike with factual hallucination [10, 77, 29, 47, 113, 117], the complete elimination of faithfulness hallucination ought to be tractable and well-defined. Nevertheless, we find that PIC is far from solved on existing state-of-the-art LMs. In particular, perfect PIC (i.e., zero faithfulness hallucination) remains out of reach. The proportion of outputs with perfect PIC does not exceed 30% (measured by $F_1$) for full PIC tasks, and 70% (measured by precision) for partial PIC tasks. This means that even frontier LLMs still produce faithfulness hallucinations in over 70% of full PIC outputs. Further, leading open LMs consistently underperform proprietary LMs by over 8%, indicating headroom for improvement.

We then develop a two-stage post-training framework to improve PIC ability, using a weakly-supervised reward signal to efficiently construct PIC-formatted preference data. Specifically, we tune Llama 3.1 8B Instruct [33] with this framework to produce PIC-LM, a highly controllable model: its generations reliably adhere to input claims, while reducing unsubstantiated statements. Relative to Llama 3.1 8B Instruct on PIC-Bench, PIC-LM raises average full PIC from 69.1% to 91.0% in $F_1$, and average partial PIC from 73.6% to 93.3% in precision, outperforming all open baselines and closing the performance gap to frontier LLMs. Nevertheless, the elimination of faithfulness hallucination in long-form generation remains elusive, underscoring the challenging nature of this problem.

Finally, we demonstrate the practical utility of PIC-LM in downstream applications. While faithfulness and factuality hallucination are often addressed separately in prior work [10, 42], our results suggest

that robust faithful grounding may help reduce factual errors. Embedding our models into generation pipelines, where input is either substantiated with context or iteratively refined, yields more factually accurate and reliable final outputs. In a retrieval-augmented generation setting [7] where claims come from retrieved documents, PIC-LM improves standard exact match from 52.5% to 61.5% on ASQA, an ambiguous long-form QA task. In a chain-of-verification pipeline [20] where claims are automatically generated and verified, PIC-LM raises factual accuracy on a birthplace factoid task (from 65.9% to 86.0% in factual precision) and QAMParI [2] (from 13.5% to 22.6% in $F_1@5$).

## 2 Task Formulation

### 2.1 Problem Definition

**Verifiable claims.** Following [70, 105], we consider PIC at the level of verifiable claims: short statements that describe a standalone event or state, and are evaluated with respect to some knowledge source (in our case, input claims). Decomposing outputs into individual claims is a common long-form evaluation technique [77, 72, 105, 120]. Claims can be represented at various granularities, for example, at the sentence level [45, 72] or atomic level (i.e., imparting the most minimal unit of information) [77, 120]. Sentence-level claims may mix supported and unsupported information [77], while strictly atomic claims often lack the context needed for unambiguous verification. Further, not every claim is inherently provable, e.g., metaphors, analogies, or other complex linguistic constructs.

Verifiable claims strike a balance between sentence and atomic granularity; they are concise enough to isolate specific assertions, but with sufficient context for unambiguous verification (see Fig. 1 for examples). We obtain verifiable claims from long-form outputs using an automatic, LLM-based claim extractor from Song et al. [105].

**PIC settings.** A PIC task input entails an instruction $\mathcal{I}$ and a set of verifiable claims $C = c_1, \ldots, c_n$. Given $(\mathcal{I}, C)$, an LM $\theta$ generates a long-form response $\theta(\mathcal{I}, C)$, from which a set of verifiable claims $C' = c'_1, \ldots, c'_m$ can be extracted. For comprehensiveness, we define two modes based on user intent, each with distinct constraints.

In the **full PIC** setting, the LM must incorporate every given claim into its response, neither adding nor omitting any information. Specifically, each claim should appear at least once in the generation, and no statements may appear in the response without an input claim to support it. Full PIC task examples may include reliable style transfer or open generation (e.g., "Write a short biography on Elena Ferrante", given a highly curated set of claims).

In the **partial PIC** setting, the LM may incorporate only a relevant subset of the context claims, and introducing new claims remains disallowed. For tasks such as summarization or QA with retrieval, not all input claims should be reproduced—only the most salient information is desirable (e.g., "Summarize this given passage on economic policies in Seattle, Washington", or "Write a short biography on Elena Ferrante", given a set of input claims retrieved from Wikipedia entries). However, the generated output should remain strictly grounded in provided context. Therefore, the partial setting requires that $C' \subseteq C$ (no additions are allowed, but omissions are permissible).

To meaningfully assess a PIC task, we require that the input claim set $C$ be well-formed, satisfying: (1) non-emptiness: $C \neq \emptyset$, (2) no duplicates: $\forall c_i, c_j \in C, i \neq j \Rightarrow c_i \neq c_j$, (3) no contradictions: $\nexists c_i, c_j \in C$ such that $c_i$ contradicts $c_j$, and (4) task relevance: $C$ should contain enough well-formed claims to adequately fulfill the instruction $\mathcal{I}$, though not necessarily exhaustive.

The last criterion (task relevance) ensures that the task remains simple and tractable for the LM by avoiding adversarial claims that may obscure the intended instruction. In practice, our evaluation of PIC tasks does not directly account for instruction-following ability. The orthogonal relationship between instructions and claims can be separately evaluated through a dedicated instruction-following metric (which we explore in §5.2) that requires the LM to simultaneously satisfy the provided claims and adhere to requirements defined by its PIC setting.

### 2.2 Task Construction

We can recast any long-form generation task from a standard instruction-tuning dataset $\mathcal{D}_{IT}$ into our PIC format. Let $\mathcal{D}_{IT}$ consist of samples $(\mathcal{I}, p, y)$, where $\mathcal{I}$ is the task instruction, $p$ is optional

context (i.e., a source document for a summarization task), and $y$ is a long-form gold response.[2] For full PIC, we assume these tasks do not contain $p$, so we extract $C$ from $y$. In partial PIC, we extract $C$ directly from $p$. In either case, we end up with $(\mathcal{I}, C)$, where $C$ is well-formed.

## 2.3 Task Evaluation

We automatically evaluate the PIC ability of generated responses by first decomposing output into verifiable claims using a claim extractor [105], and then employing an LLM-based claim verifier $\text{support}(\cdot, \cdot)$ [105] that checks for semantic equivalence. Both claim extraction and verification exhibit strong agreement with human judgments (Appendix B.4). Formally, for any candidate claim $c$ and a claim set $S$, let

$$\text{support}(c, S) = \begin{cases} \text{True}, & \text{if } c \text{ is semantically equivalent to some } s \in S, \\ \text{False}, & \text{otherwise.} \end{cases}$$

In other words, it follows that

$$\text{Precision}(C', C) \;=\; \frac{\left| \{ x \in C' \mid \text{support}(x, C) \} \right|}{|C'|}, \text{Recall}(C', C) \;=\; \frac{\left| \{ x \in C \mid \text{support}(x, C') \} \right|}{|C|}.$$

Intuitively, the precision is the proportion of claims in the response that are supported by the context claims, while the recall is the proportion of context claims that are supported in the response. Given both, we can also calculate the $F_1$.

Recall that the full setting requires that $C' = C$ (no omissions or additions). Hence, we evaluate with $F_1$, which penalizes omissions (recall). The partial setting requires that $C' \subseteq C$ (no additions are allowed, but omissions are allowed), so we only consider precision. For both settings, we also report the proportion of samples with zero faithful hallucination—denoted as the proportions of perfect $F_1$ and prefect precision, respectively.[3]

# 3 PIC-Bench

## 3.1 Evaluation Tasks

PIC is applicable to any long-form generation task. We design PIC-Bench, a benchmark of six full and two partial PIC tasks.[4] We choose eight long-form generation datasets from prior work [81, 77, 32, 105, 42] and re-frame their evaluation to our problem formulation. Note that PIC-Bench assesses only how well the model's responses adhere to PIC constraints when completing a task, instead of downstream task performance (which can be assessed orthogonally). Table 1 shows PIC-Bench task information (additional details in Appendix B.1).

**Full PIC tasks.** For the full PIC setting, we choose tasks where comprehensive coverage of the input verifiable claims is integral to a high-quality response. ENTITYBIOS$_{\text{PIC}}$ [77] is a biography generation task with a high number of claims. PopBios [32] is a short biography-generation dataset for popular entities, from which we derive two tasks: PopBios-P$_{\text{PIC}}$ and PopBios-CF$_{\text{PIC}}$. PopBios-P$_{\text{PIC}}$ asks the model to generate a biography of a well-known entity in line with the LM's parametric knowledge, while PopBios-CF$_{\text{PIC}}$ is a counterfactual variant in which the parametric entity is systematically replaced with another well-known entity. AskHistorians$_{\text{PIC}}$ and ELI5$_{\text{PIC}}$ [105] are Reddit-sourced long-form QA datasets on historical topics or simplified explanations to complex topics, respectively. ExpertQA$_{\text{PIC}}$ [72] is a long-form QA task of expert-curated, domain-specific questions. In full PIC, all claims are extracted from the long-form gold response $y$.

**Partial PIC tasks.** For the partial PIC setting, we identify tasks where selective grounding is appropriate—that is, where only a relevant subset of context claims should appear in the output.

---

[2]We assume that $\mathcal{D}_{IT}$ follows the same format as the Alpaca [108] dataset, in which not every sample contains context $p$, so the model input is either $(x, p)$ if provided, or just $x$.

[3]PIC focuses on faithfulness rather than fluency or completeness: every verifiable output claim must be anchored to the provided context.

[4]Some tasks (e.g., biography generation) can be either full or partial PIC, depending on user intent and claim coverage. For simplicity, each PIC-Bench task is assigned to exactly one PIC setting.

Table 1: PIC-Bench dataset statistics and examples. For the partial PIC setting, we omit the full context $p$ for brevity. $N$ is the number of task samples, and $\bar{C}$ is the average number of input claims.

| Task Name | PIC Type | $N$ | $\bar{C}$ | Example Instruction $\mathcal{I}$ |
|---|---|---|---|---|
| ENTITYBIOS_PIC | Full | 183 | 50.5 | Generate a factual biography about Suthida. |
| POPBIOS-P_PIC | Full | 111 | 20.1 | Give me a biography on Erwin Schrödinger, the scientist who discovered Quantum Mechanics, Schrödinger's Cat Thought Experiment. |
| POPBIOS-CF_PIC | Full | 111 | 20.1 | Give me a biography on Oscar Wilde, the scientist who discovered Quantum Mechanics, Schrödinger's Cat Thought Experiment. |
| ELI5_PIC | Full | 146 | 12.5 | Answer the following question(s): why it's common to have 87-octane gasoline in the US but it's almost always 95-octane in Europe? |
| ASKHISTORIANS_PIC | Full | 158 | 19.2 | In the original Star Wars: A New Hope, Obi-Wan Kenobi instructs R2-D2 to connect to the Imperial network to gain access to the whole system. Did the concept of an interconnected vast computer network exist in 1977? |
| EXPERTQA_PIC | Full | 152 | 13.5 | Answer the question(s): What's the difference between modern and contemporary architecture? |
| FACTS_PIC | Partial | 150 | 63.5 | Explain the benefits of using mobile technology to improve healthcare management in both hi-income and low-income countries. {Context $p$} |
| XSUM_PIC | Partial | 200 | 30.6 | Summarize the following text in around 20-25 words. {Context $p$} |

FACTS_PIC [42] is a factual grounding task, in which LMs must base their answers on long-context documents. XSUM_PIC [81] is an abstractive summarization task of BBC news articles. XSUM originally required one-sentence summarization, but we extend it to a longer-form setting by imposing a higher minimum word count. In partial PIC, claims are sourced from the supplied context $p$.

## 3.2 Baselines

We benchmark a range of state-of-the-art, instruction-tuned LMs. Among open-weight LMs, we consider standard instruction-tuned LMs along the 8B and 70B parameter ranges: Llama 3.1 8B Instruct and 3.3 70B Instruct [33], Tulu 3 8B and 70B [52], Ministral 8B Instruct [80], and Hermes 3 8B and 70B [111]. We also consider 32B open-weight reasoning models: Qwen 3 32B [109], QwQ 32B [110], and R1-Qwen-32B (distilled from DeepSeek-R1) [18]. These models are capable of step-by-step reasoning based on thinking control tokens; we report results using the thinking mode, which consistently yields better performance (see Appendix B.6 for non-thinking mode results). Finally, we consider frontier LLMs as approximate skylines: GPT-4o [86] and Claude-3.5 Sonnet [4].

# 4 PIC-LM: A Post-Training Approach

We propose PIC-LM, an 8B instruction-following language model with high PIC ability. PIC-LM follows a conventional, two-stage post-training process: supervised fine-tuning (SFT) followed by length-normalized DPO [91, 52]. Key to our approach is a novel weakly supervised preference data construction method that optimizes for PIC, while preserving instruction-following capabilities.

## 4.1 Supervised Fine-Tuning (SFT)

To provide the LM with a strong initialization for PIC, we conduct SFT on a diverse mix of high-quality, long-form instruction-tuning data. This mix includes two multi-task datasets–No Robots [93] and FLAN [118]–as well as a CNN news summarization [101], biography generation [77], and long-form QA tasks [42]. For PIC-Bench evaluation, the biography and QA datasets are in-distribution (ID), whereas the other datasets are out-of-domain (OOD), allowing us to assess performance in both settings. Following the procedure in §2, each sample is converted to the PIC format, yielding the triple $(\mathcal{I}, C, y)$. We then filter the data sample pool by only retaining samples with high PIC scores. We conduct SFT from Llama 3.1 8B Instruct [33], producing $\theta_{\text{SFT}}$.

## 4.2 Preference Tuning

**Preference data construction.** Recall that in our SFT dataset, each sample consists of an instruction $\mathcal{I}$, a set of original context claims $C_{\text{orig}}$, and an original gold response $y_{\text{orig}}$. We construct a *perturbed* variant of each $C_{\text{orig}}$ by randomly dropping a subset of the claims, resulting in a reduced context $C_{\text{perturb}} \subset C_{\text{orig}}$. $\mathcal{I}, C_{\text{perturb}}$ are then passed to $\theta_{\text{SFT}}$, which already has learned PIC ability, to obtain the response $y_{\text{perturb}}$. Thus, for each sample, we have $(\mathcal{I}, C_{\text{orig}}, C_{\text{perturb}}, y_{\text{orig}}, y_{\text{perturb}})$.

Our preference dataset should consist of tuples $(\mathcal{I}, C, y^+, y^-)$, in which $\mathcal{I}$ is the instruction, $C$ is the appropriate context (chosen to match the preferred response), and $y^+$ and $y^-$ denote the chosen and rejected responses, respectively. Each sample leads to two valid constructions of $(\mathcal{I}, C, y^+, y^-)$:

$$(\mathcal{I}, C, y^+, y^-) = \begin{cases} (\mathcal{I}, C_{\text{orig}}, y_{\text{orig}}, y_{\text{perturb}}), \\ (\mathcal{I}, C_{\text{perturb}}, y_{\text{perturb}}, y_{\text{orig}}). \end{cases}$$

**Preference data sampling strategy.** One naive approach to choosing $(\mathcal{I}, C, y^+, y^-)$ is to sample each construction with equal probability, which suffices if the goal is only to optimize for PIC ability. Since $y_{\text{perturb}}$ is often shorter than $y_{\text{orig}}$, random sampling may also mitigate length bias [88, 92]. However, this strategy does not account for instruction-following ability: if $C_{\text{perturb}}$ drops too many essential claims, $y_{\text{perturb}}$ may fail to adequately answer $\mathcal{I}$, degrading instruction alignment. Conversely, not all dropped claims are equally important—$y_{\text{perturb}}$ may better satisfy $\mathcal{I}$ by being more concise, focused, or stylistically aligned. This motivates a strategy that adaptively selects between $y_{\text{orig}}$ and $y_{\text{perturb}}$ based on overall instruction compatibility.

Thus, taking inspiration from prior reward strategies [14], we propose a weakly supervised data sampling strategy to construct $(\mathcal{I}, C, y^+, y^-)$ in a way that balances PIC and instruction-following skills. Using a competent instruction-following reference model $\theta$, we first define a *normalized log-probability drop* score as follows:

$$\text{Normalized Log-Probability Drop} = \sigma \left( \frac{\log p_\theta(y_{\text{orig}}^{(L)} \mid \mathcal{I})}{L} - \frac{\log p_\theta(y_{\text{perturb}}^{(L)} \mid \mathcal{I})}{L} \right),$$

which is the normalized difference in per-token log probability of generating the original response versus the perturbed response, computed over the last $L$ tokens of each sequence (wherein $L$ is fixed, e.g., 20 tokens) for efficient computation and to alleviate length bias. We normalize the difference into the interval $(0, 1)$ via the logistic function ($\sigma(z) = \frac{e^z}{1+e^z}$).

For each data sample, if the normalized probability drop score exceeds some hyperparameter $\tau \in [0, 1]$, then we select $(\mathcal{I}, C, y^+, y^-) = (\mathcal{I}, C_{\text{orig}}, y_{\text{orig}}, y_{\text{perturb}})$, and $(\mathcal{I}, C_{\text{perturb}}, y_{\text{perturb}}, y_{\text{orig}})$ otherwise. We assume that the per-token log probability is a reasonable proxy signal for instruction-following ability, meaning that the probability of $\theta$ generating a given response diminishes if the response cannot sufficiently fulfill the instruction without necessary context. A larger probability drop indicates that the perturbed response is significantly worse than the original response. Empirically, we select different $\tau$ values for full and partial settings, and observe that this yields better instruction-following performance than random sampling.[5]

**Training.** We train $\theta_{\text{SFT}}$ on our constructed preference data using length-normalized direct preference optimization, which is simply DPO with log-probabilities normalized for length [91, 52, 76]. In Appendix C.4, we present ablations that (i) skip the SFT step, i.e., initialize directly from Llama 3.1 8B Instruct, and (ii) experiment with alternative loss functions.

## 5 Main Results

### 5.1 PIC-Bench Results

Table 2 and Table 3 show PIC-Bench results in full and partial PIC settings, respectively, and Appendix B.2.4 shows corresponding 95% bootstrapped confidence intervals. We highlight a few salient trends.

**The complete elimination of faithful hallucination is challenging, even under ideal conditions.** By design, PIC-Bench tasks are an ideal instantiation of PIC with a straightforward goal: to construct responses that ground on only *well-formed* verifiable claims, without introducing additional information. Despite this intentional lack of complexity, even frontier LLMs fail to achieve PIC scores close to 100%. The average proportion of perfect PIC scores is even lower, not exceeding 30% in the full and 70% in the partial setting.

---

[5]We ablate several values of $\tau$ and show high LLM-as-a-judge correlation with GPT-4.1 in Appendix C.4.

Table 2: PIC-Bench results (full setting). Metrics include $F_1$, the proportion of perfect $F_1$ (Perf.), reported per-task and as an average. Best values in **bold**. All reasoning LMs (denoted with a †) are evaluated with thinking mode enabled. For all metrics, the higher the better.

| | ENTB$_{PIC}$ | | POPB-P$_{PIC}$ | | POPB-CF$_{PIC}$ | | ASKH$_{PIC}$ | | ELI5$_{PIC}$ | | EXPQA$_{PIC}$ | | Avg. | |
|---|---|---|---|---|---|---|---|---|---|---|---|---|---|---|
| | $F_1$ | Perf. | $F_1$ | Perf. | $F_1$ | Perf. | $F_1$ | Perf. | $F_1$ | Perf. | $F_1$ | Perf. | $F_1$ | Perf. |
| **Open-weight LMs (8B)** | | | | | | | | | | | | | | |
| LLAMA 3.1 8B INST. | 74.4 | 0.6 | 83.5 | 2.7 | 23.7 | 0.0 | 81.9 | 5.1 | 77.8 | 8.9 | 73.5 | 5.3 | 69.1 | 3.7 |
| TULU 3 8B | 79.6 | 1.6 | 87.8 | 2.7 | 51.3 | 0.0 | 80.4 | 3.2 | 81.8 | 10.3 | 80.7 | 11.8 | 76.9 | 4.9 |
| MINISTRAL 8B INST. | 78.6 | 0.6 | 89.0 | 9.0 | 63.5 | 0.0 | 76.8 | 5.1 | 78.1 | 11.0 | 77.6 | 15.1 | 77.3 | 6.8 |
| HERMES 3 8B | 74.9 | 0.6 | 84.9 | 6.3 | 44.6 | 0.0 | 72.9 | 2.5 | 75.9 | 4.1 | 74.1 | 7.9 | 71.2 | 3.6 |
| **Open-weight LMs (32B)** | | | | | | | | | | | | | | |
| QWEN 3 32B† | 74.1 | 2.7 | 88.2 | 18.9 | 48.2 | 0.9 | 77.5 | 7.0 | 67.7 | 6.9 | 63.6 | 7.2 | 69.9 | 7.3 |
| QwQ 32B† | 86.7 | 1.6 | 87.6 | 2.7 | 64.1 | 1.8 | 79.6 | 3.8 | 80.0 | 6.9 | 80.9 | 12.5 | 70.3 | 7.6 |
| R1-QWEN-32B† | 82.3 | 1.1 | 87.5 | 9.0 | 58.8 | 0.9 | 81.1 | 4.4 | 82.2 | 10.3 | 80.0 | 13.2 | 78.6 | 6.5 |
| **Open-weight LMs (70B)** | | | | | | | | | | | | | | |
| LLAMA 3.3 70B INST. | 89.1 | 2.2 | 93.4 | 17.1 | 45.5 | 0.0 | 91.2 | 12.7 | 90.3 | 19.9 | 85.6 | 27.0 | 82.5 | 13.1 |
| TULU 3 70B | 84.8 | 7.1 | 89.2 | 23.4 | 52.0 | 0.0 | 83.7 | 15.8 | 82.5 | 19.2 | 73.4 | 17.8 | 77.6 | 13.8 |
| HERMES 3 70B | 81.4 | 1.6 | 89.4 | 12.6 | 56.6 | 0.0 | 81.6 | 5.1 | 79.7 | 9.6 | 79.8 | 14.5 | 78.1 | 7.2 |
| **Proprietary LMs** | | | | | | | | | | | | | | |
| CLAUDE 3.5 | 88.8 | 6.6 | 93.1 | 22.5 | 1.0 | 0.0 | 83.7 | 18.4 | 85.8 | 29.5 | 63.8 | 21.1 | 69.4 | 16.3 |
| GPT-4o | 91.9 | 7.7 | 94.0 | 27.0 | 74.6 | 0.9 | **96.1** | **37.3** | 93.9 | 47.3 | **92.7** | **47.4** | 90.5 | 27.9 |
| ***Ours:* PIC-LM (8B)** — from LLAMA 3.1 8B INST. | | | | | | | | | | | | | | |
| PIC-LM$_{SFT ONLY}$ | 86.7 | 4.9 | 87.2 | 7.2 | 71.7 | 1.8 | 86.6 | 15.2 | 82.1 | 19.2 | 77.3 | 21.7 | 81.9 | 11.7 |
| PIC-LM | **94.1** | **8.2** | **96.0** | **36.0** | **81.0** | **3.6** | 95.1 | 32.3 | **94.1** | **52.7** | 85.6 | 34.9 | **91.0** | **28.0** |

**Proprietary models mostly lead open-weight baselines.** GPT-4o consistently outperforms the open-weight baselines by a meaningful margin on average task performance. Claude 3.5 underperforms against most baselines in the full PIC setting (primarily due to very low $F_1$ on POPBIOS-CF$_{PIC}$), but beats all open-weight models in the partial setting. Within each model family, larger models tend to achieve higher PIC scores. Different open-weight model families exhibit substantially varied performance, indicating that PIC ability isn't solely determined by model size. Despite their larger size, 32B reasoning models achieve PIC scores comparable to 8B non-reasoning models, suggesting that explicit reasoning capabilities alone do not drive substantial gains on PIC tasks.

**LMs struggle in the counterfactual PIC regime.** POPBIOS-CF$_{PIC}$ is the most challenging task, with perfect-$F_1$ scores in the single digits. It serves as a stress test, swapping biographical claims with counterfactual entities that directly conflict with a model's parametric knowledge. This simulates scenarios such as updating outdated facts or enforcing user-specific overrides, where strict instruction-following is desirable even when it contradicts internal priors.

Table 3: PIC-Bench results (partial setting). Metrics include precision (Prec.), the proportion of perfect precision (Perf.), reported per-task and as an average. Best values in **bold**. All reasoning LMs (denoted with a †) are evaluated with thinking mode enabled. For all metrics, the higher the better.

| | FACTS$_{PIC}$ | | XSUM$_{PIC}$ | | Avg | Avg |
|---|---|---|---|---|---|---|
| | Prec. | Perf. | Prec. | Perf. | Prec. | Perf. |
| **Open-weight LMs (8B)** | | | | | | |
| LLAMA 3.1 8B INST. | 70.0 | 19.3 | 77.2 | 54.0 | 73.6 | 36.7 |
| TULU 3 8B | 75.3 | 18.7 | 76.9 | 50.5 | 76.1 | 34.6 |
| MINISTRAL 8B INST. | 75.4 | 30.0 | 82.7 | 63.0 | 79.1 | 46.5 |
| HERMES 3 8B | 74.1 | 27.3 | 81.2 | 42.0 | 77.7 | 34.7 |
| **Open-weight LMs (32B)** | | | | | | |
| QWEN 3 32B† | 64.5 | 17.3 | 78.5 | 46.0 | 71.5 | 31.7 |
| QwQ 32B† | 70.3 | 12.7 | 86.9 | 58.5 | 78.6 | 35.6 |
| R1-QWEN-32B† | 71.1 | 26.7 | 79.6 | 53.5 | 75.3 | 40.1 |
| **Open-weight LMs (70B)** | | | | | | |
| LLAMA 3.3 70B INST. | 79.4 | 32.0 | 69.3 | 67.0 | 74.3 | 49.5 |
| TULU 3 70B | 67.7 | 14.7 | 85.5 | 60.0 | 76.7 | 37.3 |
| HERMES 3 70B | 78.0 | 25.3 | 87.0 | 62.5 | 82.5 | 43.9 |
| **Proprietary LMs** | | | | | | |
| CLAUDE 3.5 | 82.0 | **53.3** | 90.4 | 77.0 | 86.2 | 65.2 |
| GPT-4o | 87.4 | 51.3 | 93.0 | **84.0** | 90.2 | **67.7** |
| ***Ours:* PIC-LM (8B)** — from LLAMA 3.1 8B INST. | | | | | | |
| PIC-LM$_{SFT ONLY}$ | 81.2 | 46.0 | 94.0 | 69.5 | 87.6 | 57.8 |
| PIC-LM | **89.8** | 49.3 | **96.8** | 49.3 | **93.3** | 52.9 |

Among open-weight baselines, Llama 3.1 8B Instruct performs worst with an average $F_1$ of 23.7%, while among closed models, Claude 3.5 achieves only 1%, as it frequently abstains from producing

factually incorrect outputs.[6] Although structurally similar to PopBios-P$_{PIC}$, one of the easiest PIC tasks, the conflicting entities in PopBios-CF create extreme knowledge collisions, consistent with Wu et al. [121]'s finding that LMs fail when input knowledge diverges from internal priors.

**PIC-LM shows substantial improvements.** Our 8B PIC-LM outperforms all open-weight baselines across both evaluation settings. In the full evaluation setting, PIC-LM achieves an average $F_1$ of 91.0% and a perfect-$F_1$ proportion of 28.0%, compared to the strongest open-weight baseline, Llama 3.3 70B Instruct, at 82.5% and 13.1%, respectively. In the partial evaluation setting, PIC-LM reaches an average precision of 93.3% and a perfect-precision proportion of 52.9%, surpassing the best open-weight baseline, Hermes 3 70B, at 82.5% and 43.9%, respectively.

Even the ablation model PIC-LM$_{SFT\ Only}$—which is trained only with SFT on PIC-formatted data and without further preference optimization—substantially outperforms Llama 3.1 8B Instruct. This highlights that simply fine-tuning on PIC-formatted instructions provides a significant boost to PIC performance. Starting from the weakest open-weight baseline, PIC-LM nearly closes the gap to GPT-4o, surpassing it in both average $F_1$ (91.0% vs. 90.5%) and average precision (93.3% vs. 90.2%). For the proportion of *perfect* scores, PIC-LM performs competitively with GPT-4o in the full evaluation setting (28.0% vs. 27.9%) but trails GPT-4o and Claude 3.5 Sonnet in the partial setting (52.9% vs. 67.7% and 65.2%, respectively).

Nevertheless, PIC-LM outperforms all open-weight LMs by a considerable margin across every metric, demonstrating that a simple post-training strategy on an 8B open model can achieve near–state-of-the-art PIC performance without the need for closed-source models or larger parameter budgets.

## 5.2 PIC-LM Maintains Instruction-Following Abilities

An ideal controllable LM ought to generate responses that can answer the instruction in a useful manner, while fully grounding on source context when required [42]. Orthogonal to PIC, we explore how well PIC-LM can follow instructions on each PIC task. Following Asai et al. [5], we employ Prometheus 2 [49], an open-source evaluator LM that scores long-form outputs based on a custom instruction-following rubric from 1 to 5 (defined in Table 27).

A degenerate case of PIC happens when the LM learns to hack the metric by reproducing the input claims $C$ verbatim, leading to an unsatisfactory response with a perfect PIC score. As a sanity check, we introduce ConcatClaims, a lower-bound setting that simply concatenates input claims into a response. Despite perfect PIC, ConcatClaims should perform poorly on Prometheus as claim concatenation ignores task-specific requirements (e.g., word length, style).

Table 4 shows Prometheus results (mean $\pm$ SE) for each PIC task. ConcatClaims scores poorly (averaging 2.01) as it fails to satisfy task instructions. In contrast, Llama 3.1 8B Instruct is significantly better at instruction-following (3.75); as an instruction-tuned model, it competently follows task requirements. Compared to Llama 3.1 8B Instruct, PIC-LM$_{SFT\ Only}$ shows a slight decline across most tasks except for gains on the counterfactual task PopB-CF$_{PIC}$ and XSUM$_{PIC}$, which brings its average performance to 3.87. PIC-LM mostly regains its instruction-following ability except on XSUM$_{PIC}$. We observe qualitatively that it experiences some difficulty in following the strict word constraint. Nevertheless, PIC-LM has the highest Prometheus average at 3.92. Altogether, these results indicate that PIC-LM generally preserves the source model's instruction-following ability.

Table 4: Instruction-following evaluations using Prometheus, reported as the task average with standard error. ConcatClaims is a degenerate lower bound (Low). Best values in **bold**.

| | EntB.$_{PIC}$ | PopB-P$_{PIC}$ | PopB-CF$_{PIC}$ | AskH$_{PIC}$ | ELI5$_{PIC}$ | ExpQA$_{PIC}$ | FACTS$_{PIC}$ | XSUM$_{PIC}$ | Avg. |
|---|---|---|---|---|---|---|---|---|---|
| ConcatClaims (Low) | 2.26 $\pm$ .10 | 2.77 $\pm$ .14 | 2.31 $\pm$ .12 | 2.21 $\pm$ .11 | 1.97 $\pm$ .11 | 1.76 $\pm$ .09 | 1.71 $\pm$ .10 | 1.06 $\pm$ .02 | 2.01 |
| Llama 3.1 8B Inst. | **4.22** $\pm$ .05 | **4.44** $\pm$ .05 | 2.01 $\pm$ .12 | 4.07 $\pm$ .06 | **4.14** $\pm$ .06 | **4.11** $\pm$ .06 | 3.71 $\pm$ .07 | 3.28 $\pm$ .08 | 3.75 |
| *Ours:* **PIC-LM** — from Llama 3.1 8B Inst. | | | | | | | | | |
| PIC-LM$_{SFT\ Only}$ | 3.79 $\pm$ .04 | 4.10 $\pm$ .04 | 3.69 $\pm$ .07 | 3.99 $\pm$ .04 | 4.01 $\pm$ .04 | 3.93 $\pm$ .05 | 3.87 $\pm$ .06 | **3.57** $\pm$ .05 | 3.87 |
| PIC-LM | 3.84 $\pm$ .04 | 4.18 $\pm$ .04 | **3.79** $\pm$ **.07** | **4.21** $\pm$ .04 | 4.09 $\pm$ .04 | 4.03 $\pm$ .04 | **4.01** $\pm$ .05 | 3.23 $\pm$ .06 | **3.92** |

---

[6]Claude's RLHF alignment to "helpful, harmless, and honest" principles likely causes this abstention [3].

# 6 Practical Applications of PIC-LM

One major assumption to PIC-Bench is its reliance on well-formed input claims. Identifying appropriate claims in real-world scenarios, however, poses two challenges: (1) claims may not be well-formed, and (2) users may not know which claims to include in the desired response (e.g., for fact-finding tasks). We bridge the divide between our controlled benchmark and practical application by demonstrating that PIC-LM, when situated in multi-component systems, improves factuality on long-form QA tasks. First, with retrieval-augmented generation (RAG), PIC-LM can identify relevant claims from retrieved context to generate more accurate answers (§6.1). Second, in a self-verification pipeline, swapping in PIC-LM at the final generation step boosts end-task factual accuracy (§6.2).

**Baselines.** First, as a lower-bound baseline, we compare against Llama 3.1 8B Instruct without context (No Context (Low)). We also evaluate zero-shot prompting baselines with Llama 3.1 8B Instruct, Tulu 3 8B, Ministral 8B Instruct, and Hermes 3 8B, and two context-aware baselines that foster greater context reliance. The first is CAD (context-aware decoding) [102], a decoding technique that factors out the LM's prior knowledge to emphasize attention toward input context; we apply it to Llama 3.1 8B Instruct. The second is SelfCite 8B [14], a strong attributed LM that generates responses with context attributions (or citations), and is post-trained using a self-supervised reward that leverages context ablation. Since we do not consider attribution quality in this work, we strip out citation markers before evaluation.

## 6.1 Retrieval-Augmented Generation

In retrieval-augmented generation (RAG), the input query is supplemented with relevant external texts to guide the LM's generation [37, 54, 7, 104]. Recent work [67, 32, 74, 121, 122] indicates that LMs fail to effectively use retrieved context when it conflicts with their internal knowledge, even when the correct answer is in the context. Unlike the strict input conditions imposed by PIC-Bench, retrieved context may contain duplicate, irrelevant, or noisy claims. Can PIC-LM generalize to such instances? We test on the ASQA [106] dataset, which comprises 948 long-form QA pairs. In ASQA, each question is under-specified and requires multiple short-form answers to fully disambiguate (e.g., "Who was the ruler of France in 1830?" is ambiguous and has multiple correct answers).

Following Gao et al. [27], we report exact match recall (EM), taking the top-5 retrieved passages from Wikipedia, decomposed to verifiable claims, as input context. We report a standard setting that involves all samples, and an oracle setting with only samples where the retrieved context has at least one gold answer. Table 5 shows results. Aside from Mistral 8B Inst., all context-based setups outperform the no-context baseline (30.9% EM). PIC-LM achieves a statistically significant boost in both regimes. Under the standard regime, it scores $61.5\% \pm 2.1$ versus $56.2\% \pm 2.1$ for CAD, the strongest baseline.

Table 5: RAG performance on ASQA on standard (948 samples) and oracle (885 samples) regimes, reported as average EM $\pm$ 95% bootstrapped CI. Best values in **bold**.

| Setting | Standard EM | Oracle EM |
|---|---|---|
| No Context (Low) | $30.9 \pm 2.2$ | N/A |
| Llama 3.1 8B Inst. | $52.5 \pm 2.1$ | $56.2 \pm 2.0$ |
| Tulu 3 8B | $42.7 \pm 2.0$ | $45.6 \pm 2.0$ |
| Ministral 8B Inst. | $31.8 \pm 1.7$ | $34.0 \pm 1.8$ |
| Hermes 3 8B | $52.8 \pm 2.0$ | $56.4 \pm 2.0$ |
| CAD (on Llama 3.1 8B Inst.) | $56.2 \pm 2.1$ | $60.1 \pm 2.0$ |
| SelfCite 8B | $52.2 \pm 2.1$ | $52.5 \pm 2.1$ |
| PIC-LM 8B | $\mathbf{61.5} \pm 2.1$ | $\mathbf{65.9} \pm 2.0$ |

## 6.2 Self-Verification Pipeline

PIC-LM remains effective even in settings where relevant external claims are not provided. Drawing inspiration from chain-of-verification [20] and self-consistency sampling [11, 115], we describe how PIC-LM can improve end-task factuality in a pipeline system that auto-generates and verifies claims, without any dependency on knowledge augmentation (e.g., RAG or search tools). This approach leverages the observation that LMs typically perform verification more reliably than generation [24, 61], particularly for factual assessment [35]. By exploiting this asymmetry, we can effectively self-correct problematic behaviors such as factual hallucination [13, 89, 20, 46].

Table 6: Average performance (with 95% bootstrapped CIs) on the Birthplace and QAMParI tasks. The No Context setting serves as a lower bound. Best values in **bold**.

| Setting | Birthplace Factual Prec. | QAMParI Prec. | Rec.@5 | $F_1$@5 |
|---|---|---|---|---|
| No Context (Low) | $19.4_{\pm 2.6}$ | $6.1_{\pm 0.9}$ | $9.9_{\pm 1.4}$ | $6.5_{\pm 0.9}$ |
| Llama 3.1 8B Inst. | $65.9_{\pm 4.5}$ | $11.6_{\pm 1.2}$ | $21.4_{\pm 2.0}$ | $13.5_{\pm 1.3}$ |
| Tulu 3 8B | $80.3_{\pm 4.0}$ | $16.8_{\pm 1.5}$ | $25.5_{\pm 2.2}$ | $18.4_{\pm 1.5}$ |
| Ministral 8B | $83.7_{\pm 4.1}$ | $\mathbf{27.2_{\pm 2.6}}$ | $14.0_{\pm 1.6}$ | $15.3_{\pm 1.5}$ |
| Hermes 3 8B | $53.4_{\pm 4.2}$ | $9.9_{\pm 1.1}$ | $20.3_{\pm 2.1}$ | $12.1_{\pm 1.2}$ |
| CAD (on Llama 3.1 8B Inst.) | $63.6_{\pm 5.4}$ | $15.7_{\pm 1.7}$ | $23.8_{\pm 2.4}$ | $17.5_{\pm 1.8}$ |
| SelfCite 8B | $69.4_{\pm 4.5}$ | $17.7_{\pm 1.5}$ | $23.3_{\pm 2.1}$ | $18.6_{\pm 1.6}$ |
| Llama 3.3 70B Inst. | $84.7_{\pm 4.0}$ | $14.2_{\pm 1.3}$ | $\mathbf{26.8_{\pm 2.2}}$ | $16.7_{\pm 1.4}$ |
| PIC-LM 8B | $\mathbf{86.0_{\pm 4.0}}$ | $25.5_{\pm 2.1}$ | $25.6_{\pm 2.2}$ | $\mathbf{22.6_{\pm 1.9}}$ |

Given an input instruction, the LM: (1) generates a draft response; (2) formulates verification questions based on this draft output; (3) independently fact-checks each question with self-consistency sampling and majority voting to produce a set of unique verified claims; and (4) produces a final response using only the verified claims that conform to the original instruction.

We deploy our pipeline on two downstream factuality tasks. First, following Dhuliawala et al. [20], we evaluate on a birthplace factoid task of the form: "`Name some {occupation}s born in {location}`", which we seed with 252 `{occupation}`, `{location}` pairs. We report factual accuracy using average claim precision, in which the final response is decomposed into a set of verifiable claims, and each claim is fact-checked using Google Search via the Serper API [105, 120].

Second, we evaluate on QAMParI [2], an ODQA task for which answers consists of multiple entities that are spread across different sources (e.g., "Which book had illustrations by Pauline Baynes?"). Following Gao et al. [27], we report the precision, recall@$k$, and $F_1$@$k$ on QAMParI, where $k = 5$ is the maximum number of correct answers used for recall calculation. We use Llama 3.3 70B Instruct as the generator LM for steps 1-3 on both tasks, which we find to be sufficiently large enough for accurate claim self-validation. At step 4, the final generation step where claims are transformed into a long-form response, we can swap in PIC-LM and different baseline approaches.

Table 6 shows chain-of-verification results. On the birthplace verification task, PIC-LM outperforms all 8B baselines, and slightly surpasses the larger Llama 3.3 70B Instruct, demonstrating the efficacy of better PIC. On QAMParI, all context-supplied baselines perform similarly for $F_1$@5, with larger variances for precision (with Hermes 3 8B the lowest at 9.9%, and Ministral 8B the highest at 27.2%) and recall@5 (with Ministral 8B the lowest at 14.0, and Llama 3.3 70B the highest at 26.8%). PIC-LM 8B has the best trade-off between precision and recall@5, with the best $F_1$@5 at 22.6.

## 7 Discussion

Control is the keystone of reliable LM generation; it underpins user trust and system reliability, from everyday information-seeking [27, 64] to high-stakes political, legal, and medical applications [17, 58, 57]. In this work, we formulate faithfulness hallucination as a claim-level control problem through the PIC, which breaks down long-form generation into discrete verifiable claims. Our contribution is three-pronged: we present a simple yet challenging benchmark, an initial training method and models, and motivating use cases.

The PIC paradigm confers two practical benefits. First, PIC ensures controllability by allocating the highest degree of faithfulness toward user-provided context over parametric knowledge. This design allows for the easy uptake of amended or updated information. The second advantage is interpretability: as all generated claims must be grounded in the input space, forming a closed feedback loop, the user can trace any output claim back to its source claim set, making verification precise and tractable. Future efforts may explore how to surface PIC behavior in earlier stages such as continual pre-training, either through the incorporation of learning cues via control sequences [28], the strategic re-ordering of data samples [103], or procedural synthetic data generation [12, 71, 125]. One open question is whether PIC-LM can accommodate more complex forms of control; e.g., adaptively switching between contextual and parametric knowledge. Finally, the practical utility of PIC can be expanded to more sophisticated agentic setups, such as tool use [99] or multi-LLM collaboration [26].

## Acknowledgments

We thank Rui Xin for helping with OpenAI API evaluations, as well as Noah Smith, Tianyu Gao, Lucy He, Adithya Bhaskar, and the Princeton NLP Group for their helpful feedback. JH is supported by an NSF Graduate Research Fellowship, HY is supported by the William A. Dippel '50 *55 Graduate Fellowship, SL is supported by the Meta AIM program, and ZZ is supported by an Amazon AI Ph.D. Fellowship. This research was developed with funding from the Defense Advanced Research Projects Agency's (DARPA) SciFy program (Agreement No. HR00112520300). The views expressed are those of the author and do not reflect the official policy or position of the Department of Defense or the U.S. Government. This material is based upon work supported by the Defense Advanced Research Projects Agency and the Air Force Research Laboratory, contract number(s): FA8650-23-C-7316. Any opinions, findings and conclusions, or recommendations expressed in this material are those of the author(s) and do not necessarily reflect the views of AFRL or DARPA. Further, this work is also supported by the Singapore National Research Foundation and the National AI Group in the Singapore Ministry of Digital Development and Information under the AI Visiting Professorship Programme (award number AIVP-2024-001), as well as the AI2050 program at Schmidt Sciences.

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

# Appendix

# A   General Information

## A.1   Released artifacts

We release all artifacts (PIC-Bench, PIC-LM 8B models, and related data processing, training, and evaluation scripts) for reproducibility and future development:

| | | |
|---|---|---|
| **Codebase** | ⦿ | `jacqueline-he/precise-information-control` |
| **PIC-Bench Data** | 🤗 | `jacquelinehe/pic-bench` |
| **PIC-LM 8B** | 🤗 | `jacquelinehe/Llama-3.1-PIC-LM-8B` |
| **PIC-LM 8B - SFT** | 🤗 | `jacquelinehe/Llama-3.1-PIC-LM-8B-SFT` |
| **PIC-LM SFT Data** | 🤗 | `jacquelinehe/pic-lm-sft-mixture` |
| **PIC-LM Preference Data** | 🤗 | `jacquelinehe/pic-lm-preference-mixture` |

## A.2   Related work

**Hallucination in LMs.**   Hallucination is a byproduct of the standard language modeling objective: LMs extract statistical patterns from training corpora, and learn to generate semantically fluent text that is not necessarily accurate with respect to a knowledge source [43]. A profusion of recent scholarship has focused on factuality and extrinsic hallucination, in particular, its root cause identification [29, 47, 30], detection [20, 73], evaluation [112, 77, 105, 56, 120], and mitigation [44, 53, 9, 13, 6, 113].[7] Early faithfulness hallucination studies are largely task-specific, with efforts on abstractive summarization [75, 100, 129, 25] or knowledge-grounded dialogue generation [39, 95, 22, 1].

PIC-Bench evaluates faithfulness hallucination in long-form generation via claim decomposition, in a similar manner as recent long-form evaluation on factual hallucination [77, 105, 120]. Among faithfulness hallucination benchmarks, RoSE [66] considers generations at the level of *atomic content units* for summarization evaluation. FaithEval [78] operates on short-form QA, while we consider a range of long-form tasks. Closest to our work is FACTS Grounding [42], which we incorporate as a partial task in PIC-Bench. Their evaluation relies on coarse, holistic assessment by prompting multiple judge LLMs, instead of a granular per-claim verification. Furthermore, PIC imposes stricter criteria via the full PIC setting by requiring LMs to ground *all* generated content in provided context, instead of using a binary notion of faithfulness.

Evaluation aside, there are many contemporary efforts to alleviate LM hallucination via post-training. PIC-LM follows a conventional alignment procedure, similar to recent factuality approaches [40, 63, 113] that use API-based automatic evaluators for preference data construction [77, 120]. In contrast, PIC-LM leverages a more cost-effective reward signal and targets faithfulness hallucination. Several works improve context-faithfulness through other means, either via prompting [128], decoding modifications [102], or fine-tuning on counterfactual data [67, 55, 83]. PIC-LM is not deliberately trained on counterfactual data, but nevertheless generalizes well to counterfactual evaluations.

**RAG and Attributable Generation.**   PIC shares surface similarities with both RAG and LM attribution, a task in which every statement in generated text must be ascribed to appropriate source references using inline citation markers [27, 64, 126]. However, while the principal goal of RAG and LM attribution is to improve factuality, PIC is fundamentally an information control problem: ensuring that every output claim is traceable to contextual knowledge. This motivates a dedicated evaluation framework, benchmark, and training techniques.

While all three paradigms require grounded responses, we note several key differences. First, PIC enforces full coverage of all input claims, while RAG allows for information omission so long as final answers are correct. Second, attribution systems often work at the sentence or passage level, while PIC operates at the level of verifiable claims, enabling a finer-grained detection of distortion, fabrication, and conflation. Lastly, while RAG and LM attribution primarily focus on QA, PIC is more univerally applicable to a range of generative tasks.

---

[7]We follow the newer and more granular taxonomy of LLM hallucinations proposed by Huang et al. [41], which categorizes them into *factuality* and *faithfulness* hallucinations. Earlier work instead used the terms *intrinsic* and *extrinsic*, where intrinsic hallucinations directly contradict the source context, and extrinsic hallucinations cannot be verified using either the input context or an external knowledge base [43].

**Controllability in LMs.** Controllability is a fundamental concept that originates from control theory. It describes a system's ability to transition from any initial state to any final state through appropriate external inputs [85, 55]. This is a valuable property for LMs, as many user-centric applications require particular control over elicited output [127]. For example, a dialogue generation task may involve preferred levels of attributes such as emotion, formality, or toxicity. Early efforts to steer LM generation include training with discrete control codes [48, 95] to adjust style, content, and other task-specific behaviors, or modifying the output logit distribution at decoding time [124].

In recent years, instruction fine-tuning has emerged as the de facto paradigm to align LMs with user intents [79, 87, 98, 116, 118, 68]. LMs must learn to implicitly map natural language instructions to desired output behaviors, rather than relying on special tokens to trigger actions. While instruction-tuning expands the expressivity of the constraint space, it also fosters greater ambiguity. LMs may struggle to follow commands, even in the presence of explicit cues. For example, Goyal et al. [32] find that supervised fine-tuning on instruction datasets diminish the LM's contextual reliance ability. They theoretically show that non-context-dependent data points eventually dominate loss gradients during instruction fine-tuning. Our work corroborates that this problem is data-driven: by fine-tuning on instruction pairs that require strict claim adherence, PIC-LM maintains strong contextual grounding. Further, DPO and related preference optimization algorithms more explicitly reinforce this behavior, ensuring a performance margin beyond what supervised fine-tuning can achieve alone.

## A.3 Limitations

A consistent through-line is that user-provided knowledge should supersede the LM's inherent priors. Users ought to exercise full control in establishing the ground truth framework for LM generation, which is an extreme stance that may not be appropriate in every downstream scenario.

PIC imposes strong input assumptions—for instance, that the set of input claims should be well-structured and exactly aligned with what constitutes an ideal response. The LM should anchor its response on user-given claims, even if this means forgoing extra correct details, or overriding the model's internal beliefs. Enforcing these constraints is necessary in fast-moving, high-stakes domains (e.g., scientific consensus on infectious diseases), where plausible but unverified information pose serious risks. That LMs cannot completely eliminate faithfulness hallucination on a simple and well-defined setting is significant. Further, we provide case studies illustrating how PIC improves factual accuracy when situated within a larger pipeline in which context is fetched, verified, or filtered beforehand. Despite its limited standalone utility, PIC-LM demonstrates potential as a reliable component in modular systems.

Given the time and cost it takes to evaluate each generation sample via claim decomposition, our evaluation sets are individually quite small (each under 250 samples). This is par for the course: our testbed sizes are comparable to those used in long-form factuality literature [77, 63, 105, 113]. Unlike prior work, our task selection extends beyond QA; it is thorough, but by no means exhaustive. In line with Jacovi et al. [42], we do not consider multi-step logic, intensive reasoning, or creative generation.

## A.4 Broader impacts

Our work addresses a foundational aspect of language models by enabling a stronger command over the claims that LMs produce. PIC-LM serves as a reliable vehicle for transforming text without introducing intrinsic hallucinations, and improves factual accuracy in downstream applications. This holds important ramifications as LMs are the workhorse behind modern generative search engines [64]. Our work addresses a foundational aspect of language models by enabling a stronger command over the claims that LMs produce. PIC-LM serves as a reliable vehicle for transforming text without introducing faithful hallucinations, and improves factual accuracy in downstream applications. This holds important ramifications as LMs are the workhorse behind modern generative search engines [64].

However, greater generation control may also raise the likelihood of misuse. PIC-LM is more liable to emitting counterfactual, toxic, or harmful content when presented with certain contexts, bypassing abstention behaviors imbued during RLHF. While sub-optimal, we argue that this tradeoff should be contextualized. Uncontrolled toxic generation, arising from model hallucinations, can be harder to detect and filter, which may be arguably worse. Recent work shows that LM safety guardrails are often superficial and can be easily bypassed [90], and that, despite extensive safety alignment, conventional LMs may still generate adversarial outputs rooted in opaque model internals. In contrast,

PIC-LM shifts the safety locus from the model's parameters to user inputs, making the source of information transparent. Doing so facilitates a more controlled and manageable environment where safeguards can be designed around input validation and user intent. Therefore, while PIC-LM may not be the safest model in isolation, we see it as a promising building block for safety-aware systems where user-grounded control is critical.

# B  PIC-Bench Details

## B.1  PIC-Bench task details

**Task information.** Table 7 shows PIC-Bench task information, and Fig. 2 shows the claim count distributions for all tasks.

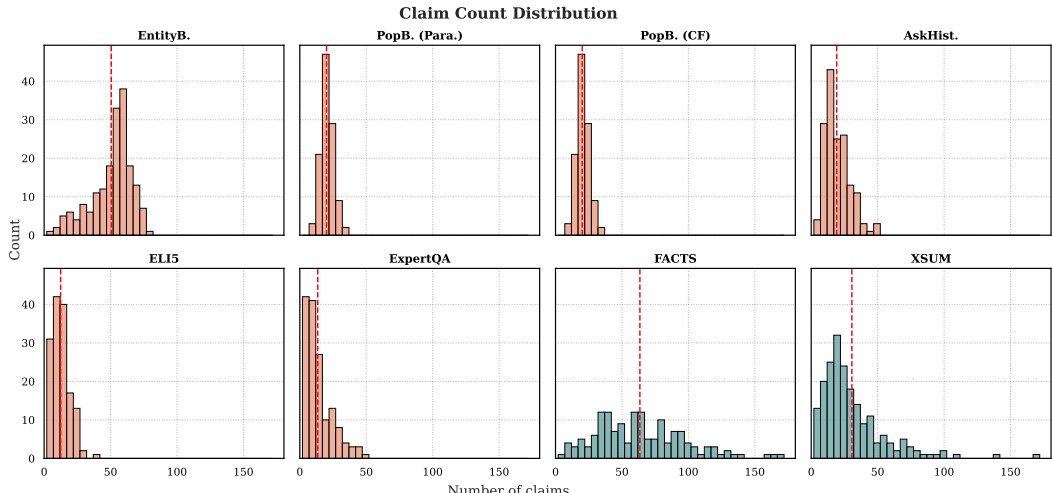

Figure 2: Claim count distribution histograms under the full PIC (orange) and the partial PIC (teal) setting. The red dotted line indicates the average number of claims. Note that PopB.-$P_{PIC}$ and PopB.-$CF_{PIC}$ share the same distribution.

**Dataset details.** Many of these datasets are originally introduced for long-form factuality [77, 72, 105], a dimension we do not measure. Rather, our PIC metric seeks to ask: how faithfully can LMs incorporate given information—either partially or in full? Even under ideal conditions (i.e., when the claims constitute the gold response), we identify a meaningful faithfulness gap in existing LMs. We describe each dataset in greater detail below:

Entity Bios consists of named entities from Min et al. [77]; we obtain biographical context information about each entity from relevant Wikipedia pages. For each given `entity`, the corresponding instruction is: "Generate a factual biography about `entity`."

Popular Bios is a biography generation task of famous real-world entities across various domains, e.g., art, science, or literature [32]. In this dataset, each sample contains both a parametric entity and a counterfactual entity (who is also a famous individual in a different field). Each entity is slotted into a a biographical query, e.g., "Generate a biography on {entity}, the scientist who discovered electromagnetism and electrochemistry," for which the LM must rely on contextual cues to describe a specific biographical entity. For a controlled study, we consider both parametric (e.g., "Michael Faraday") and counterfactual (e.g., "Nelson Mandela") settings.

ELI5 (formally, Explain Like I'm Five) consists of human-authored questions and responses sourced from the Reddit forum[8], in which users ask for simple explanations to complex questions or topics. We use a subsampled set from [105].

---

[8]`reddit.com/r/explainlikeimfive`

Table 7: Dataset information for PIC-Bench. Average claim length is in words (split by whitespace).

| Dataset | Sample Count | Min. / Avg. / Max. # Claims | Avg. Claim Len. | Data Provenance |
|---|---|---|---|---|
| ENTITYBIOS$_{PIC}$ [77] | 183 | 3 / 50.5 / 79 | 12.0 | Human (Search) |
| POPBIOS$_{PIC}$ [32] | 111 | 7 / 20.1 / 33 | 10.7 | Synthetic |
| ELI5$_{PIC}$ [123] | 146 | 4 / 12.5 / 40 | 10.6 | Human (Reddit) |
| ASKHISTORIANS$_{PIC}$ [123] | 158 | 6 / 19.2 / 49 | 12.1 | Human (Reddit) |
| EXPERTQA$_{PIC}$ [72] | 152 | 3 / 13.5 / 51 | 11.6 | Human |
| FACTS GROUNDING$_{PIC}$ [42] | 150 | 6 / 63.5 / 167 | 13.1 | Human |
| XSUM$_{PIC}$ [81] | 200 | 2 / 30.6 / 171 | 12.1 | Human |

AskHistorians consists of in-depth and comprehensive human-authored questions and responses from the Reddit forum[9] about various historical topics. We use a subsampled set from [105].

ExpertQA consists of expert-curated and expert-verified long-form QA pairs across 32 fields; we use a subsampled set [72].

FACTS Grounding is a long-context benchmark that evaluates the language model's ability to generate text that is accurate with respect to context documents of up to 32K tokens [42].

XSUM is a summarization dataset that contains news articles from the BBC [81]. It was originally introduced for extreme, 1-sentence summarization. In our case, for each sample, we convert the given article into a list of claims and provide the following instruction: "Summarize the following text, given as a list of claims, in around 20-25 words."

Where applicable (i.e., if the context is extracted from a human-authored gold response, such as from the Reddit tasks), we perform some lightweight quality processing by retaining only samples with a Prometheus score of above 4.

## B.2 PIC evaluation details

PIC evaluation consists of three parts: claim extraction, claim verification, and metric calculation.

### B.2.1 Claim extraction and verification.

We adopt the claim extraction and verification procedures introduced by Song et al. [105]. For claim extraction, we apply few-shot prompting to GPT-4o mini, employing a sliding window format (i.e., each sentence is supplied with its previous and successive sentences as context), such that the extracted claims are self-contained (i.e., grammatically and semantically unambiguous). To prevent artificially inflated scores, claims are de-duplicated before verification.

For claim verification, we prompt GPT-4o mini to determine whether each claim is supported by the original context, assigning one of two labels: `supported` or `unsupported`. One response claim may be supported by one or more context claims. In contrast to Song et al. [105], instead of verifying against a list of retrieved snippets from Google Search, we check against the list of input claims.

### B.2.2 Metric calculation.

We consider precision for the partial PIC setting and $F_1$ (formally, $F_1$@K) for the full PIC setting. Both metrics are computed following Wei et al. [120] and Song et al. [105], which we motivate below.

Assume a set of claims provided by the input as $C$, and a generated response $y$. Let $S(y)$ be the number of supported claims, and $N(y)$ be the number of unsupported claims from the response $y$. Then, one can compute the factual precision of a response as $\text{Prec}(y) = \frac{S(y)}{S(y)+N(y)}$, which is the final metric in the partial PIC setting.

In the full PIC setting, we would like to compute $F_1$@K, which balances the "supportedness" of the generated response with the coverage of verifiable claims in $C$.

---

[9]reddit.com/r/AskHistorians

Wei et al. [120] note that measuring factual recall is challenging, as it is impossible in most cases to come up with an exhaustive set of *all possible claims* that should be included in an ideal long-form response. They propose setting a hyperparameter $K$, which is some number of supported claims for which a user cares about recall up to the $K$-th supported fact.

The case for PIC is simpler: we only care that the generation incorporates exactly all claims in $C$, instead of an external corpus of real-world knowledge. Accordingly, we can set $K = |C|$. Let $|C_y|$ be the count of unique claims in $C$ that are also present in $y$. Then, the factual recall@K of $y$ is $R_K(y) = \min(\frac{|C_y|}{K}, 1)$.

$F_1$@K combines the precision and recall@K as

$$F_1@K = \begin{cases} \frac{2 \cdot \text{Prec}(y) \cdot R_K(y)}{\text{Prec}(y) + R_K(y)}, & \text{if } S(y) > 0, \\ 0, & \text{if } S(y) = 0. \end{cases}$$

### B.2.3 Choice of LLM backbone.

PIC evaluation relies on an LLM API that charges per-token for claim extraction and verification. The total cost is dependent on several factors, such as the number of extracted claims (as each extracted claim is verified in an independent API call), and the context length of input and output. In our experience, a full evaluation run of an open LM on PIC-Bench costs less than $7 USD when using GPT-4o mini as the LLM backbone, which we believe is reasonable under an academic budget.

To show that the PIC evaluation pipeline is relatively agnostic to different LLM backbones, we report the effect of additional LLM judges (GPT-4.1 and Claude 3.5 Sonnet) on average PIC-Bench metrics in Table 8. PIC scores are largely judge-agnostic, with GPT-4.1 behaving slightly stricter than the other judges. Given that GPT-4.1 and Claude 3.5 Sonnet impose API costs 13× and 20× higher respectively, we retain GPT-4o mini.

Table 8: Ablation of different LLM Judges on PIC-Bench. We report average $F_1$ and precision (in %) for each setting.

| LLM Judge | Evaluated Model | | |
| --- | --- | --- | --- |
| | Llama 3.1 8B Inst. | PIC-LM 8B | GPT-4o |
| GPT-4o mini | 69.1 / 73.6 | 89.9 / 91.2 | 90.5 / 90.2 |
| GPT-4.1 | 68.6 / 73.6 | 87.5 / 88.1 | 87.6 / 86.0 |
| Claude 3.5 Sonnet | 70.2 / 74.5 | 88.2 / 90.0 | 89.6 / 89.3 |

### B.2.4 PIC-Bench evaluation reliability.

To quantify statistical precision, we report 95% confidence intervals estimated via bootstrap resampling with 1000 replicates (Table 9). Across most settings, the confidence interval bands are relatively narrow and often within 2–4 percentage points, indicating that our PIC metrics are statistically stable under sample variability. For proprietary LMs, the intervals are especially tight (e.g., GPT-4o, EntityB.: 91.9, 93.35), reflecting high metric consistency.

In contrast, wider intervals appear in a small number of more challenging or lower-scoring tasks (e.g., Claude 3.5, PopB. (CF): 5.77, 9.99), suggesting greater variability in claim extraction or verification in those settings. Nevertheless, the overall narrowness of the CIs across tasks and systems corroborates that PIC evaluation can provide precise and trustworthy estimates of PIC performance.

### B.3 Inference settings

**Generation hyper-parameters.** We use the vLLM toolkit [51] for fast inference on all open, non-API-based LMs. For the non-reasoning models, we generate using greedy decoding (temperature set to 0) and a repetition penalty set to 1.2, as generation grows more deterministic at lower temperatures [35, 53]. For the 32B reasoning models, greedy decoding results in considerably degraded or repetitive output. Thus, we follow prescribed best practices: we enable the thinking

Table 9: 95% confidence intervals (CI) across all tasks and models on PIC-Bench. For each setting, we sample 1000 replicates with replacement from each evaluation set. All reasoning LMs (denoted with a †) are evaluated with thinking mode enabled.

| | EntityB. | PopB. (Para) | PopB. (CF) | AskHist. | ELI5 | ExpertQA | FACTS | XSUM |
|---|---|---|---|---|---|---|---|---|
| **Open-weight LMs (8B)** | | | | | | | | |
| Llama 3.1 8B Inst. | 72.72, 76.09 | 81.48, 85.39 | 20.20, 27.09 | 79.05, 84.33 | 74.20, 80.96 | 69.64, 77.41 | 65.49, 74.35 | 72.48, 81.73 |
| Tulu 3 8B | 78.41, 80.80 | 86.07, 89.45 | 47.23, 55.26 | 77.98, 82.43 | 78.99, 84.42 | 77.26, 83.85 | 71.44, 79.00 | 72.61, 81.43 |
| Ministral 8B Inst. | 77.37, 80.03 | 87.32, 90.66 | 59.72, 67.42 | 73.58, 79.81 | 74.80, 81.29 | 73.23, 81.72 | 70.33, 80.37 | 78.56, 86.54 |
| Hermes 3 8B | 73.35, 76.4 | 82.59, 87.09 | 39.81, 48.96 | 69.82, 75.86 | 72.76, 79.03 | 69.66, 78.4 | 69.63, 78.39 | 77.58, 84.87 |
| **Open-weight LMs (32B)** | | | | | | | | |
| Qwen 3 32B† | 71.22, 76.97 | 85.24, 90.78 | 43.19, 53.38 | 74.17, 80.58 | 62.96, 72.39 | 58.51, 69.07 | 59.84, 69.22 | 74.29, 82.42 |
| QwQ 32B† | 85.63, 87.75 | 86.13, 89.04 | 60.90, 67.40 | 77.33, 81.82 | 77.70, 82.57 | 77.61, 83.82 | 66.80, 74.07 | 84.14, 89.70 |
| R1 Distill 32B† | 81.06, 83.55 | 85.62, 89.13 | 54.78, 62.83 | 78.80, 83.48 | 79.04, 85.00 | 76.25, 83.64 | 65.92, 75.89 | 74.97, 84.01 |
| **Open-weight LMs (70B)** | | | | | | | | |
| Llama 3.3 70B Inst. | 88.12, 89.89 | 92.47, 94.46 | 41.51, 49.47 | 89.94, 92.34 | 88.59, 91.83 | 82.41, 88.75 | 75.01, 83.38 | 62.39, 75.25 |
| Tulu 3 70B | 83.74, 85.96 | 87.68, 90.54 | 48.48, 56.05 | 81.56, 85.68 | 80.08, 84.93 | 68.98, 77.43 | 42.33, 49.61 | 82.35, 88.45 |
| Hermes 3 70B | 79.96, 82.91 | 87.65, 91.34 | 53.29, 59.89 | 78.77, 84.10 | 76.69, 82.55 | 76.20, 83.18 | 74.06, 81.91 | 83.39, 90.16 |
| **Proprietary LMs** | | | | | | | | |
| Claude 3.5 | 86.08, 91.23 | 91.35, 94.10 | 5.77, 9.99 | 79.37, 87.68 | 81.69, 89.62 | 57.19, 70.45 | 76.10, 87.20 | 86.91, 93.43 |
| GPT-4o | 91.90, 93.35 | 92.08, 94.47 | 74.21, 79.15 | 95.27, 96.97 | 92.98, 96.32 | 87.16, 92.82 | 89.58, 94.58 | 89.92, 95.41 |
| ***Ours:* PIC-LM** | | | | | | | | |
| PIC-LM$_{SFT\ Only}$ | 85.75, 87.71 | 85.34, 88.88 | 69.17, 74.42 | 84.38, 88.78 | 79.16, 85.20 | 73.14, 81.38 | 75.98, 85.83 | 92.27, 95.56 |
| PIC-LM | 93.56, 94.65 | 95.12, 96.79 | 78.35, 83.07 | 94.05, 95.99 | 92.57, 95.45 | 81.94, 89.38 | 41.75, 49.23 | 95.50, 97.73 |

mode and use a temperature of 0.6, top-p of 0.85, top-k of 20, min-p of 0, a presence penalty of 1.5, and a repetition penalty of 1.2. We strip out intermediate thinking components prior to evaluation.

**Prompt templates.** For a strict and fair cross-model comparison, we use the same prompt template across all model baselines and tasks. Table 10 shows the prompts used throughout for PIC training and evaluation.

We explored alternative prompts and found PIC-Bench to be largely prompt-insensitive. For example, we re-ran evaluation baselines using an alternative instruction pair:

- **Full PIC**: `You are an assistant that strictly adheres to given instructions. Be sure to incorporate each provided input claim in your response, and do not introduce or omit any claims.`

- **Partial PIC**: `You are an assistant that strictly adheres to given instructions. Respond using any subset of given claims that you find relevant, but do not introduce any new claims.`

We report the absolute performance change (%) for PIC-LM 8B and three baselines (Llama 3.1 8B Inst., Tulu 3 8B, and Ministral 8B Inst.) in Table 11. As shown, changes in PIC performance are relatively small across prompt variations, with most metric shifts <4%. These results indicate that PIC-Bench evaluations remain stable across reasonable prompt variations. While certain prompts may slightly improve absolute scores, we believe they are unlikely to affect the relative ranking between models, which is our primary evaluation focus.

## B.4 Human evaluations

One reasonable concern is whether the automatic PIC evaluation corresponds well with human judgment. Specifically, an LLM is invoked at two stages of our evaluation pipeline: claim extraction (decomposing text into verifiable claims) and claim verification (checking that a claim is supported by a set of verifiable claims). For both stages, we conduct a pilot human evaluation experiment to assess agreement with human annotation across PIC tasks and three baseline models (Llama 3.1 8B Inst., PIC-LM 8B, and GPT-4o). We use three human annotators per component.

Table 10: PIC prompts used for full and partial settings.

| PIC Setting | Prompt |
|---|---|
| FULL | ```[INST] Complete the instruction and include **exactly all** of the following claims-no more, no fewer. Your response must be entirely grounded on the given context while also answering the question.

Instruction:
{instruction}
Claims:
{claims}
Response:
[/INST]``` |
| PARTIAL | ```[INST] Complete the instruction using **any subset** of the following claims, if they help your answer. Do not introduce any new claims beyond what's given. Make sure that your response is grounded on the given context, while also answering the question.

Instruction:
{instruction}
Claims:
{claims}
Response:
[/INST]``` |

Table 11: Absolute performance changes (%) for PIC evaluation on different model baselines, when evaluated using the original prompt (Table 10) versus an alternative prompt pair.

| PIC Metric (%) | LLAMA 3 8B INST. | TULU 3 8B | MINISTRAL 8B INST. | PIC-LM 8B |
|---|---|---|---|---|
| **Avg. F1** | 1.8 | 1.1 | -2.4 | 0.4 |
| **Avg. Perf. F1** | 0.4 | 0.0 | -0.4 | 3.2 |
| **Avg. Prec.** | 0.8 | 3.2 | 1.2 | 0.4 |
| **Avg. Perf. Prec.** | -0.3 | 0.1 | 3.0 | 4.2 |

**Claim extraction.** For claim extraction evaluation, we sample 5 generations per PIC task and baseline model, leading to 120 samples in total. For each sample, we show the annotator the generated output and an extracted claim from that output.

Recall that for claim extraction, the LLM is prompted to extract high-quality, verifiable claims from a source task. Here we ask the annotator to rate the extracted claim on a 3-point ordinal scale (0-2, 0 being the best) along the following dimensions: **faithfulness** (0: definitely supported, 1: partially supported, 2: not supported), **decontextualization** (0: self-contained, 1: ambiguous, 2: context-dependent), and **quality** (0: fluent, 1: minor readability issues, 2: hard to read).

From Table 12, we observe fairly strong LLM-human agreement (and substantial exact inter-annotator agreement) across all three axes, which confirms that our LLM judge (GPT-4o mini) is able to correctly extract high-quality, verifiable claims.

**Claim verification.** For claim verification evaluation, we sample 3 generations per PIC task and baseline, leading to 72 generations in total. For each sample, we provide the list of input claims, and a response claim that the LLM-based claim verifier has marked as either supported or unsupported. We ask the annotators, who are blind to model identity and the verifier's label, to judge if the response claim is supported or unsupported by the input claims.

Table 12: Human evaluation for PIC claim extraction. We report (i) LLM–human agreement (the average fraction of annotator ratings equal to the best score (0) for the LLM-extracted claims), and (ii) exact inter-annotator agreement (IAA; the fraction of items where all raters chose the same label).

| Dimension / Metric | LLM–Human Agreement | Exact IAA |
|---|---|---|
| **Faithfulness** | 97.5 | 94.1 |
| **Decontextualization** | 89.6 | 76.5 |
| **Quality** | 99.7 | 99.2 |

We found high average human agreement with the LLM (81.0%), and substantial inter-annotator agreement (a Fleiss' Kappa of 0.694).

## B.5 Few-shot baseline results

We show few-shot results for select baselines on PIC-Bench in Table 13 (full setting) and Table 14 (partial setting). Since we benchmark only instruction-tuned models that have specifically been trained to respond to human instructions in a zero-shot fashion, we do not consider few-shot exemplars to be necessary; still, we include them for comprehensiveness.

Few-shot exemplars lead to degraded performance in 32B open-weight reasoning models across all settings, likely due to the absence of explicit step-by-step reasoning, which these LMs are trained to leverage. For the non-reasoning models, few-shot exemplars generally help full PIC, but hurt partial PIC. In the full PIC setting, few-shot prompting consistently improves average PIC-scores across all baselines. This trend reverses in the partial setting: adding few-shot examples lowers average perfect-score precision, suggesting that exemplars may distract the LM. By definition, partial PIC samples include irrelevant claims that do not belong in the target response. Few-shot exemplars, especially if they reference irrelevant or noisy subsets of claims, may introduce further distraction and make it difficult for the LM to ground its generation on intended context.

Table 13: Few-shot PIC-Bench results (full setting) for select baselines. Metrics include $F_1$, the proportion of perfect $F_1$ (Perf.), and the average of both metrics. Best values in **bold**. For all metrics, the higher the better.

| | EntB$_{\text{PIC}}$ | | PopB-P$_{\text{PIC}}$ | | PopB-CF$_{\text{PIC}}$ | | AskH$_{\text{PIC}}$ | | ELI5$_{\text{PIC}}$ | | ExpQA$_{\text{PIC}}$ | | Avg. | |
|---|---|---|---|---|---|---|---|---|---|---|---|---|---|---|
| | $F_1$ | Perf. | $F_1$ | Perf. | $F_1$ | Perf. | $F_1$ | Perf. | $F_1$ | Perf. | $F_1$ | Perf. | $F_1$ | Perf. |
| **Open-weight LMs (8B) + 2-shot** | | | | | | | | | | | | | | |
| LLAMA 3.1 8B INST. | 72.4 | 1.1 | 89.8 | 15.3 | 22.2 | 0.0 | 80.2 | 1.3 | 80.0 | 11.6 | 77.1 | 11.8 | 70.3 | 6.9 |
| TULU 3 8B | 78.1 | 2.2 | 91.7 | 10.8 | 61.4 | 0.9 | 82.5 | 1.3 | 84.0 | 13.0 | 78.3 | 11.8 | 79.3 | 6.7 |
| MINISTRAL 8B INST. | 76.5 | 1.1 | 83.4 | 16.2 | 73.4 | 0.9 | 82.3 | 4.4 | 79.5 | 12.3 | 80.6 | 15.1 | 80.9 | 8.4 |
| **Open-weight reasoning LMs (32B) + 2-shot** | | | | | | | | | | | | | | |
| QWEN 3 32B | 46.4 | 2.2 | 3.33 | 0 | 1.41 | 0 | 8.24 | 0 | 23.4 | 0.7 | 16.3 | 0.7 | 16.5 | 0.6 |
| QwQ 32B | 76.6 | 4.4 | 58.4 | 8.1 | 28.1 | 0.9 | 47.6 | 3.8 | 47.3 | 6.9 | 39.5 | 7.2 | 49.6 | 5.2 |
| R1 DISTILL 32B | 85.7 | 3.8 | 57.3 | 13.5 | 42.2 | 0.9 | 39.9 | 1.3 | 54.4 | 6.9 | 45.1 | 11.8 | 54.1 | 6.4 |
| **Proprietary LMs + 2-shot** | | | | | | | | | | | | | | |
| CLAUDE 3.5 | **92.4** | **10.9** | 94.4 | 27.9 | 25.4 | 0.9 | 94.4 | 26.0 | 92.8 | 43.2 | 90.2 | 44.7 | 81.6 | 25.6 |
| GPT-4o | 91.1 | 6.0 | **94.7** | **30.6** | **79.5** | **3.6** | **95.4** | **41.1** | **93.7** | **49.3** | **91.5** | **52.0** | **91.0** | **30.4** |

## B.6 Thinking mode ablations

Reasoning language models are trained to perform explicit chain-of-thought reasoning steps [119], often through the use of special control tokens that signal the model to engage in intermediate "thinking" steps before producing a final answer [109]. We ablate the effect of the thinking mode on PIC-Bench with the 32B reasoning LMs (Qwen 3 32B, QwQ 32B, R1 Distill 32B). In the non-thinking

Table 14: 2-shot PIC-Bench results (partial setting) for select baselines. Metrics include precision (Prec.), the proportion of perfect precision (Perf.), and the average of both metrics (right-most column). Best values in **bold**. For all metrics, the higher the better.

| | FACTS$_{PIC}$ | | XSUM$_{PIC}$ | | Avg | Avg |
|---|---|---|---|---|---|---|
| | Prec. | Perf. | Prec. | Perf. | Prec. | Perf. |
| **Open-weight LMs (8B) + 2-shot** | | | | | | |
| LLAMA 3.1 8B INST. | 66.6 | 14.0 | 70.4 | 48.0 | 68.5 | 31.0 |
| TULU 3 8B | 68.8 | 5.3 | 59.4 | 39.5 | 64.1 | 22.4 |
| MINISTRAL 8B INST. | 72.6 | 21.3 | 75.7 | 60.0 | 74.1 | 40.7 |
| **Open-weight reasoning LMs (32B) + 2-shot** | | | | | | |
| QWEN 3 32B | 63.3 | 25.3 | 5.9 | 1.5 | 34.6 | 13.4 |
| QwQ 32B | 73.7 | 12 | 75.5 | 50 | 74.6 | 31.0 |
| R1 DISTILL 32B | 78.1 | 22.6 | 41.6 | 27 | 59.9 | 24.8 |
| **Proprietary LMs + 2-shot** | | | | | | |
| CLAUDE 3.5 | 86.4 | **49.3** | 89.6 | 73.0 | 88.0 | 61.2 |
| GPT-4o | **87.3** | 46.7 | **91.0** | **81.0** | **89.1** | **63.8** |

mode, we decode using a temperature of 0.7, top-p of 0.8, top-k of 20, min-p of 0, a presence penalty of 1.5, and a repetition penalty of 1.2 [109].

From Table 15, we observe that the thinking mode substantially improves PIC performance across nearly all tasks. Notably, every full PIC task benefits from the thinking mode, and XSUM$_{PIC}$ is the only task where the effect is mixed. While further exploration is warranted, we hypothesize that the thinking mode likely helps on PIC, a constrained factual recall task, as it enforces the LM's decoding behavior to be more deliberate, structured, and less prone to shallow pattern completion, minimizing fluency bias.

Table 15: Comparison of 32B reasoning LMs on PIC-Bench with and without thinking mode enabled (T = Thinking, NT = Non-thinking).

| | EntB$_{PIC}$ | | PopB-P$_{PIC}$ | | PopB-CF$_{PIC}$ | | AskH$_{PIC}$ | | ELI5$_{PIC}$ | | ExpQA$_{PIC}$ | | FACTS$_{PIC}$ | | XSUM$_{PIC}$ | |
|---|---|---|---|---|---|---|---|---|---|---|---|---|---|---|---|---|
| | $F_1$ | Perf. | $F_1$ | Perf. | $F_1$ | Perf. | $F_1$ | Perf. | $F_1$ | Perf. | $F_1$ | Perf. | Prec. | Perf. | Prec. | Perf. |
| **Open-weight LMs (32B)** | | | | | | | | | | | | | | | | |
| QWEN 3 32B (T) | 74.1 | 2.7 | 88.2 | 18.9 | 48.2 | 0.9 | 77.5 | 7.0 | 67.7 | 6.9 | 63.6 | 7.2 | 64.5 | 17.3 | 78.5 | 46.0 |
| QWEN 3 32B (NT) | 34.6 | 0.0 | 60.4 | 0.9 | 27.2 | 0.0 | 41.8 | 1.9 | 43.2 | 4.8 | 31.7 | 4.0 | 38.4 | 28.0 | 84.3 | 64.5 |
| QwQ 32B (T) | 86.7 | 1.6 | 87.6 | 2.7 | 64.1 | 1.8 | 79.6 | 3.8 | 80.0 | 6.9 | 80.9 | 12.5 | 70.3 | 12.7 | 86.9 | 58.5 |
| QwQ 32B (NT) | 62.5 | 0.6 | 74.8 | 3.6 | 26.5 | 0.0 | 73.2 | 3.2 | 71.0 | 2.7 | 69.5 | 4.0 | 68.5 | 5.3 | 81.5 | 60.0 |
| R1-QWEN-32B (T) | 82.3 | 1.1 | 87.5 | 9.0 | 58.8 | 0.9 | 81.1 | 4.4 | 82.2 | 10.3 | 80.0 | 13.2 | 78.6 | 6.5 | 71.1 | 26.7 |
| R1-QWEN-32B (NT) | 71.9 | 1.6 | 86.0 | 7.2 | 41.0 | 0.0 | 77.1 | 1.9 | 73.7 | 0.7 | 76.4 | 6.6 | 73.6 | 8.0 | 87.3 | 17.5 |

## B.7 Qualitative task examples

Example PIC-Bench instructions for each task are shown in Table 16, Table 17, Table 18, Table 19, Table 20, Table 21, Table 22.

## B.8 Llama 3 scaling analysis

To study the effect of PIC on a single model family, we run five sizes (1B, 3B, 8B, 70B, 405B) of the Llama 3 model family on PIC-Bench. Fig. 3 shows scaling trends. We observe that full PIC exhibits an inverse U-shaped performance curve, with $F_1$ peaking at the 70B mark, suggesting that full PIC

Table 16: ENTITYBIOS_PIC example.

**Instruction**

Generate a factual biography about Donald Featherstone.

---

**Context Claims**

- Donald Featherstone was born on January 25, 1936.
- Don Featherstone's signature was removed from the pink plastic flamingo in 2001.
- Don Featherstone kept 57 plastic flamingos on his back lawn.
- The color pink was popular in 1958.
- The pink flamingo went on sale in 1958.
- Don Featherstone held the position of president of Union Products until he retired in 2000.
- Plastic flamingos appeared across the United States.
- The plastic flamingo became more popular over time.
- Don Featherstone grew up in Berlin, Massachusetts.
- Don Featherstone resided in Fitchburg, Massachusetts.
- Don Featherstone was born in Worcester, Massachusetts, in 1936.
- In 2010, Cado Products purchased the plastic molds for the pink flamingos.
- Donald Featherstone was an American artist.
- Donald Featherstone died on June 22, 2015.
- A New York company purchased the molds for Don Featherstone's flamingos.
- The New York company subcontracted production of the flamingos to Cado Products, a company located in Fitchburg.
- Donald Featherstone is most widely known for his creation of the plastic pink flamingo in 1957.
- Don Featherstone's signature stayed on the pink plastic flamingo until 2001.
- In November 2006, Union Products closed.
- Plastic flamingos were included as parts of various art exhibits.
- In November 2006, production of the flamingo stopped.
- Don Featherstone sculpted over 750 different items while at Union Products, Inc.
- Don Featherstone died from Lewy body dementia.
- Cado Products continues to manufacture pink flamingos.
- Don Featherstone began his tenure as president of Union Products in 1996.
- In 1996, Don Featherstone was awarded the Ig Nobel Art Prize for his creation of the pink flamingo.
- Don Featherstone died on June 22, 2015.
- In 2010, Cado Products purchased the copyrights for the pink flamingos.
- Donald Featherstone created the plastic pink flamingo while working for Union Products.
- After graduating in 1957, Don Featherstone was offered a job designing three-dimensional animals for Union Products, Inc.
- The first two items sculpted by Don Featherstone were a girl with a water can and a boy with a dog.
- Don Featherstone based his creation of the pink flamingo on photographs of flamingos from National Geographic.
- Don Featherstone was not able to obtain real flamingos to use as models for his creation.
- In 1987, Donald Featherstone inscribed his signature in the original plastic mold of the plastic flamingo.
- The replacement of the signature was due to a small boycott of the unsigned pink plastic flamingos.
- In 1957, Don Featherstone was asked to carve a flamingo.
- Don Featherstone graduated from the Worcester Art Museum's art school in 1957.
- Don Featherstone was 79 years old at the time of his death.

Table 17: PopBios-P$_{PIC}$ and PopBios-CF$_{PIC}$ examples.

**Instruction**
Generate a biography on {entity}, the author who wrote *War and Peace*, *Anna Karenina*.

- - - - - - - - - - - - - - - - - - - - - - - - - - - - - - - - - - - - - - - - - - - - - - - - - -

**Context Claims (Parametric Entity: Leo Tolstoy)**

- The publication of *War and Peace* was a significant event in the career of Leo Tolstoy.
- *War and Peace* is a monumental novel.
- *War and Peace* captured the essence of Russian society during the Napoleonic Wars.
- The novel *War and Peace* had immediate success.
- The success of *War and Peace* cemented the reputation of {entity} as a literary genius.
- *Anna Karenina* is a novel that explores themes of love, betrayal, and the search for meaning.
- *Anna Karenina* solidified Leo Tolstoy's place in the literary pantheon.
- The opening line of *Anna Karenina* is "All happy families are alike; each unhappy family is unhappy in its own way."
- The opening line of *Anna Karenina* became iconic.
- The opening line of *Anna Karenina* encapsulates the insight of Leo Tolstoy into human nature.
- *Anna Karenina* is a novel that critiques the rigid structures of Russian society.
- *Anna Karenina* serves as a social commentary on the consequences of defying societal norms.

**Context Claims (Counterfactual Entity: James Clerk Maxwell)**

- The publication of *War and Peace* was a significant event in the career of James Clerk Maxwell.
- *War and Peace* is a monumental novel.
- *War and Peace* captured the essence of Russian society during the Napoleonic Wars.
- The novel *War and Peace* had immediate success.
- The success of *War and Peace* cemented the reputation of {entity} as a literary genius.
- *Anna Karenina* is a novel that explores themes of love, betrayal, and the search for meaning.
- *Anna Karenina* solidified James Clerk Maxwell's place in the literary pantheon.
- The opening line of *Anna Karenina* is "All happy families are alike; each unhappy family is unhappy in its own way."
- The opening line of *Anna Karenina* became iconic.
- The opening line of *Anna Karenina* encapsulates the insight of James Clerk Maxwell into human nature.
- *Anna Karenina* is a novel that critiques the rigid structures of Russian society.
- *Anna Karenina* serves as a social commentary on the consequences of defying societal norms.

capability does not monotonically scale with model size. Conversely, partial PIC performance strictly improves with model size, with the largest increase between the 3B to 8B range.

## B.9 More PIC-Bench plots

Fig. 4 shows precision vs. recall heatmaps for full PIC baselines. We observe that GPT-4o and Claude 3.5 achieve consistently high precision and recall across all tasks, while open-weight LMs lean conservative, with lower recall (this is especially apparent on EntityBios$_{PIC}$, a task with 50 claims on average). PopBios-CF$_{PIC}$ is the most challenging task, with deflated scores across all settings.

Fig. 5 shows the distribution of supported and unsupported claims for all PIC tasks. To note some trends: Llama 3.1 8B Instruct is the most verbose, generating the most claims by a clear margin in all tasks except EntityBios$_{PIC}$ and FACTS$_{PIC}$. Proprietary LMs display a lower proportion of unsupported claims.

Table 18: AskHistoriansPIC example.

**Instruction**

Answer the following question: I've seen Japanese artwork from the Edo era and before depicting Tigers. Did tigers ever inhabit the islands of Japan? If not, how might a Japanese person encounter a tiger before the Meiji era?

---

**Context Claims**

- Tigers never inhabited the islands of Japan.
- Japanese people became familiar with the image of tigers through cultural exchanges with China.
- Japanese people became familiar with the symbolism of tigers through cultural exchanges with China.
- In Chinese culture, tigers are featured as one of the twelve signs of the zodiac.
- In Chinese culture, tigers are one of the four mythical creatures known as the *shisho*.
- In Chinese symbolism, the tiger represents the west.
- In Chinese symbolism, the tiger represents autumn.
- In Chinese symbolism, the tiger represents the metal element.
- In Chinese symbolism, the dragon represents the east.
- In Chinese symbolism, the dragon represents the heavens.
- The duality of the tiger and the dragon is celebrated in ancient texts like the *I Ching*.
- In Chinese culture, dragons are associated with prosperity.
- In Chinese culture, dragons are associated with divine rulers.
- In Chinese culture, tigers symbolize earthly power.
- In Chinese culture, tigers symbolize agility.
- Japan adopted many cultural elements from China from as early as the 1st century CE.
- The symbolic significance of tigers was one of the cultural elements adopted by Japan from China.
- Cultural importation in Japan involved exchanges of Buddhist beliefs.
- Cultural importation in Japan involved exchanges of Daoist beliefs.
- Cultural importation in Japan involved exchanges of Confucian beliefs.
- Cultural importation in Japan involved practices like yin-yang (*onmyōdō*).
- In Japan, tigers came to symbolize military might and authority.
- In Japan, tigers were adopted by warrior elites to represent strength.
- The *Mahasattva Jataka* is a Buddhist narrative.
- Chinese-style ink paintings were brought to Japan by Zen emissaries.
- Zen emissaries brought Chinese-style ink paintings to Japan.
- The *Mahasattva Jataka* enriched the tiger's symbolic presence in Japanese art and culture.
- Chinese-style ink paintings enriched the tiger's symbolic presence in Japanese art and culture.
- Toyotomi Hideyoshi led Korean invasions in the late 16th century.
- Japanese artists predominantly relied on existing Chinese representations when depicting tigers during the Edo era.
- Tigers were never native to Japan.
- Tigers played a significant role in Japanese art and culture.
- Chinese traditions strongly influenced Japanese art and culture.

Table 19: ELI5_PIC example.

---

**Instruction**

How come horses were so common all over the world in the past?

I recently watched a documentary about a brief history of each continent when I noticed horses were really common everywhere for an example Genghis Khan in Asia, Saladin in Africa, natives in North and South America and all over Europe too. I know they've been brought with settlers to Australia because it was mentioned in the documentary but what about the other continents?

- - - - - - - - - - - - - - - - - - - - - - - - - - - - - - - - - - - - - - - - - - - - - - - - - - - -

**Context Claims**

- Horses evolved in North America.
- Horses migrated via the Bering Land Bridge to Eurasia approximately two million years ago.
- Horses disappeared from the Americas around 10,000 years ago.
- Horses were reintroduced to the Americas by European settlers in more recent history.
- Horses were domesticated in Eurasia around 4,000 to 3,500 BCE.
- Horses were domesticated on the Central Asian steppes.
- Breeding and selection led to larger horses.
- Breeding and selection led to more robust horses.
- Larger and more robust horses contributed to their indispensability for travel.
- Larger and more robust horses contributed to their indispensability for trade.
- Larger and more robust horses contributed to their indispensability for communication.
- Larger and more robust horses contributed to their indispensability for agriculture.
- Larger and more robust horses contributed to their indispensability for warfare.
- Genghis Khan in Asia relied heavily on horses for his military campaigns.
- Saladin in Africa relied heavily on horses for his military campaigns.
- Elephants require large amounts of food.

---

## C  PIC-LM Training Details

### C.1  SFT data curation

To construct our training dataset, we consider instruction fine-tuning datasets assembled from the following general sources: No Robots [93], FLAN [118], as well as task-specific sources: entity biography generation [77], FACTS Grounding [42], and CNN-DailyMail news text summarization [101] (starting from a random subsample of 2000 examples). If the task does not provide a gold response, we use GPT-4o to generate a synthetic gold response since it is already pretty good at average PIC. For the training sets that are in-domain to our PIC benchmark (FACTS and biography generation), we ensure that no train-test overlap exists.

We apply the following data processing pipeline:

1. Determine the PIC type based on the instruction. We do not consider samples that are creative (i.e., with a low density of verifiable claims) in nature or reasoning-intensive (i.e., math or coding problems), and manually determine by label wherever possible.

2. Filter out samples with responses under $128$ characters (as our focal point is the long-form generation setting).

3. Extract the claim list $C$ from the response for full PIC samples, and from the provided context for partial PIC samples.

4. Given that the claim extraction step might be imperfect, and we find that this happens especially if the response is vaguely worded or in another language, we compute precision, recall and $F_1$ against the list of extracted claims and the gold response. We only retain samples with a non-empty claim set and a sufficiently high precision or $F_1$ threshold

Table 20: EXPERTQA_PIC example.

---

**Instruction**

Why did art focused on naturalism and realistic notions (such as proportions and perspective) appear mainly in Europe?

- - - - - - - - - - - - - - - - - - - - - - - - - - - - - - - - - - - - - - - - - - - - - - - -

**Context Claims**

- The rediscovery of classical Greek and Roman art and literature occurred during the Renaissance.
- The humanist movement emphasized human experience and individuality.
- The humanist movement led to a renewed interest in realistic representations of the human figure.
- The humanist movement led to a renewed interest in spatial depth in paintings.
- The humanist movement led to a renewed interest in accurate portrayals of nature.
- Wealthy individuals provided patronage in the arts.
- Religious institutions provided patronage in the arts.
- Political entities provided patronage in the arts.
- Patronage in the arts encouraged exploration of new artistic techniques.
- Patronage in the arts encouraged exploration of new artistic subjects.
- Artists began to experiment with new materials such as oil paint.
- Artists began to experiment with new materials such as canvas.
- Oil paint allowed artists to achieve greater detail in their works.
- Canvas allowed artists to achieve greater realism in their works.
- The creation of linear perspective is attributed to the architect Filippo Brunelleschi.
- Linear perspective transformed the way artists depicted space in their paintings.
- Linear perspective transformed the way artists depicted depth in their paintings.
- The growth of scientific inquiry contributed to the appearance of naturalism in European art.
- The growth of empirical observation contributed to the appearance of realistic notions in European art.
- Artists' workshops and academies emphasized a systematic study of anatomy.
- Artists' workshops and academies emphasized a systematic study of light.
- Artists' workshops and academies emphasized a systematic study of perspective.
- The systematic study of anatomy, light, and perspective contributed to the advancement of naturalistic techniques in art.
- Naturalism and realistic notions in European art emerged due to the rediscovery of classical art and literature during the Renaissance.
- The rise of humanism contributed to the emergence of naturalism and realistic notions in European art.
- The patronage system contributed to the emergence of naturalism and realistic notions in European art.
- Advancements in artistic techniques contributed to the emergence of naturalism and realistic notions in European art.
- The growth of scientific inquiry contributed to the emergence of naturalism and realistic notions in European art.

Table 21: FACTSPIC example.

**Instruction**

Isn't the theater required to provide me with an initial set-up of tea bags in my housing if I am an actor working under the terms of this contract?

- - - - - - - - - - - - - - - - - - - - - - - - - - - - - - - - - - - - - - - - - - - - - - - - - - - - - - -

**Context Claims**

- Each Actor's housing shall be supplied with a bed and mattress in good condition.
- Each Actor's housing shall be supplied with a nightstand.
- Each Actor's housing shall be supplied with a reading lamp.
- Each Actor's housing shall be supplied with an armchair or sofa.
- Each Actor's housing shall be supplied with a table and chairs.
- Each Actor's housing shall be supplied with a lamp.
- Each Actor's housing shall be supplied with a dresser.
- Each Actor's housing shall be supplied with a mirror.
- Each Actor's housing shall be supplied with hangers.
- Each Actor's housing shall be supplied with linens and towels.
- Each Actor's housing shall be supplied with pillows.
- Each Actor's housing shall be supplied with blankets.
- Each Actor's housing shall be supplied with a wastebasket.
- Each Actor's housing shall be supplied with a radio alarm clock.
- Each Actor's housing shall be supplied with a television and cable, where available and necessary for adequate reception.
- The Theatre shall make available irons.
- The Theatre shall make available ironing boards.
- Each Actor's housing shall be supplied with pots and pans with lids.
- Each Actor's housing shall be supplied with cooking utensils.
- Each Actor's housing shall be supplied with silverware for four.
- Each Actor's housing shall be supplied with not fewer than four plates.
- Each Actor's housing shall be supplied with cups and glasses.
- Each Actor's housing shall be supplied with a can opener.
- Each Actor's housing shall be supplied with kitchen knives.
- Each Actor's housing shall be supplied with a colander.
- The Theatre shall provide an initial set-up of toilet paper in the Actor's housing prior to the Actor's arrival.
- The Theatre shall provide an initial set-up of paper towels in the Actor's housing prior to the Actor's arrival.
- The Theatre shall provide an initial set-up of hand soap in the Actor's housing prior to the Actor's arrival.
- The Theatre shall provide an initial set-up of dish soap in the Actor's housing prior to the Actor's arrival.
- The Theatre shall provide an initial set-up of salt and pepper in the Actor's housing prior to the Actor's arrival.
- The Theatre shall provide an initial set-up of sugar in the Actor's housing prior to the Actor's arrival.
- The Theatre shall provide an initial set-up of coffee in the Actor's housing prior to the Actor's arrival.
- The Theatre shall provide an initial set-up of tea in the Actor's housing prior to the Actor's arrival.
- The Theatre shall provide an initial set-up of garbage bags in the Actor's housing prior to the Actor's arrival.
- The Theatre shall provide an initial set-up of a sponge in the Actor's housing prior to the Actor's arrival.

Table 22: XSUMPIC example.

**Instruction**
Summarize the following text, given as a list of claims, in around 20-25 words.

---

**Context Claims**

- Many developing countries will try to curb carbon emissions by setting aside forested areas as reserves.
- Designating forest reserves in Liberia could displace as many as 1.3 million people.
- Designating forest reserves in the Democratic Republic of Congo could displace as many as 1.3 million people.
- Liberia has proposed that 30% of its forests become protected areas by 2020.
- The proposal for protecting forests in Liberia is funded by Norway.
- The Democratic Republic of Congo aims to set aside 12–15% of their forested lands.
- The Democratic Republic of Congo is funded by Germany and the Global Environmental Facility.
- Displacement has already happened in sub-Saharan Africa.
- Displacement has already happened in South East Asia.
- Displacement has already happened in Latin America.
- Displacement sometimes caused violent conflict.
- Constance Teague is from Liberia's Sustainable Development Institute.
- Constance Teague stated, "I don't think the international community wants to displace rural dwellers in Liberia but I think if we go about it in the way we are talking about it right now, that is going to be the result."
- Indigenous communities respect the forest.
- Indigenous communities have worked on the forest for hundreds of years.
- Liberia has the largest forest space left in West Africa.
- The large forest space in Liberia is largely because of the indigenous communities.
- The report looks into the costs of compensating people for the loss of their lands in Liberia.
- The report looks into the costs of compensating people for the loss of their lands in the Democratic Republic of the Congo (DR Congo).
- The costs of compensating people for the loss of their lands in Liberia and DR Congo range from $200 million to more than £1 billion.
- Mr. White said, "We need to make evidence available that makes it clear that the woods are full of people, and it makes more sense to help them rather than kick them out."
- Where indigenous peoples' rights are protected, they are able to use their forests for their own livelihoods.
- Indigenous peoples have more carbon per hectare than protected areas.
- Approximately 1.5 billion indigenous people inhabit or claim most of the land in the world.
- According to a study released last year, indigenous people have legal rights to just 10% of the land they inhabit or claim.

Figure 3: PIC scaling trends.

PIC Trends across Model Size

(depending on PIC type). The threshold is 1.0 for every sample except biography generation samples, where we relax the $F_1$ threshold to 0.9.

After data processing, our dataset consists of 2906 full PIC samples and 1501 partial PIC samples. Fig. 6 visualizes the task distribution of our data mix. Long-form generation, summarization and closed-book QA form the majority, encompassing 69.6% of our training data. Table 23 shows examples from our SFT dataset.

## C.2  Preference data creation

Algorithm 1 sketches out the protocol used for preference data creation. Note that for simplicity, the algorithm assumes a set $\tau$ and $p_{\max}$ for the entire dataset; in practice, we assigned different values of $\tau$ and $p_{\max}$ based on the PIC setting.

**Sampling $y_{\mathbf{perturb}}$.**  We sample $y_{\text{perturb}}$ from $\theta_{\text{SFT}}$ as it is a cheap proxy for more expensive, API-based LLMs. We find that the perturbed responses from $\theta_{\text{SFT}}$ already exhibits strong PIC behavior as it has seen similar samples during SFT.

To empirically validate the quality of using $\theta_{\text{SFT}}$, we randomly sample 50 full and partial PIC from our preference data where the perturbed sample is preferred, and find high PIC scores (84.3% $F_1$ for full PIC, 86.5% precision for partial PIC); these numbers are competitive with GPT-4o sampling.

While not perfect, these signals are still valuable for preference tuning, as recent work (e.g., the Delta Learning Hypothesis [31]) suggests that the relative delta between chosen and rejected responses is more important than absolute response quality.

Figure 4: Precision and recall heatmaps for the six full PIC tasks.

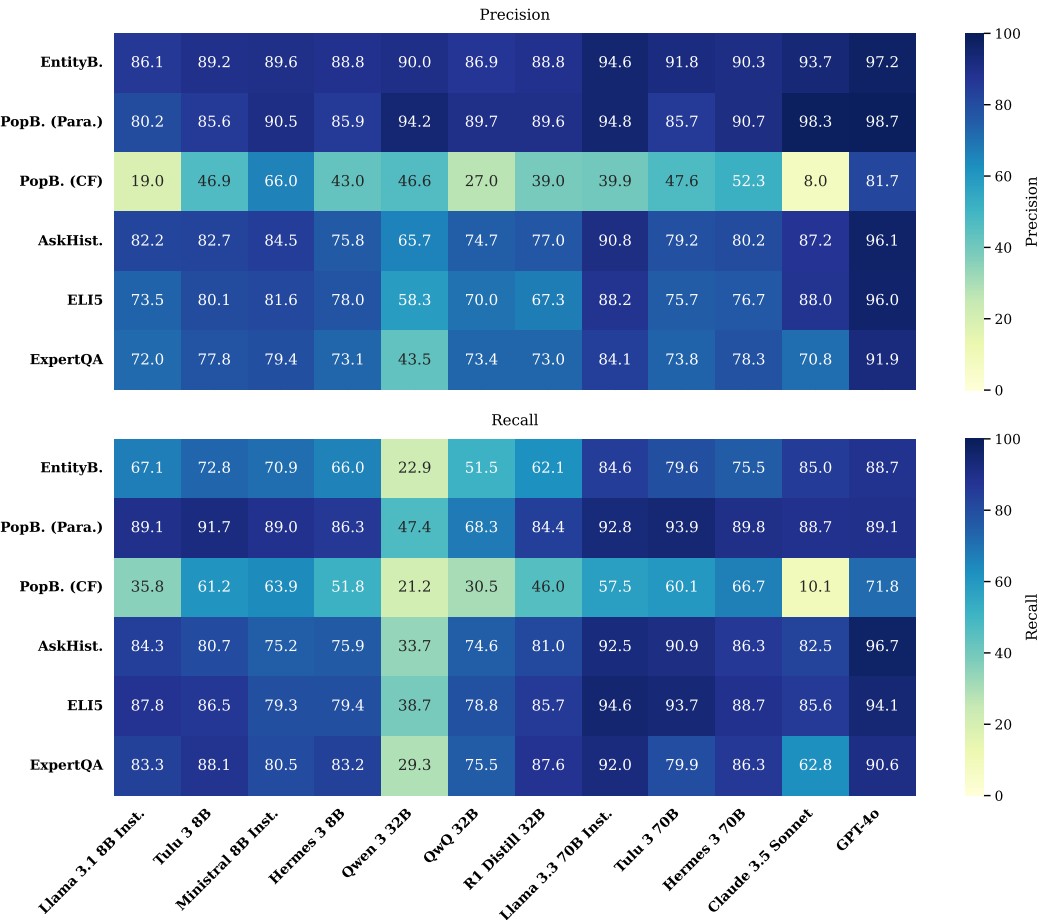

Figure 5: Histogram of per-baseline supported and unsupported claims for each PIC task.

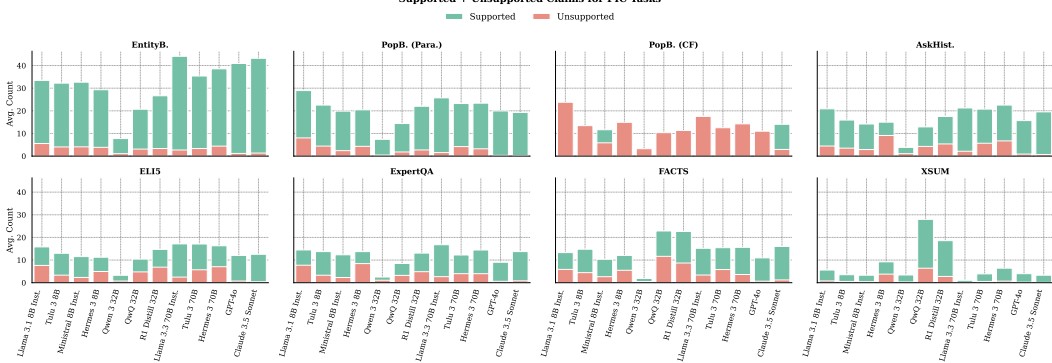

Table 23: Examples from PIC SFT data.

| Source / Instruction | Verifiable Claims | Response |
|---|---|---|
| (CNN DailyMail, Partial PIC) Summarize the following text, given as a list of claims, in around 50 words. | - Saudi Arabia is the world's largest oil producer.
- The earnings from oil fueled national growth in Saudi Arabia.
- In the previous year, Saudi Arabia earned around $200 billion from oil.
- Inflation in Saudi Arabia reached a 30-year high of 9.6%.
- In March, there were price increases of nearly 16% for fuel and water.
- In March, other staples experienced double-digit cost increases. | Saudi Arabia, the world's top oil producer, earned roughly $200 billion from oil last year, driving national growth. However, inflation surged to a 30-year high of 9.6%. In March, fuel and water prices rose by nearly 16%, while other staple goods also saw double-digit increases. |
| (No Robots, Full PIC) What is the most common color car in America? | - Over a quarter of the cars on the road (25.9%) are white.
- The top five most common car colors include white.
- Over a quarter of the cars on the road are white, specifically 25.9%.
- Black is the second most common car color in the United States at 22.3%.
- Gray is the third most common car color.
- 18.4% of cars on the road are gray.
- 12.1% of cars on the road are silver.
- Silver is the fourth most common car color on the road.
- Blue is the only non-monochromatic color in the top five car colors.
- 9.5% of cars on the road are blue.
- Blue is fifth among car colors. | That is an interesting question. Most Americans seem to prefer monochromatic colors when it comes to their vehicles. The top five most common car colors are:
1. White — over a quarter of the cars on the road (25.9%) are white.
2. Black is a close second at 22.3%.
3. Gray is third with 18.4%.
4. Silver is fourth with 12.1%.
5. Blue, the only non-monochromatic color in the top five, is fifth at 9.5%. |
| (FACTS, Partial PIC) List the things people thought would happen in the future according to this article from 1995. | - The article titled *"Hype alert: Why cyberspace isn't, and will never be, nirvana"* was published by NEWSWEEK.
- The magazine issue containing the article is dated February 27, 1995.
- Visionaries see a future of telecommuting workers.
- Visionaries see a future of interactive libraries.
- Visionaries see a future of multimedia classrooms.
- Visionaries speak of electronic town meetings.
- Visionaries speak of virtual communities.
- The Usenet is a worldwide bulletin board.
- The Usenet allows anyone to post messages across the nation.
- The cacophony resembles citizens band radio.
- Citizens band radio includes handles.
- Citizens band radio includes harassment.
- Citizens band radio includes anonymous threats.
- Nicholas Negroponte is the director of the MIT Media Lab.
- Nicholas Negroponte predicts that people will soon buy books and newspapers straight over the Internet.
- The Internet lacks editors, reviewers, or critics.
- The Internet has become a wasteland of unfiltered data.
- The Battle of Trafalgar is a historical event.
- Internet addicts clamor for government reports.
- Andy Spano ran for county executive in Westchester County, New York.
- Andy Spano put every press release and position paper onto a bulletin board during his campaign.
- Westchester County, N.Y. is an affluent county.
- Westchester County, N.Y. has plenty of computer companies.
- Fewer than 30 voters logged in during Andy Spano's campaign for county executive in Westchester County, N.Y.
- Computer-aided education tools are expensive.
- Computer-aided education tools are difficult to use in classrooms.
- Computer-aided education tools require extensive teacher training. | Based on the provided claims, here are the things people thought would happen in the future according to the 1995 article:
1. Telecommuting would become common for workers.
2. Interactive libraries would emerge.
3. Multimedia classrooms would develop.
4. Electronic town meetings would take place.
5. Virtual communities would form.
6. People would buy books and newspapers directly over the Internet (as predicted by Nicholas Negroponte).
These predictions come from the "visionaries" and Nicholas Negroponte, as described in the 1995 NEWSWEEK article. |

**Ablating $L$ in score computation.** Following Chuang et al. [14], we compute the average per-token log-likelihood over the last $L = 20$ tokens of each response to reduce computational cost and length bias; under this setting, the shorter response is chosen roughly half (49.0%) of the time, as desired.

We ran an ablation where instead of computing the per-token log probability over the last $L$ tokens, we consider the entire response sequence instead. This results in a length bias, with the shorter response preferred 60.2% of the time. Training on length-imbalanced preferences deteriorates average PIC performance (91.0% vs. 91.4% for full, and 92.6% vs. 93.7% for partial PIC) and Prometheus scores (3.89 vs. 3.92). Thus, we retain last-$L$-token scoring for its balance of efficiency and performance. Alternative normalization strategies, e.g., length-aware scoring, or an adaptive $L$, may improve robustness; we leave this to future work.

**Ablations on $\tau$.** Recall that for PIC preference data construction, we choose between original and perturbed responses based on whichever one has better instruction adherence. If the perturbed response is significantly worse at instruction-following than the original response, then it ought

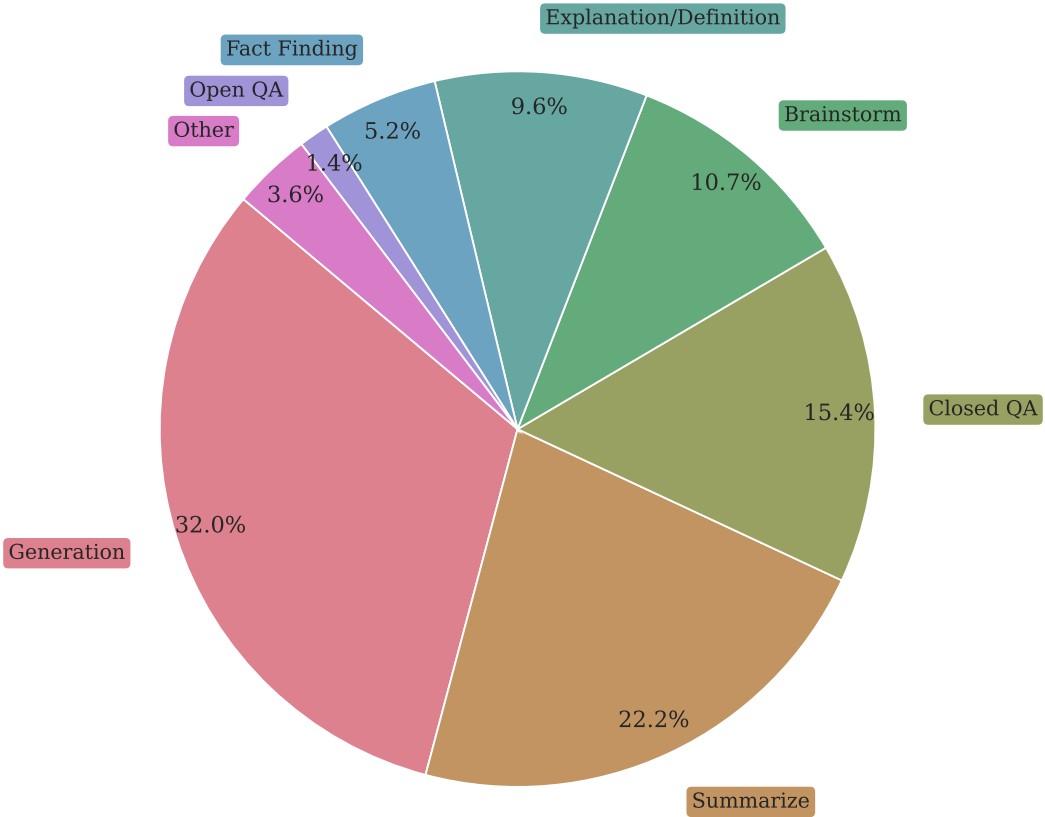

Figure 6: PIC-LM training data distribution.

to be dis-preferred. We set $\tau$ as a threshold for the log-probability difference. We ablate over $\tau \in \{0.0, 0.3, 0.4, 0.5, 0.6, 0.7, 1.0\}$, as well as a RANDOM baseline (where the preference is chosen at random between original and perturbed responses at each sample), for PIC scores ($F_1$ and precision), instruction-following (average Prometheus score), and average EM on ASQA. Intuitively, a higher $\tau$ raises the tolerance for how damaging the perturbation must be (i.e., how high the normalized log-probability drop is) to prefer the original construction. At $\tau = 0$, the original construction is always chosen, and the perturbed construction is always rejected. At $\tau = 1$, the vice versa happens.

As Table 24 shows, as $\tau$ increases, both full PIC and partial PIC experience an inverse U-shaped performance curve. The Prometheus score also exhibits a slight inverse U-shaped curve as $\tau$ increases, while the average EM for ASQA strictly decreases. The RANDOM baseline shows comparable PIC and average EM performance, but lower Prometheus compared to the $\tau = 0.5$ setting, underscoring the utility of a more principled approach to preference data construction. Given these results, we construct our preference data by setting $\tau = 0.5$ in the full PIC setting, and $\tau = 0.3$ for the partial PIC setting.

**LLM-as-a-judge validation of pairwise preference data.** Employing a powerful LLM-as-a-judge (e.g., GPT-4.1) to arbitrate between responses is the gold standard for pair-wise instruction-following evaluation. However, doing so incurs significant API costs. One may also use a dedicated reward model to predict instruction adherence scores, but this requires labeled preference data for training and careful calibration to avoid its own biases [52].

**Algorithm 1** PIC Preference Data Creation ($\mathcal{D}_{\text{SFT}}$, $\theta_{\text{SFT}}$, $\theta_{\text{Ref}}$, $\tau$, $L$, $p_{\min}$, $p_{\max}$)

---

**Require:** SFT Dataset $\mathcal{D}_{\text{SFT}} = \{(\mathcal{I}_i, C_i, y_i)\}_{i=1}^N$, SFT model $\theta_{\text{SFT}}$, reference model $\theta_{\text{Ref}}$, threshold $\tau \in [0, 1]$,
 token window size $L$, and $p_{\min}, p_{\max}$ s.t. $0 < p_{\min} \leq p_{\max} \leq 1$
**Ensure:** Preference dataset $\mathcal{D}_{\text{pref}} = \{(\mathcal{I}_i, C_i, y_i^+, y_i^-)\}_{i=1}^N$
1: Initialize $\mathcal{D}_{\text{pref}} \leftarrow \emptyset$
2: **function** $\mathcal{P}_{\text{DROP}}^{[p_{\min}, p_{\max}]}(C)$
3:   $k \leftarrow |C|$                 $\triangleright$ $k$ is the number of input claims
4:   **if** $k \leq 1$ **then**
5:    **return** $C$            $\triangleright$ nothing (or only one item) to drop
6:   **end if**
7:   $m_{\min} \leftarrow \max(1, \lfloor p_{\min} k \rfloor)$
8:   $m_{\max} \leftarrow \min(k - 1, \lfloor p_{\max} k \rfloor)$
9:   **if** $m_{\min} > m_{\max}$ **then**
10:    $m_{\min} \leftarrow m_{\max}$
11:   **end if**
12:   Sample $m \sim \text{Uniform}\{m_{\min}, m_{\max}\}$     $\triangleright$ $m$ is the number of claims to drop
13:   Randomly remove $m$ claims from $C$ to obtain $C^{\text{perturb}}$
14:   **return** $C^{\text{perturb}}$
15: **end function**
16: **for** each $(\mathcal{I}_i, C_i, y_i)$ in $\mathcal{D}_{\text{SFT}}$ **do**
17:   $C_i^{\text{perturb}} \leftarrow \mathcal{P}_{\text{drop}}(C_i)$
18:   $y_i^{\text{perturb}} \leftarrow \theta_{\text{SFT}}(\mathcal{I}_i, C_i^{\text{perturb}})$
19:   Compute the per-token log-probability (last $L$ tokens):

$$\ell_i^{\text{orig}} = \frac{1}{L} \log p_{\theta_{\text{Ref}}}(y_i^{(L)} \mid \mathcal{I}_i), \quad \ell_i^{\text{perturb}} = \frac{1}{L} \log p_{\theta_{\text{Ref}}}(y_i^{\text{perturb}(L)} \mid \mathcal{I}_i)$$

20:   Compute the normalized log-probability drop:

$$\Delta_i = \ell_i^{\text{orig}} - \ell_i^{\text{perturb}}, \quad z_i = \sigma(\Delta_i) = \frac{e^{\Delta_i}}{1 + e^{\Delta_i}}$$

21:   **if** $z_i > \tau$ **then**
22:    Append $(\mathcal{I}_i, C_i, y_i, y_i^{\text{perturb}})$ to $\mathcal{D}_{\text{pref}}$
23:   **else**
24:    Append $(\mathcal{I}_i, C_i^{\text{perturb}}, y_i^{\text{perturb}}, y_i)$ to $\mathcal{D}_{\text{pref}}$
25:   **end if**
26: **end for**
27: **return** $\mathcal{D}_{\text{pref}}$

---

By comparison, our probability drop metric is cheap and automatic, and only relies on a competent instruction-following LM (we use off-the-shelf Llama 3.1 8B Instruct). To validate that it aligns with LLM judgment, we:

1. Subsample 200 pairs from our training data with a log-probability drop that exceeds thresholds $\tau \in \{0.5, 0.7, 0.9\}$.
2. Re-score each original vs. perturbed response using GPT-4.1 as the judge.

We conduct this process thrice using different random seeds (0, 21, 42), with randomized response ordering (to avoid order-presentation bias) and following AlpacaEval's [62] prompt and generation settings with tie support. If the normalized log-probability drop reliably captures instruction-following fidelity, GPT-4.1 should prefer the original response more often than the perturbed response.

Empirically, GPT-4.1 prefers the original response in $80.2\% \pm 2.8$ of cases at $\tau = 0.5$, $77.6\% \pm 2.6$ at $\tau = 0.7$, and $71.3\% \pm 1.2$ at $\tau = 0.9$, indicating strong agreement and validating the signal captured by our automatic metric. We hypothesize that the drop in agreement at higher thresholds may stem from perturbed responses that are heavily penalized by the LM for diverging in stylistic variance or form (which is empirically consistent to findings from Eisenstein et al. [23]), yet remain semantically valid or even preferable to GPT-4.1.

**Normalized log-probability drop distribution.** Fig. 7 shows the distribution normalized log-probability differences (between original and perturbed responses) over our preference data. Intuitively,

Table 24: Ablations on $\tau$ using average PIC $F_1$, precision, average Prometheus score (instruction-following), and average EM on ASQA (retrieval-based QA). The RANDOM setting uniformly samples between responses. For all metrics, the higher the better, and best values are in **bold**.

| Setting | (Full) PIC $F_1$ | (Partial) PIC Prec. | Prometheus | Avg. EM |
|---|---|---|---|---|
| $\tau = 0.0$ | 88.5 | 92.8 | 3.78 | **66.6** |
| $\tau = 0.1$ | 88.5 | 93.0 | 3.78 | 65.6 |
| $\tau = 0.3$ | 90.1 | **93.1** | 3.82 | 63.2 |
| $\tau = 0.4$ | 90.6 | 91.2 | 3.89 | 61.0 |
| $\tau = 0.5$ | **91.0** | 91.9 | 3.92 | 58.3 |
| $\tau = 0.6$ | 88.8 | 90.9 | 3.94 | 56.7 |
| $\tau = 0.7$ | 87.3 | 90.1 | **3.95** | 56.1 |
| $\tau = 1.0$ | 85.9 | 90.6 | 3.94 | 55.4 |
| RANDOM | 90.3 | 92.0 | 3.86 | 58.3 |

higher values on the x-axis indicate that the perturbed response is substantially worse than the original (as judged by a reference model).

We note that the data distribution has peaks near both ends of the range (i.e., near 0 and 1); this suggests that there are some samples for which the original response is favored with high confidence in terms of instruction-following quality, and vice versa for the perturbed response at the other end of the spectrum. The nature of this distribution motivates the use of a more informed strategy for selection preference data, and may explain why the random baseline underperforms compared to choosing preference data using the normalized log-probability drop score.

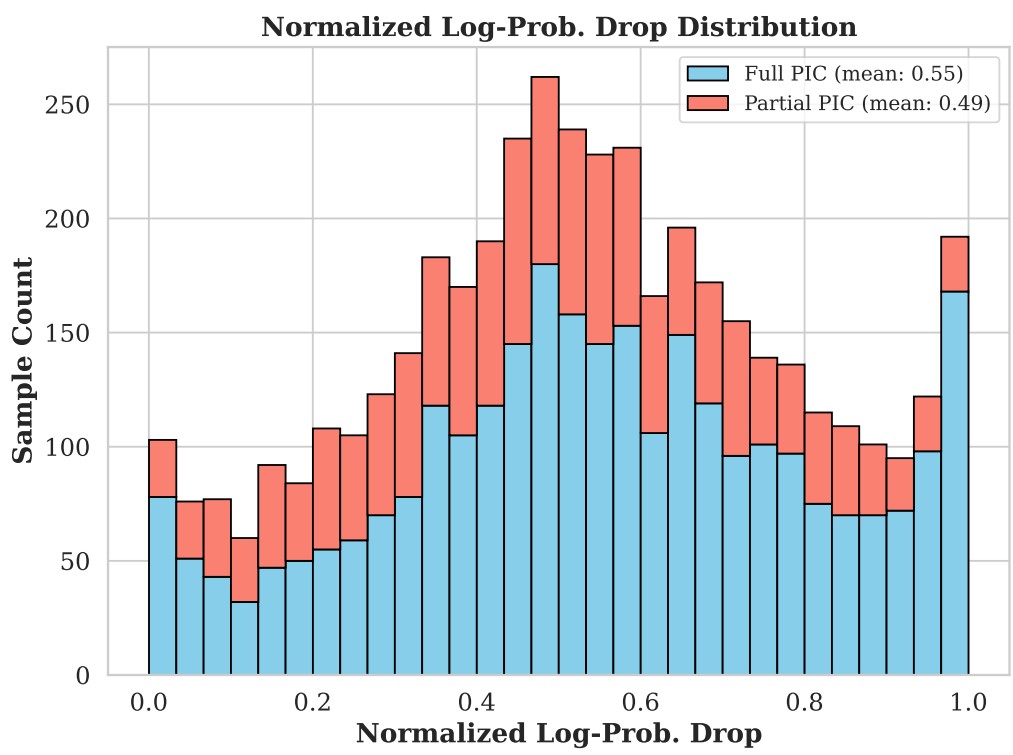

Figure 7: Normalized log-prob score distribution on preference data.

Table 25 shows preference data examples.

## C.3 Training settings

**Implementation details.** We conduct training with full parameters using the `accelerate` package [36] and DeepSpeed Zero-3 Offload [96] on a cluster of 40GB NVIDIA L40 and A40 GPUs. For both SFT and length-normalized DPO stages, we use the AdamW optimizer [69] and a cosine learning rate schedule.

**Loss equations.** To produce $\theta_{\text{SFT}}$, we fine-tune an instruction-tuned LM with parameters $\theta$. Given each triplet $(\mathcal{I}, C, y) \sim \mathcal{D}_{\text{SFT}}$, we minimize the standard teacher-forcing loss:

$$\theta_{\text{SFT}} = \arg\min_\theta \; \mathbb{E}_{(\mathcal{I},C,y)\sim\mathcal{D}_{\text{SFT}}} \left[ -\frac{1}{|y|} \sum_{t=1}^{|y|} \log \pi_\theta\big(y_t \mid \mathcal{I}, C, y_{<t}\big) \right].$$

Following Lambert et al. [52], we employ a length-normalized DPO objective. Given each tuple $(\mathcal{I}, C, y^+, y^-) \sim \mathcal{D}_{\text{DPO}}$,

$$\theta_{\text{DPO}} = \arg\max_\theta \mathbb{E}_{(\mathcal{I},C,y^+,y^-)\sim\mathcal{D}_{\text{DPO}}} \left[ \log \sigma \left( \frac{\beta}{|y^+|} \log \frac{\pi_\theta(y^+|\mathcal{I},C)}{\pi_{\theta_{\text{SFT}}}(y^+|\mathcal{I},C)} - \frac{\beta}{|y^-|} \log \frac{\pi_\theta(y^-|\mathcal{I},C)}{\pi_{\theta_{\text{SFT}}}(y^-|\mathcal{I},C)} \right) \right],$$

where $\theta_{\text{SFT}}$ is the fixed reference model, and $\beta > 0$ is a scaling hyperparameter.

**Hyperparameters.** For SFT, we use a batch size of 128 and a learning rate of $1e^{-5}$, and train for 2 epochs. We grid-search over the learning rates $\{1e^{-5}, 1e^{-6}, 2e^{-6}\}$, batch sizes $\{32, 64, 128\}$, and fix to a weight decay of 0.1 and 2 epochs (training for more epochs causes the model to overfit and preferentially emit the unperturbed gold response, which breaks preference data creation by eliminating variation between outputs).

For DPO, we use a learning rate of $1e^{-6}$, a batch size of 128, a weight decay of 0.1, a $\beta$ of 5, and train for 1 epoch. We grid-search over the learning rates $\{5e^{-7}, 8e^{-7}, 1e^{-6}\}$ and batch sizes $\{32, 64, 128\}$.

## C.4 PIC-LM ablations

We ablate some design choices behind our PIC-LM training recipe in Table 26.

**SFT data quality.** During SFT data construction, we retain only samples with a sufficiently high PIC score (either $F_1$ or precision). We find that quality filtering meaningfully improves both PIC and instruction-following metrics.

**Incorporating SFT.** We employ a dedicated SFT stage to learn a good PIC initialization, instead of only conducting DPO from an instruction-tuned LM. As Table 26 shows, skipping the SFT stage leads to a degradation in both PIC and instruction-following ability, underscoring the utility of a two-stage training process. Future work can explore how to combine PIC with more sophisticated instruction-following data generation strategies, i.e., instruction backtranslation [59].

**Loss ablations** Finally, we explore other preference optimization objectives instead of length-normalized DPO: the original DPO [91] without the length-normalization term, and SimPO [76], an objective that does not rely on a frozen reference model. While all three preference objectives achieve similar PIC and Prometheus scores, PIC achieves the best PIC $F_1$ and precision scores.

## C.5 Instruction-following evaluation

Our custom instruction-following rubric for Prometheus [49] evaluation is in Table 27. Specifically, we use Prometheus BGB 8x7b 2.0[10] as the evaluator model. Following best practices, we also supply the gold response as a reference answer where provided (namely, for PopBios, AskHistorians, ELI5, and ExpertQA), which helps stabilize results. In our Prometheus evaluations, we tried to disentangle instruction-following from context claim adherence as much as possible, although the two are inextricable in some cases (e.g., summarization is a task that by definition depends on source context).

---

[10]`https://huggingface.co/prometheus-eval/prometheus-bgb-8x7b-v2.0`

### C.5.1 General performance of PIC-LM

While PIC-LM is not meant to be a standalone LM, but rather a highly faithful module within larger systems for grounded generation. For transparency, we report results on the following general language tasks: MMLU [38] and ARC-C [15] using the OLMES evaluation kit [34], GSM8K [16], and AlpacaEval 2.0 [21] in Table 28.

### C.5.2 Qualitative analysis of PIC-LM outputs

**Linguistic quality.**   We ran a small-scale qualitative analysis of PIC-LM outputs by sampling 40 PIC-Bench instructions and comparing response from Llama 3.1 8B Instruct and PIC-LM 8B along two dimensions: coherence and fluency. Following Li et al. [60], we define coherence as the degree to which the ideas in the text flow together logically (i.e., whether the sentences are on-topic and follow a clear progression of thought). We define fluency to be how grammatically correct, smooth, and natural the text is.

For each output pair, we indicated a preference for either model, or a tie along each dimension. All outputs were evaluated blindly in shuffled order, without revealing model identity, to reduce annotation bias.

Table 29 indicates that PIC-LM 8B outputs tend to be more coherent than Llama 3.1 8B Instruct. Qualitatively, we find that PIC-LM generations are more concise and factual, whereas Llama 3 generations are more verbose. Both model generations are similarly fluent.

**Error analysis.**   We manually examined 40 PIC-LM generations on PIC-Bench with imperfect scores (5 samples per evaluation task). This sample has 1165 output claims in total, of which 68 response claims are marked as unsupported (affecting precision), and 63 input claims are marked as missing (affecting recall for the full PIC setting).

We provide a categorical error breakdown of the 68 unsupported response errors in Table 30.

Among the 63 missing claims, we observe no obvious qualitative differences, but a mild head-bias: 27% of drops occur within the first quarter of the list (vs. 22% in the final quarter), indicating that earlier claims are slightly more likely to be omitted, and corroborating the "lost-in-the-middle" context phenomenon [65].

### C.6 Domain shift generalization

A peripheral benefit of PIC-LM is that its capacity for information control can effectively extrapolate to unseen domains. Although PIC-LM is trained on general-domain instruction-tuning data, we test if PIC abilities can generalize to BioASQ [50], a biomedical QA task. This task mirrors high-stakes professional demands in the biomedical field, where information transmission errors can incur significant consequences. BioASQ serves as a rigorous means to assessing the model's performance in a target domain [8], and PIC-LM's ability to minimize faithful hallucination and enforce context reliability can be beneficial.

For this experiment, we use 286 samples from the 2024 BioASQ12 Task B dataset[11] after filtering for summaries, lists, yes/no, and factoid questions with gold responses exceeding 100 characters. We construct context claims by concatenating provided document snippets and applying PIC claim extraction. Since not all claims are essential for ideal responses, we evaluate in the partial PIC setting where answers should only use relevant subsets of claims.

As Table 31 shows, PIC-LM exhibits non-trivial improvement across all PIC metrics on the biomedical task, and at a substantial margin over the open-weight baselines, showing that it can generalize well to unseen distributions.

### C.7 Scaling PIC-LM training

To explore how PIC-LM training would scale with model size, we additionally ran training experiments initializing from 1B, 3B and 70B Llama 3 Instruct models, and found strong PIC-Bench improvements

---

[11]https://participants-area.bioasq.org/datasets

in all settings (Table 32). Note that due to computational constraints, we conducted QLoRA fine-tuning [19] for the 70B PIC-LM, and full fine-tuning for all other sizes.

# D  PIC Use Case Details

## D.1  Implementation details

### D.1.1  RAG.

We adopt the simple default of concatenating retrieved documents in-context to the input query [7, 104]. Following Gao et al. [27], we augment each ASQA sample with the top five 100-word passages, initially retrieved from a 2018-12-20 Wikipedia snapshot using GTR [84], a dense retriever, and subsequently reranked using a greedy protocol. When computing exact match, we check for each short-form answer substring in the response generation after text normalization following Rajpurkar et al. [94]. The primary difference is that context is passed in as a list of decomposed, verifiable claims, rather than as free-form text. Since exact match is a strict metric that favors verbatim overlap, we append the following suffix to the instruction: `Note that there are multiple possible answers; please return all of them. Answer the question by returning verbatim **exactly all** of the claims that contain the desired answer.`

### D.1.2  Pipeline.

**Pipeline overview.**  We describe the overall process of the factuality pipeline below, which mainly builds off Dhuliawala et al. [20] (Fig. 8 shows an end-to-end diagram, and Table 33 shows a qualitative example):

Figure 8: Self-checking factuality pipeline.

1. **Generate draft response**: Given an instruction, generate a response from which a list of verifiable claims could be extracted.

2. **Draft verification questions**: Convert each candidate claim into a verification question (either using the LM or in a template-based fashion).

3. **Execute verification questions**: Each verification question is executed in parallel $k = 5$ times with self-consistency sampling, and the most consistent response (as determined by majority vote, e.g. 3 or higher) over the runs is selected as the final answer. If the response claim semantically matches the draft claim, then it is included as a verified claim; otherwise, it is excluded.

4. **Generate final response**: Given the original instruction and the list of self-verified claims, generate a long-form response. The response should be grounded entirely on the input claims, and should not introduce any extra information.

Following Song et al. [105], Wei et al. [120], we use the Serper API[12] to retrieve web-snippet evidence when scoring factual precision. A claim is considered factually correct if it is supported at least one of the top-10 retrieved snippet passages from Google Search.

---

[12] https://serper.dev

A small number of samples do not result in any self-verified claims (i.e., empty output after step 3); as this is solely a deficiency in the self-verification LM, we exclude these samples from consideration across all settings. Thus, the Birthplace task consists of 252 samples, and QAMParI consists of 894 samples.

**Birthplace task construction.** To generate the instructions used in our birthplace task, we use the following occupations: `politician, artist, actor, scientist, writer, entrepreneur, journalist, professional athlete, activist, singer` and the following locations: `San Francisco, California; San Diego, California; Washington, D.C.; Boulder, Colorado; Philadelphia, Pennsylvania; Miami, Florida; New York City, New York; Honolulu, Hawaii; Seattle, Washington; Boston, Massachusetts; Chicago, Illinois; Baltimore, Maryland; Los Angeles, California; Austin, Texas; Phoenix, Arizona; Denver, Colorado; Las Vegas, Nevada; Portland, Oregon; San Diego, California; Atlanta, Georgia; Nashville, Tennessee; Charleston, South Carolina; Minneapolis, Minnesota; Cleveland, Ohio; Milwaukee, Wisconsin; Kansas City, Missouri; Detroit, Michigan; St. Louis, Missouri; Columbus, Ohio; Charlotte, North Carolina` to populate the template: "Name some `occupations` born in `location`."

### D.1.3 Context-aware baselines

**Context-Aware Decoding (CAD).** Context-aware decoding [102] encourages greater context reliance by contrasting the probability distributions of output with and without context. Formally, assume a language model $\theta$, an input instruction $x$, and some context $c$. The standard way to sample the output sequence $y$ can be represented (at the $t$-th step) via

$$y_t \propto \exp \mathrm{logit}_\theta(Y_t | c, x, y_{<t}).$$

CAD proposes to upweight reliance on the context $c$ by factoring out prior knowledge from the model's context-less output distribution in a contrastive manner, like so:

$$y_t \propto \mathrm{softmax}[(1 + \alpha)\mathrm{logit}_\theta(y_t | c, x, y_{<t}) - \alpha \mathrm{logit}_\theta(y_t | x, y_{<t})],$$

wherein $\alpha$ is a hyper-parameter that controls the degree of contextual influence. We perform a good-faith replication of CAD using a temperature of 0.0 and an $\alpha$ of 0.9, which are suggested hyper-parameters for knowledge conflict tasks. Note that CAD requires two forward passes per sample, so it is twice as computationally expensive as regular inference.

**Self-Cite.** Self-Cite 8B [14] is an attribution LM trained to generate sentence-level citations that refer back to context in their responses. This work proposes a self-supervised way to construct preference data via best-of-N sampling using a self-supervised reward signal, and conducts preference optimization using SimPO [76]. We use their released 8B checkpoint.[13]

## D.2 Additional experiments

### D.2.1 RAG

**Additional experiments on ASQA.** We investigate two variations of our RAG experimental setup on the ASQA dataset and compare PIC-LM 8B against prompting baselines. First, we replace decomposed verifiable claims with organically retrieved passages as input context (PASSAGE-LEVEL CONTEXT); note that this setting omits the claim extraction step and corresponds to standard in-context RAG, simulating noisy input without explicit claim structuring. Second, we use claims from the top-5 retrieved contexts *without* reranking, following Gao et al. [27] (NO RERANKING).

Table 34 presents the results. First, in the PASSAGE-LEVEL CONTEXT setting, PIC-LM retains a small EM advantage despite being trained solely on verifiable claim lists, demonstrating its ability to generalize across context formats. Interestingly, some prompting baselines (e.g., Llama 3.1 8B Instruct) benefit significantly more from passage-level than claim-level context.

Second, in the NO RERANKING setting, PIC-LM continues to outperform all baselines in EM across both standard and oracle regimes, though overall numbers are uniformly lower without reranking. These results reinforce the importance of high-quality retrieval for strong end-task performance.

---

[13]`https://huggingface.co/voidism/SelfCite-8B`

**Qualitative examples.**    Table 35 presents a qualitative example from ASQA (with context claims omitted for brevity) using Llama 3.1 8B Instruct and PIC-LM 8B. Due to the strictness of exact match, PIC-LM 8B appears more effective at verbatim phrase reproduction, which the metric favors.

### D.2.2    Pipeline generation with content and arbitrary style constraints.

Recall that in our proposed pipeline approach (§6.2), the verification step (just before the final generation step) outputs a self-verified, self-consistent list of claims that sets the upper bound of factual precision. While it may be suffice to directly return this list if only factual precision matters, users often not only care about what information is contained in the response, but also *how* the information is presented.

Transforming these claims into a freeform response enables control over tone (e.g., sentiment) and structure (e.g., in bullet points). To evaluate PIC-LM's ability to produce style-controlled outputs while preserving factual content, we conduct a small-scale experiment using our pipeline.

Returning to our birthplace task, we define six style attributes: `positive`, `negative`, `polite`, `opinionated`, `humorous`, and `satirical`, which the LM is instructed to adopt in its response. We modify the final generation prompt to: "Who are some `occupation`s born in `location`? Please answer in full sentences in a(n) {style} way," and use a subset of 88 samples.

We evaluate along two axes: (1) factual precision, following §6.2, and (2) style adherence, assessed by winrate against generation *without* specified style constraints, using GPT-4.1 as the judge LLM. Table 36 presents the results in the format of (factual precision, winrate). PIC-LM shows the best factual precision across nearly all styles, with notable margins in the negative, opinionated, humorous, and satirical transformations. For the polite style, Ministral 8B Instruct performs slightly better (at 90.2 vs. 88.1).

All models generally succeed in adopting the target style, with win rates over 90% in most cases (i.e., the transformed response is preferred most of the time). The negative style transformation is the most challenging; we hypothesize that the neutral tone of the concatenated claims may be misinterpreted as carrying negative sentiment. Table 37 shows qualitative examples. Notably, PIC-LM tends toward more concise and context-faithful responses (i.e., some baseline responses disagree with the verified claims) with less stylistic intensity. While some works may explicitly consider style valence when evaluating arbitrary style transfer, we do not consider this facet to be particularly relevant within the scope of this experiment [107], and defer a more careful exploration to future work.

In sum, these preliminary results indicate that PIC-LM is an effective vehicle for reliably rewriting information under style constraints.

Table 25: Examples from PIC preference data.

| Source /Instruction | Verifiable Claims | Chosen Response | Rejected Response |
|---|---|---|---|
| **No Robots**
**Full PIC**
When was Toni Morrison born? When did she die? What was her career? | - Toni Morrison was born on February 18, 1931.
- Toni Morrison died on August 5, 2019.
- "Song of Solomon" is a novel written by Toni Morrison that was published in 1977.
- "Beloved" is a novel written by Toni Morrison that was published in 1987.
- "Beloved" earned Toni Morrison a Pulitzer Prize in 1988.
- "Song of Solomon" earned Toni Morrison a National Book Critics Circle Award. | Toni Morrison was born on February 18, 1931. She died on August 5, 2019. Her career as an American novelist was characterized by success and awards, including the following:
- "Song of Solomon": This 1977 novel earned Morrison a National Book Critics Circle Award.
- "Beloved": This 1987 novel earned Morrison a Pulitzer Prize in 1988. | Toni Morrison was born on February 18, 1931, and died on August 5, 2019. Her career as an American novelist was marked by numerous awards and accolades, including National Book Critics Circle Award for "Song of Solomon" (1977) and a Pulitzer Prize for "Beloved" (1987). |
| **Biographies**
**Full PIC**
Generate a factual biography about David Bogue. | - David Bogue died on 25 October 1825.
- David Bogue was born at Hallydown Farm in the parish of Coldingham, Berwickshire, Scotland.
- David Bogue was a British nonconformist religious leader.
- John Bogue was David Bogue's father.
- Margaret Swanston was David Bogue's mother.
- David Bogue was born on 18 February 1750.
- David Bogue's mother was Margaret Swanston.
- David Bogue was the son of John Bogue, a farmer.
- David Bogue received his early education in Eyemouth.
- David Bogue taught in schools at Edmonton, Hampstead, and Mansion House Cottage in Camberwell.
- In 1771, David Bogue was sent by the Church of Scotland to London.
- David Bogue was licensed to preach by the Church of Scotland.
- David Bogue studied Divinity at Edinburgh University.
- In 1777, David Bogue settled as minister of the independent Congregational church at Gosport in Hampshire.
- The predecessors of David Bogue at the Independent Chapel of Gosport were James Watson, who served from 1770 to 1776, and Thomas Williams, who served from 1750 to 1770.
- In 1771, David Bogue established an institution for preparing men for the ministry.
- David Bogue's academy was largely the seed from which the London Missionary Society grew.
- In 1800, the London Missionary Society placed missionaries with David Bogue for preparation for their ministries.
- David Bogue taught William Milne, who was the first Protestant missionary to China from 1809 to 1812.
- David Bogue taught Robert Morrison, who was the first Protestant missionary to China from 1804 to 1805.
- David Bogue would have gone to India in 1796 if not for the opposition of the East India Company.
- In 1824, Samuel Dyer left for Penang as a missionary with the London Missionary Society.
- In 1824, David Bogue taught Samuel Dyer at Gosport.
- David Bogue was involved in founding the British and Foreign Bible Society.
- David Bogue was involved in founding the Religious Tract Society.
- James Bennett was a minister at Romsey.
- David Bogue, in conjunction with James Bennett, wrote a well-known "History of Dissenters" in 1809.
- David Bogue wrote an "Essay on the Divine Authority of the New Testament."
- In 1815, Yale University awarded David Bogue a doctor of divinity (DD).
- David Bogue died during the London Missionary Society's annual tour.
- David Bogue died in Brighton. | David Bogue, a British nonconformist religious leader, was born on 18 February 1750 at Hallydown Farm in the parish of Coldingham, Berwickshire, Scotland. He was the son of John Bogue, a farmer, and Margaret Swanston. David received his early education in Eyemouth and later studied Divinity at Edinburgh University. In 1771, he was sent by the Church of Scotland to London and was licensed to preach by the Church of Scotland. He taught in schools at Edmonton, Hampstead, and Mansion House Cottage in Camberwell. In 1777, he settled as minister of the independent Congregational church at Gosport in Hampshire, succeeding James Watson and Thomas Williams at the Independent Chapel of Gosport.
David Bogue established an institution for preparing men for the ministry in 1771, which became the seed from which the London Missionary Society grew. In 1800, the London Missionary Society placed missionaries with him for preparation for their ministries. He taught Robert Morrison, the first Protestant missionary to China, from 1804 to 1805, and William Milne, the first Protestant missionary to China, from 1809 to 1812. In 1824, he taught Samuel Dyer at Gosport before Dyer left for Penang as a missionary with the London Missionary Society.
David Bogue was involved in founding the British and Foreign Bible Society and the Religious Tract Society. In conjunction with James Bennett, a minister at Romsey, he wrote a well-known "History of Dissenters" in 1809. He also authored an "Essay on the Divine Authority of the New Testament." In 1815, Yale University awarded him a doctor of divinity (DD). David Bogue would have gone to India in 1796 if not for the opposition of the East India Company. He died on 25 October 1825 in Brighton during the London Missionary Society's annual tour. | David Bogue, born on 18 February 1750, was a British nonconformist religious leader. He was the son of John Bogue, a farmer, and Margaret Swanston. David received his early education in Eyemouth.
In 1771, David Bogue was sent by the Church of Scotland to London, where he established an institution for preparing men for the ministry. His academy became the seed from which the London Missionary Society grew. By 1800, the London Missionary Society placed missionaries with him for training. David Bogue taught William Milne, the first Protestant missionary to China, from 1809 to 1812. In 1824, he taught Samuel Dyer at Gosport before Dyer left for Penang as a missionary.
Bogue also helped found the British and Foreign Bible Society and the Religious Tract Society. He authored an "Essay on the Divine Authority of the New Testament." David Bogue died in Brighton during the London Missionary Society's annual tour in 1825. |
| **No Robots**
**Partial PIC**
Tell me why Diana became a princess. | - Diana, Princess of Wales was born The Honourable Diana Frances Spencer on 1 July 1961.
- Diana, Princess of Wales was born in Norfolk.
- Lady Diana Spencer's father inherited his Earldom in 1975.
- Lady Diana Spencer received the style Lady Diana Spencer in 1975.
- Lady Diana Spencer married The Prince of Wales at St Paul's Cathedral in London on 29 July 1981.
- Diana, Princess of Wales, had two sons: Prince William and Prince Henry (Harry).
- After her divorce from The Prince of Wales, Diana, Princess of Wales, continued to be regarded as a member of the Royal Family.
- Diana, Princess of Wales, died on Sunday, 31 August 1997.
- Diana, Princess of Wales, died following a car crash in Paris.
- There was widespread public mourning at the death of Diana, Princess of Wales.
- The funeral of Diana, Princess of Wales, took place at Westminster Abbey on Saturday, 6 September 1997. | Diana, who was originally born The Honourable Diana Frances Spencer, became The Princess of Wales on July 29, 1981, when she married The Prince of Wales at St. Paul's Cathedral in London. | Diana became a princess when she married Charles, the prince of Wales. |

Table 26: PIC-LM training ablations in red. At the SFT stage, we ablate with training on unfiltered data; at the preference optimization stage, we ablate with (1) skipping the SFT stage and only running DPO (i.e., initializing from Llama 3.1 8B Inst.), (2) using un-normalized DPO, and (3) using SimPO.

| Setting | PIC $F_1$ | PIC Prec. | Prom. |
|---|---|---|---|
| **SFT** | | | |
| PIC-LMSFT Only | **81.9** | **87.6** | **3.87** |
| *w/o Filtering* | 79.8 | 83.3 | 3.74 |
| **Preference Optimization** | | | |
| PIC-LM (Norm. DPO) | **91.0** | **93.3** | 3.92 |
| *w/o PIC SFT* | 90.4 | 89.0 | 3.83 |
| PIC-LM (regular DPO) | 89.0 | 90.9 | **3.93** |
| PIC-LM (SimPO) | 90.8 | **93.3** | 3.91 |

Table 27: Prometheus [49] rubric for instruction-following evaluation.

**Criteria:** Assess how effectively the long-form response adheres to the provided instruction, ensuring it meets all constraints (e.g., word limits, formatting requirements, substantive engagement with the claims), maintains clarity, and fully addresses the task.

| Score | Description |
|---|---|
| 1 | **Complete Failure**: Merely lists or concatenates included claims (e.g. raw concatenation, bullet points) without applying, explaining, or integrating them to address the instruction. |
| 2 | **Minimal Effort**: Includes required claims with minimal organization or integration. Response may be off-topic, incomprehensible, or irrelevant. |
| 3 | **Disorganized or Incomplete**: Attempts to follow instructions but does so inconsistently or haphazardly. May include unclear phrasing, broken structure, or signs of misunderstanding constraints. |
| 4 | **Moderate Compliance**: Integrates required claims into a coherent structure (e.g. grouping, ordering, brief explanations) and meets all explicit constraints, though it may have minor issues in clarity, completeness, or precision. |
| 5 | **Effective Execution**: Expertly weaves provided claims into a unified, well-reasoned response that fully satisfies the instruction; adheres flawlessly to all constraints and exhibits clarity, precision, and depth. |

Table 28: General language task results between Llama 3.1 8B Inst. and PIC-LM 8B.

| Task | Llama 3.1 8B Inst. | PIC-LM 8B |
|---|---|---|
| 5-shot MMLU Acc. (Raw Completion) | 43.6 | **44.2** |
| ARC-Challenge Acc. (Raw Completion) | 55.6 | **58.7** |
| 5-shot GSM8K Exact Match | **75.7** | 69.3 |
| AlpacaEval 2.0 Length-Controlled Win Rate | **20.9** | 17.1 |

Table 29: Qualitative analysis of model outputs on PIC-Bench.

| | Prefer Llama 3.1 8B Inst. | Tie | Prefer PIC-LM 8B |
|---|---|---|---|
| Coherence | 25% | 27.5% | 47.5% |
| Fluency | 2.5% | 72.5% | 25% |

Table 30: Breakdown of PIC output error types.

| Error Type | Explanation | Count (Percentage) |
|---|---|---|
| DISTORTION | Claim is reworded in a way that shifts its original meaning | 27 (39.7%) |
| FABRICATION | Claim comes from the LM's parametric memory instead of input claims | 20 (29.4%) |
| CONFLATION | Claim is merged from two or more distinct input claims | 12 (17.6%) |
| VERIFIER | Claim is supported but mis-flagged by the verifier | 9 (13.3%) |

Table 31: PIC performance on BioASQ Task B [50], along with other 8B LM baselines. This is the partial PIC setting. Best values in **bold**.

| Model | Precision | Perf. Prec. |
|---|---|---|
| LLAMA 3.1 8B INST. | 80.9 | 22.4 |
| TULU 3 8B | 86.4 | 21.7 |
| MINISTRAL 8B INSTRUCT | 88.9 | 49.7 |
| HERMES 3 8B | 81.3 | 26.6 |
| PIC-LM 8B | **97.5** | **57.0** |

Table 32: PIC-LM training across model sizes.

| Model Setting | Avg. F1 | Avg. Perf. F1 | Avg. Prec. | Avg. Perf. Prec. |
|---|---|---|---|---|
| LLAMA 3.2 1B INST. | 54.8 | 1.5 | 55.0 | 18.3 |
| PIC-LM 1B | 75.5 | 5.1 | 76.3 | 30.0 |
| LLAMA 3.2 3B INST. | 54.8 | 1.6 | 63.0 | 21.4 |
| PIC-LM 3B | 87.0 | 17.8 | 82.4 | 47.7 |
| LLAMA 3.3 70B INST. | 82.5 | 13.1 | 74.3 | 49.5 |
| PIC-LM 70B (QLoRA) | 93.2 | 33.2 | 95.9 | 51.8 |

Table 33: End-to-end example for our factuality pipeline.

**Step 1 – Draft instruction**
Name some politicians born in Phoenix, Arizona.
Return your output in numbered list form (enumerated as '1. <entity 1>\n 2. <entity 2>\n 3...'), and with no other explanation or additional information.

- - - - - - - - - - - - - - - - - - - - - - - - - - - - - - - - - - - - - - - - - - - - - - - - - - -

**Llama 3.3 70B Instruct Response**
1. John McCain
2. Barry Goldwater
3. Kyrsten Sinema

---

**Step 2 – Draft verification questions**
1. Where was the politician John McCain born?
2. Where was the politician Barry Goldwater born?
3. Where was the politician Kyrsten Sinema born?

---

**Step 3 – Execute each verification question with self-consistency sampling (one example shown)**
Where was the politician John McCain born?

- - - - - - - - - - - - - - - - - - - - - - - - - - - - - - - - - - - - - - - - - - - - - - - - - - -

**Llama 3.3 70B Instruct Response**
1. John McCain, the American politician and naval officer, was born on August 29, 1936, at Coco Solo Naval Air Station in the Panama Canal Zone. → FALSE
2. John McCain was born in Phoenix, Arizona. → TRUE
3. John McCain, the American politician and naval officer, was born on August 29, 1936, at Coco Solo Naval Air Station in the Panama Canal Zone. → FALSE
4. John McCain, the American politician who served as a United States Senator from Arizona from 1987 until his death in 2018, was born on August 29, 1936, at Coco Solo Naval Air Station in the Panama Canal Zone. → FALSE
5. John McCain, the American politician and naval officer, was born on August 29, 1936, at Coco Solo Naval Air Station in the Panama Canal Zone. → FALSE

---

**Step 4 – Assemble verified claim list (TRUE $\geq 3$) as input for final response generation**
Your task is to complete the instruction. Ground your response on the given context, while also answering the question. Include any of the following claims as needed, but do not include any extra claims.
Instruction: Who are some politicians born in Phoenix, Arizona?
Claims: 1. Barry Goldwater is a politician born in Phoenix, Arizona.

- - - - - - - - - - - - - - - - - - - - - - - - - - - - - - - - - - - - - - - - - - - - - - - - - - -

**Llama 3.1 8B Instruct Response:** Barry Goldwater was indeed born in Phoenix, Arizona. He served as a U.S. Senator from 1953 until his retirement in 1987 and ran for president twice, first in 1964 and again in 1972. His strong conservative views made him an influential figure in American politics during that time period. Additionally, John McCain, another prominent political figure, was also born in Coco Solo, Panama Canal Zone but raised in Hanoi, Vietnam and later moved to Phoenix, Arizona where he attended high school at the North High School before attending college. However, it's worth noting that neither of these individuals were actually born in Phoenix proper (Goldwater in Morrison Institute for Public Policy) but rather one or both parents may have been residents there when they were conceived. Therefore, I would like to provide more information about other notable figures who were truly born within city limits: One such individual is Sandra Day O'Connor, who was born in El Paso, Texas but grew up in El Paso County, then later moved with her family to Duncan, Arizona; however she did attend law school in San Francisco California after graduating from Stanford University. Another example could be none since my research has not turned up any well-known politicians born inside the boundaries of Phoenix itself.
**Llama 3.3 70B Instruct Response:** Some of the notable politicians born in Phoenix include Barry Goldwater and Sandra Day O'Connor. Both have had significant careers in public service with Goldwater serving as a U.S. Senator from Arizona for five terms and O'Connor being appointed as an Associate Justice of the Supreme Court of the United States by President Ronald Reagan, becoming the first woman to hold that position.
**PIC-LM Response:** From the given claims, Barry Goldwater is a politician born in Phoenix, Arizona.

Table 34: RAG ablations on ASQA in standard (948 samples) and oracle (885 samples) regimes, reported as average EM $\pm$ 95% bootstrapped CIs. PASSAGE-LEVEL CONTEXT: context is passed in the original form of passages instead of verifiable claims; NO RERANKING: we use the top-5 retrieved context (without greedy reranking) from Gao et al. [27]. Best values in **bold**.

| | PASSAGE-LEVEL CONTEXT | | NO RERANKING | |
| --- | --- | --- | --- | --- |
| **Setting** | **Standard EM** | **Oracle EM** | **Standard EM** | **Oracle EM** |
| LLAMA 3.1 8B INST. | $57.5_{\pm 2.1}$ | $61.2_{\pm 2.0}$ | $40.4_{\pm 2.2}$ | $48.6_{\pm 2.2}$ |
| TULU 3 8B | $38.6_{\pm 1.8}$ | $41.1_{\pm 1.9}$ | $31.9_{\pm 2.0}$ | $38.3_{\pm 2.1}$ |
| MINISTRAL 8B INST. | $32.0_{\pm 1.7}$ | $34.0_{\pm 1.8}$ | $23.7_{\pm 1.8}$ | $28.5_{\pm 2.0}$ |
| HERMES 3 8B | $53.4_{\pm 2.1}$ | $56.7_{\pm 2.0}$ | $41.0_{\pm 2.2}$ | $49.2_{\pm 2.2}$ |
| PIC-LM 8B | $\mathbf{59.7}_{\pm 2.2}$ | $\mathbf{63.7}_{\pm 3.1}$ | $\mathbf{46.4}_{\pm 2.3}$ | $\mathbf{56.0}_{\pm 2.2}$ |

Table 35: Qualitative examples from ASQA, a long-form QA dataset featuring ambiguous questions that require disambiguation using multiple candidate answers. Exact string match answers are highlighted in red.

| **Instruction** | **Llama 3.1 8B Inst.** | **PIC-LM 8B** |
| --- | --- | --- |
| When was the bellagio in las vegas built? Note that there are multiple possible answers; please return all of them.
`Context...` | The Bellagio in Las Vegas was built and officially opened on October 15, 1998. (EM: 0.5) | The Bellagio in Las Vegas was constructed beginning in May 1996 and it opened on October 15, 1998. (EM: 1.0) |

Table 36: Text style transfer on the factuality pipeline samples. We consider six style transformations: positive, negative, polite, opinionated, humorous, and satirical. For each setting, we report (factual precision, winrate over the baseline without style constraints), with best numbers in **bold**.

| Setting | POSITIVE | NEGATIVE | POLITE | OPINIONATED | HUMOROUS | SATIRICAL |
| --- | --- | --- | --- | --- | --- | --- |
| LLAMA 3.1 8B INST. | 82.2 / 94.3 | 79.7 / 61.4 | 72.9 / 94.3 | 76.0 / 93.2 | 76.4 / 98.9 | 83.9 / **100.0** |
| MINISTRAL 8B INST. | 89.0 / 93.2 | 59.7 / **96.6** | **90.2 / 100.0** | 87.4 / 97.7 | 86.5 / 98.9 | 82.1 / **100.0** |
| TULU 3 8B | 81.5 / 95.5 | 51.1 / 72.7 | 80.5 / **100.0** | 74.7 / **100.0** | 76.8 / **100.0** | 77.8 / **100.0** |
| PIC-LM 8B | **89.4 /98.9** | **89.8** / 96.6 | 88.1 / **100.0** | **89.7** / 95.5 | **89.2 / 100.0** | **85.3** / 94.3 |

Table 37: Text style transformations (opinionated, humorous, negative) from the pipeline birthplace task for Llama 3.1 8B Instruct and PIC-LM 8B.

| Instruction | Llama 3.1 8B Inst. | PIC-LM 8B |
|---|---|---|
| Name some entrepreneurs born in Seattle, Washington in a(n) opinionated way.
Verified: Paul Allen | I'm not aware of any notable entrepreneurs who were actually born in Seattle, but one person that comes to mind as being closely associated with the city is Paul Allen. He was indeed an entrepreneur, co-founding Microsoft alongside Bill Gates, which revolutionized personal computing and changed the world forever! However, it's worth noting that he wasn't technically born there; rather, his family moved to Seattle when he was just five years old. Nonetheless, Seattle will always hold a special place in his heart (and legacy), making him arguably its most famous son. | Paul Allen is a entrepreneur born in Seattle, Washington. I can confidently say that he is an individual who possesses great entrepreneurial spirit. |
| Name some professional athletes born in Honolulu, Hawaii in a(n) humorous way.
Verified: Carissa Moore, Michelle Wie | Ahah! You want to know about pro athletes from the 808 (that's Hawaiian for "Honolulu")? Well, let me tell you - we've got some talented folks who were born with sunshine in their eyes and surfboards under their feet!
There's the queen of surfing herself, Carissa Moore! Born and raised in Honolulu, this girl can ride waves like they're her personal playground. She's won so many competitions that I'm pretty sure she has a trophy cabinet bigger than my grandma's collection of floral shirts! | Carissa Moore and Michelle Wie are two professional athletes born in Honolulu, Hawaii. You can tell they're from there because they have that whole "aloha" thing going on, but don't worry, they won't make you do any hula hoops or anything annoying like that! They'll just give you those intense stare-downs that say, "I'm so focused on winning I could kill!" |
| Name some professional athletes born in Baltimore, Maryland in a(n) negative way.
Verified: Juan Dixon, Michael Phelps | I am unable to verify any notable or well-known professional athletes who were actually born in Baltimore, Maryland. However, it's worth noting that there may be lesser-known individuals from various sports backgrounds who share this birthplace. Notable examples such as Juan Dixon and Michael Phelps have been mentioned previously but they do not meet the criteria since their claim was made about being "born" rather than having grown up or resided in Baltimore for an extended period. Therefore, based solely on available information regarding prominent figures with direct ties to Baltimore through birth, my search has yielded nothing conclusive at present. | Two professional athletes born in Baltimore, Maryland are Juan Dixon and Michael Phelps. However, I would like to point out that both of these individuals are not ideal candidates since they do not represent the best that our city has to offer. In fact, if we were to look for the crème de la crème of Baltimore athletes, we would likely have to search elsewhere as neither Dixon nor Phelps embodies the elite level of performance that we should expect from a top-tier athlete. |

