# OpenReview forum: "Precise Information Control in Long-Form Text Generation"
_NeurIPS.cc/2025/Conference — NeurIPS 2025 poster_

### Official Review · Reviewer_8i1m · 2025-06-19

**Clarity:** 4
**Significance:** 2
**Originality:** 3
**Rating:** 4
**Confidence:** 3

**Summary:**

The manuscript proposes a novel task, "Precise Information Control" (PIC), which involves long-form generation that includes all or part of the suggested claims.
By adapting existing language generation benchmarks, the authors introduce PIC-Bench, a dataset consisting of (Instruction, Claims, and Gold-Response) tuples.
By applying SFT and DPO to this dataset, PIC-LM achieves the best or near-best performance among open-source models and demonstrates comparable results to closed-source models.
The authors also verify its viability in real-world scenarios, such as retrieval-augmented generation and self-verification.

**Questions:**

- Please provide some qualitative results analyzing questions (1) and (2) above.
- I am curious whether using all tokens versus only the last $L$ tokens for score computation (as mentioned in line 185) yields similar results.
- Please provide evidence of ChatGPT–Human agreement in your evaluation process.
- Please reconsider the presentation of the POPB-CF benchmark, as it may be misleading.

**Ethical Concerns:**

["NO or VERY MINOR ethics concerns only"]

**Final Justification:**

Please check the additional comments.

**Limitations:**

Although the authors mention their setting—where user-provided knowledge should supersede the LM's inherent priors—in lines 317–319, its implications are not thoroughly discussed. This approach could raise ethical concerns, as it may allow individuals to manipulate the LM to generate any content. This is precisely why several guardrails have been established in the AI community. It would be better if the authors addressed these concerns in more detail.

**Quality:**

3

**Strengths And Weaknesses:**

- **Strenght**
  - Overall, I like this paper. The manuscript is well written. It proposes a dataset, training method, model, and its performance on real-world scenarios. The experiments are thorough and well-motivated.

- **Weakness**
  - **Presentation**
    - Please check the definitions of intrinsic and extrinsic hallucinations. As far as I know, intrinsic hallucination refers to hallucination that can be inferred from the given context, while extrinsic hallucination refers to hallucination that cannot be inferred from the context. ([1] – Sections 2.1 and 2.2 mention that intrinsic and extrinsic hallucinations are subcategories of factual hallucination.) Consider using the terms "faithful" instead of "intrinsic" and "factual" instead of "extrinsic." For example, in the manuscript, the POPB-CF benchmark is faithful but not factual.
    - It would be helpful to include some qualitative results showing (1) why a perfect match is so rare, especially in the PIC-Bench-Full setting, and (2) why instruction-following performance decreases in Table 4 after fine-tuning.
  - **Experiment**
    - I'm not fully convinced why XSUM data is used, as it is "extreme summarization" in a single sentence, as the name implies. Doesn't this conflict with the paper's focus on "long-form text generation"?
    - Why did you sample the $y_{perturb}$ from $\theta_{SFT}$, not superior models? It's a bit worrisome as $\theta_{SFT}$'s capacity does not seem good from your results.
    - I'm also not convinced by the strategy of using the last $L$ tokens for score computation, as described in lines 184–189. The authors mention that this approach is due to efficient computation and length bias, but couldn't the length bias be mitigated by using a different denominator?
    - I read in the appendix, line 1259, that the train and test sets do not overlap. However, it would be better to ensure that the dataset does not have any overlap between the training set and the test benchmark. In other words, it would be better not to use FACTS and biography generation datasets for training. Since the training data uses the same (or similar) test instructions during the training procedure, this would definitely affect the final score, unlike other non-fine-tuned models.
    - It would be helpful to compute the ChatGPT–Human agreement score, as the evaluation process heavily depends on ChatGPT's ability to extract and verify claims, at least for a few examples.
  - **Misc.**
    - It would be better to exclude POPB-CF from the average calculation, as this benchmark directly conflicts with some models' instructions to be factual. From Tables 2 and 4, the main source of the average score gap appears to come from this benchmark, which does not seem fair. (I'm also not sure whether this is an appropriate benchmark, as it could raise certain ethical issues.)

- **Typos**
  - Line 213: "... 8B open-weight baseline (Llama 3.3 70B ..."
  - Check bold in Table 6 (Ministral 8B - QAMParI - Prec.)

**References**
[1] On Faithfulness and Factuality in Abstractive Summarization, ACL 2020

---

> ### Author Rebuttal · Authors · 2025-07-31
>
> We thank reviewer **8i1m** for the thoughtful assessment! We are encouraged that they like this paper, and that they find the manuscript “well-written,” and the experiments “thorough and well-motivated.” Thanks for catching the typos as well! Responding point-by-point:
>
> *“Check the definitions of intrinsic and extrinsic hallucinations…consider using the terms "faithful" instead of "intrinsic" and "factual" instead of "extrinsic.”*
>
> We agree that intrinsic and extrinsic hallucination have particular meanings in prior work [7,8]: (i) intrinsic denotes outputs that contradict the source, and (ii) extrinsic denotes outputs that cannot be verified from the source (and may or may not be true in reality).
>
> To avoid confusion and align with reviewer suggestions, we'll revise our terminology to using “faithful” instead of “intrinsic”, and “factual” instead of “extrinsic.” This means that PIC evaluates faithfulness with respect to supplied verifiable claims; improving faithfulness also improves factuality. We'll update the text and expand the discussion in App. A.3.
>
> **“[Include] qualitative results showing (1) why a perfect match is so rare, especially in the PIC-Bench-Full setting, and (2) why instruction-following performance decreases [after SFT].”**
>
> For (1), we find that a perfect F1 is rare due to distortion errors (where the model rewords claims in ways that distort the original meaning) and fabrication errors (where the model “fills in the blanks” by introducing unsubstantiated information). Please see the categorical error breakdown in our response to reviewer **pcWR**; we'll also include qualitative examples in our revision. We also observe that claim recall tends to be lower than precision, particularly when the sample has many claims, which harms the F1. This is a known problem, as LMs face a lost-in-the-middle effect and struggle to attend to long contexts.
>
> For (2), we notice that instruction-following performance decreases after SFT as the LM learns to optimize for PIC at the expense of following instruction constraints, e.g., not following word counts.
>
> Here is an ExpertQA example (with a Prometheus score of 3/5). While this response provides a basic overview, it fails to abide by the pros-and-cons format required by the instruction:
>
> ```
> Instruction: Give me an overview of the free granular synthesizer software with pros and cons for each option. [Claims omitted]
>
> PIC-LM SFT Output:
> Here are some options for free granular synthesizers
> 1. Granulab - This program has many features including real-time audio processing
> 2. Argotlunar - Another feature-rich granulator
> 3. Hourglass - A good choice for those who want more control over their sound
> 4. Fragment - Great for collaboration or live performance
> 5. PaulXStretch - For advanced users looking to push the limits of time-stretching
> Each one has its own strengths and weaknesses so it really depends on how you plan to use them.
> ```
>
> **“Doesn't [XSUM’s "extreme summarization"] conflict with the paper's focus on "long-form text generation"?”**
>
> While we use the data from XSUM, we do not adopt its one-sentence “extreme” summarization protocol, and instead prompt for standard summaries. We consider this a long-form generation task as we could extract a sufficient number of verifiable output claims; e.g., the PIC XSUM task has 30.6 avg. input claims, and PIC-LM 8B responses average around 21.8 claims. We’ll clarify the misleading task name in the revision.
>
> **“Why did you sample the y_perturb from θ_SFT, not superior models?”**
>
> We sampled $y_{perturb}$ using $θ_{SFT}$ as it’s a cheap proxy for API-based LLMs. We reuse SFT data for preference data construction, so these perturbed responses already exhibit strong PIC behavior as $θ_{SFT}$ has seen similar samples. To empirically validate the quality of $y_{perturb}$, we randomly sample 50 full and partial PIC samples from our preference data where the perturbed sample is preferred, and find high PIC scores (**84.3%** F1 for full PIC, **86.5%** precision for partial PIC); these numbers are competitive with GPT-4o sampling. While not perfect, these signals are still valuable for preference tuning, as recent work (the Delta Learning Hypothesis [9]) suggests that the relative delta between chosen and rejected responses is more important than absolute response quality.
>
> **“[For using] the last L tokens for score computation…couldn't the length bias be mitigated by using a different denominator?”**
>
> Following [10], we compute the avg. per-token log-likelihood over the last L=20 tokens of each response to reduce computational cost and length bias; in this setting, the shorter response is chosen **49.0%** of the time.
>
> We ran a new ablation where we recreate preference data by computing the per-token log probability over the entire response instead. This results in a length bias with the shorter response preferred **60.2%** of the time. Training on length-imbalanced preferences leads to worse avg. PIC performance (**91.0%** vs. **91.4%** for full, and **92.6%** vs. **93.7%** for partial PIC) and Prometheus scores (**3.89** vs. **3.92**). Thus, we retain last-L-token scoring for its balance of efficiency and performance. Alternative normalization strategies, e.g., length-aware scoring, or an adaptive L, may improve robustness; we leave this to future work.
>
> **“The train and test sets do not overlap [and] it would be better…not to use FACTS and biography generation datasets for training.”**
>
> Thanks for raising this point! First, we emphasize that the PIC-LM training and PIC-Bench evaluation datasets are completely disjoint; there’s no overlap between training and test splits.
>
>  Second, our evaluation is designed to measure both in-distribution (ID) and out-of-distribution (OOD) generalization. We include EntityBios and FACTS in PIC-Bench to assess ID performance for PIC-LM, which had exposure to these tasks during training. The other tasks (along with a biomedical task in App. C.7) serve as OOD evaluations, allowing us to measure performance on unseen domains. PIC-LM shows good OOD performance (**90.4%** avg. OOD F1, competitive with GPT-4o at **90.3%**). We'll make the ID/OOD distinction clear in the revision.
>
> Lastly, EntityBios and FACTS contain significantly more claims (**51** and **64** on average, respectively) and are more challenging than other domains. High‑quality instruction data with long, rich contexts is scarce and expensive to curate, so these domains are instrumental to improving PIC.
>
> (Note that the ID/OOD PIC-LM numbers cited above reflect updated training results after stricter data quality control.)
>
> **“It would be helpful to compute the ChatGPT–Human agreement score.”**
>
> We ran human evaluations; please see our response to reviewer **Dux5** for setup details. For claim extraction, the LLM-human agreement averaged 97.5%, with inter-annotator agreement at **94.1%**. For claim verification, the LLM-human agreement is **81.0%**, with substantial inter-annotator agreement (Fleiss kappa=**.694**).
>
> **“It would be better to exclude POPB-CF from the average calculation…[the] average score gap appears to come from this benchmark.”**
>
> Thanks for this perspective. PopB-CF is intentionally designed as a stress test: it modifies biographical claims to conflict with models’ parametric knowledge to assess whether LMs can follow explicit user instructions, even when doing so contradicts their internal priors. This scenario simulates realistic safety-relevant settings, such as updating outdated facts or enforcing user-specific overrides. In these cases, instruction-following behavior is desirable, even if it overrides a model’s factual default.
> To ensure a fair comparison, we agree it’s valuable to report both the average with and without PopB-CF. In our revision, we'll include both versions to make these tradeoffs transparent. Without PopB-CF, the top-3 highest avg. PIC F1s are GPT4o (**93.7%**), PIC-LM 8B (**93.0%**), and Llama 3.3 70B Inst. (**89.9%**), and PIC-LM 8B shows similar Prometheus scores to Llama 3.1 8B Inst. (**3.94** vs. **3.99**).
>
> **“[The] implications [that user-provided knowledge should supersede the LM's inherent priors] are not thoroughly discussed.”**
>
> We appreciate the reviewer’s concern. Our work aims to make LMs more controllable by giving users finer-grained command over their generations.
>
> Stronger control over model outputs may indeed exacerbate misuse. PIC-LM is more capable of generating counterfactual or harmful content if such claims are passed as input, and may override abstention behaviors instilled via RLHF. However, this tradeoff should be contextualized. Uncontrolled toxic generation, arising from model hallucinations, can be harder to detect and filter. By contrast, PIC-LM makes the provenance of outputs attributable to user input, allowing developers to place safeguards upstream.
>
> Our contribution follows work showing that safety alignment is shallow and brittle [11]. PIC-LM shifts the safety locus from the model’s parameters to auditable inputs. This can lead to more robust safety systems, such as pairing PIC-LM with a pre-hoc safety classifier (e.g., LlamaGuard) that screens for malicious claims. Therefore, while PIC-LM may not be the safest model in isolation, we see it as a promising building block for safety-aware systems where user-grounded control is critical.
>
> We'll expand the limitations and broader impacts (App. A.4) in our revision.
>
> Citations
>
> [7] Maynez et al., 2020. On Faithfulness and Factuality in Abstractive Summarization
>
> [8] Huang et al., 2025. A Survey on Hallucination in Large Language Models: Principles, Taxonomy, Challenges, and Open Questions
>
> [9]  Geng et al., 2025. The Delta Learning Hypothesis: Preference Tuning on Weak Data can Yield Strong Gains
>
> [10] Chuang et al., 2025. SelfCite: Self-Supervised Alignment for Context Attribution in Large Language Models
>
> [11] Qi et al., 2024. Safety Alignment Should Be Made More Than Just a Few Tokens Deep

---

> > ### Comment · Reviewer_8i1m · 2025-08-01
> >
> > Thank you for the thorough rebuttal, and I feel more positive about this work.
> >
> > However, I choose to leave my score as before, as I'm still concerned about the suggested limitation.
> >
> > Though authors mentioned PopB-CF as a stress test, I guess this point is not limited to the "model’s factual default", the authors mentioned.
> >
> > For example, if a model is instructed to write an article with a false claim A, a usual instruction-tuned model is trained to say, 'I cannot write an article with false information'.
> >
> > I am not sure training a model to write against this instruction is genuinely beneficial... and I am also not sure whether it's valid to include this task as a benchmark.
> >
> > However, I want to mention that I definitely became more positive with the paper. I would choose 4.5 as my final score if there were.

---

> > > ### Author Response · Authors · 2025-08-02
> > >
> > > Thank you again for the support and the thorough feedback! We are very glad that our rebuttal has positively shifted your view, and appreciate the time you have spent engaging with our paper.
> > >
> > > If I’m understanding your point correctly, an instruction-tuned LM should ordinarily refuse to output a given false claim. We agree that this behavior is generally useful; however, the LM should also be able to faithfully follow input claims when explicitly asked to anchor on context. Under these instructions, the LM is expected to follow user-supplied claims even when they conflict with its own internal beliefs.  For PIC-Bench’s evaluation purposes (which targets claim-level controllability), we believe this is acceptable for two reasons:
> > >
> > > First, the truth is not always accurately encoded in the LM parameters. Parametric knowledge can be stale or wrong. For example, “Agatha Christie’s *The Seven Dials Mystery* is a novel  under U.S. copyright law.” is factually incorrect (as it entered the U.S. public domain on Jan. 1, 2025), but LMs trained in 2024 and earlier would consider this to be true. In such cases, deference to user-provided claims is desirable.
> > >
> > > Second, PopB-CF’s primary goal is to measure context compliance, even under the extreme scenario where context contradicts world knowledge. This stress-test follows a line of established work [1, 2, 3, 4, 5] in which LMs must anchor to user-supplied, deliberately counterfactual knowledge; we even adapt PopB-CF directly from [1].
> > >
> > > In practice, developers look at benchmarks holistically and may choose particular subsets to evaluate on. PopB-CF tests an edge case of context grounding which indeed may not suit every scenario. To reflect this, we will report PIC-Bench numbers with and without PopB-CF.
> > >
> > > Finally, PIC-LM is designed to be a more controllable LM that reliably adheres to input claims (notably, it is not trained on any counterfactual data). While PIC-LM does yield gains on PopB-CF, we also show that it can produce *more factual* generations when integrated into larger systems (i.e., RAG and chain-of-verification pipelines), suggesting that controllability and factuality can be complementary.
> > >
> > >
> > > Citations:
> > >
> > > [1] Goyal et al. Context-Parametric Inversion: Why Instruction Finetuning May Not Actually Improve Context Reliance. ICLR 2025.
> > >
> > > [2] Li et al. Large Language Models with Controllable Working Memory. ACL Findings 2023.
> > >
> > > [3] Longpre et al. Entity-Based Knowledge Conflicts in Question Answering. EMNLP 2021.
> > >
> > > [4] Ming et al. FaithEval: Can Your Language Model Stay Faithful to Context, Even If “The Moon is Made of Marshmallows.” ICLR 2025.
> > >
> > > [5] Yan et al. RefuteBench: Evaluating Refuting Instruction-Following for Large Language Models. ACL Findings 2024.

---

### Official Review · Reviewer_UZLz · 2025-06-28

**Clarity:** 3
**Significance:** 2
**Originality:** 2
**Rating:** 4
**Confidence:** 5

**Summary:**

The paper introduces a task called Precise Information Control (PIC) in an attempt to measure the ability of LMs to generate long-form text grounded in a set of provided input statements, i.e. without intrinsic hallucinations. In this task, the LM is provided with a set of short verifiable claims and an instruction to generate the long form output in one of 2 settings - a *full setting* in which the model is expected to utilize all provided claims in the generation, and a *partial setting* in which the model can choose to select relevant claims to be included. A verifiable claim extractor from past work (VeriScore, Song et. al. 2024) is utilized for extracting claims from summarization, biography generation & QA datasets to construct the input data, which results in a set of 8 datasets curated in the PIC setting and (expected to be) released as a benchmark for this task. The claim extractor is also applied to the generated response in order to compute support with supplied input claims.
Apart from off the shelf LLMs, a Llama 3.1 8B Instruct model is fine tuned using a) SFT b) DPO and compared against baselines. The model finetuned with DPO shows better performance than most instruction tuned & reasoning models for both full & partial settings.

**Questions:**

1. Given that the LLMs are sensitive to the instruction prompts, were there experiments conducted by varying the prompts used for the problem for the full & partial settings? Specific instruction prompts are likely to garner better performance from the models.
2. Could you elaborate on how the gold response was obtained (line 111)?
3. A few lines on the model used to compute support for eval (line 115) would be helpful. Support is typically posed as an NLI task in these cases, but it would help to understand what was used here.
4. Which instruction-following model is referred to in line 184?
5. It is good to see the DPO-finetuned Llama 8B model to be at par/outperform other baselines in Tables 2 & 3. However, it's curious to see the SFT model degrade in performance w.r.t the same base model in F1 for AskH, ELI5, ExpertQA & FACTS. Do you have any thoughts on why that might be the case?

**Ethical Concerns:**

["NO or VERY MINOR ethics concerns only"]

**Final Justification:**

Updating my score by 1 based on the rebuttal, particularly the details provided on comparing PIC with RAG, including the new experimental results.

**Limitations:**

Yes

**Quality:**

3

**Strengths And Weaknesses:**

Strengths:
1. Several types of datasets have been covered for this problem, i.e. summarization, biography generation & QA, which is commendable.
2. The paper is fairly well-written and contains exhaustive experiments in several directions, e.g. analyses on instruction following capabilities of the LM, comparison with RAG settings, etc.

Weaknesses:
1. Precise Information Control has been presented as a new task. However, it is very similar to RAG & attributable generation (without the need for explicit citations) as the paper also states (Sec 6, 7), where the generated response is required to be grounded in provided source references. Both the task setting and the evaluation processes are similar between the two. The authors attempt to address this by stating that the full PIC setting requires inclusion of all input claims, but that does not suffice in terms of novelty since this would be achievable in the attributable generation setting by modifying the input instruction prompt to direct the LM to include all source refs as well.
2. The task expects claim extraction as a necessary step to be conducted from typically noisy input source material that is generally present in RAG settings. This creates a large dependency on efficacy of the claim extractor model, and assumes that only relevant claims will make their way into the input. This seems to be an unrealistic setting to expect in the real-world.

---

> ### Author Rebuttal · Authors · 2025-07-31
>
> We thank reviewer **UZLz** for the constructive feedback. We are encouraged that the reviewer finds the breadth of datasets covered in PIC-Bench to be “commendable,” and that our paper is “well-written [with] exhaustive experiments in several directions.” Responding point-by-point to their comments:
>
> **“[PIC] is very similar to RAG & attributable generation (without the need for explicit citations)...[full PIC] would be achievable in the attributable generation setting by modifying the input instruction prompt to direct the LM to include all source refs as well.”**
>
> Although PIC shares surface similarities with RAG and attribution, this framework is fundamentally about information control—ensuring that every output claim is traceable to user-provided input. This goal differs from that of RAG and attribution generation, both of which primarily aim to improve factuality. We show that PIC is surprisingly difficult for LMs; when provided with exactly which claims to include, even frontier LMs may drop, distort, or fabricate content. This fundamental control problem motivates a dedicated evaluation framework, benchmark, and training techniques.
>
> While PIC, RAG, and attributable generation all require grounded responses, we note several key differences. First, PIC enforces full coverage of all input claims, while RAG allows for information omission if final answers are correct. Second, attribution systems often work at the sentence or passage level, while PIC operates at verifiable claim level, enabling a finer-grained detection of distortion, fabrication, and conflation. Prompting an LM to “include all sources” can only approximate this in a best-effort way, and cannot guarantee coverage of every input claim. Lastly, PIC has broad task coverage. While RAG and attribution generation primarily focus on QA, PIC is more universally applicable to a range of generative tasks.
>
> Improvements on PIC may also benefit related tasks. We demonstrate strong factuality gains in our RAG experiment (Section 6.1), and leave attribution generation to future work. Notably, PIC-LM outperforms Self-Cite 8B, a strong attribution generation LM, across both partial and full PIC settings (our RAG and self-verification pipeline case studies).
>
> **“[PIC] expects claim extraction as a necessary step to be conducted from typically noisy input source material…This creates a large dependency on [the] efficacy of the claim extractor model, and assumes that only relevant claims will make their way into the input…[which seems] unrealistic.”**
>
> Thanks for pointing this out. For PIC-Bench evaluation and our RAG experiments, we use a claim extractor from [6] that decomposes a given source text into its constituent verifiable claims. We emphasize that this extractor is not prompt-aware, and does not filter claims based on user instruction.
>
> We would also like to correct the assumption that PIC assumes that only relevant claims will be in the input. In partial PIC (which is used for RAG), the LM is fed a complete claim set that may contain noisy, irrelevant, or contradictory claims. The LM must select *a relevant subset of claims* to ground its output, without adding any unsupported claims.  Concretely, it's the LM, not the claim extractor, that must decide which claims to incorporate.
>
> We ran RAG experiments with retrieved passages processed as claims as PIC-LM is trained with this format. However, we conduct a new RAG ablation, where we *omit the claim extraction step entirely* and instead pass in raw retrieved passages to PIC-LM 8B (“PIC-LM 8B (Psg.-level)”). This setup corresponds to standard in-context RAG, and simulates noisy input without explicit claim structuring. Despite being trained only on structured claims, PIC-LM generalizes well to this unstructured setting and still outperforms Llama 3.1 8B Inst. by a significant margin (**59.7** vs. **52.3** EM) on ASQA:
>
> | **Baseline**                  | **Exact Match** |
> |------------------------------|------------------|
> | Llama 3.1 8B Inst.           | 52.3             |
> | PIC-LM 8B                    | 59.4             |
> | **[New]** PIC-LM 8B (Psg.-level)       | **59.7**         |
>
> PIC-LM improves downstream RAG performance as it grounds on retrieved context more thoroughly. Please note that not all PIC use cases rely on an auxiliary claim extractor. For example, in our chain-of-verification pipeline experiments, the LM autonomously generates and self-corrects its claims to improve end-task factuality. We will incorporate these results and discussion into our revision.
>
> **“Were there experiments conducted by varying the prompts used for [PIC evaluation]?”**
>
> We ran a new ablation experiment with stronger, more explicit prompts and found PIC-Bench to be largely prompt-insensitive. We report absolute performance changes (%)  for PIC-LM 8B and three baselines when evaluated using the original prompt (from the paper) versus an alternative prompt pair:
> - **Full PIC:** “You are an assistant that strictly adheres to given instructions. Be sure to incorporate each provided  input claim in your response, and do not introduce or omit any claims.”
> - **Partial PIC:** “You are an assistant that strictly adheres to given instructions. Respond using any subset of given claims that you find relevant, but do not introduce any new claims.”
>
> | **PIC Metric (%)** | **Llama 3 8B Inst.** | **Tulu 3 8B** | **Ministral 8B Inst.** | **PIC-LM 8B** |
> |:-------------------------------:|:--------------------:|:------------:|:----------------------:|:-------------:|
> | **Avg. F1**                     | 1.8                  | 1.1          | -2.4                   | 0.4           |
> | **Avg. Perf. F1**               | 0.4                  | 0.0          | -0.4                   | 3.2           |
> | **Avg. Prec.**                  | 0.8                  | 3.2          | 1.2                    | 0.4           |
> | **Avg. Perf. Prec.**            | -0.3                 | 0.1          | 3.0                    | 4.2           |
>
> As shown, changes in PIC performance are relatively small across prompt variations, with most metric shifts **<4%**. These results indicate that PIC-Bench evaluations remain stable across reasonable prompt variations. We will include these results in our revision. While certain prompts may slightly improve absolute scores, we believe they are unlikely to affect the relative ranking between models, which is our primary evaluation focus.
>
> **“Could you elaborate on how the gold response was obtained (line 111)?”**
>
> The long-form gold response $y$ comes from the instruction-following dataset, $D_{IT}$. Many off-the-shelf instruction-following datasets (e.g., Alpaca) are formatted as instruction-response pairs $(I, y)$, which can be easily converted to the PIC format $(I, C, y)$, wherein $C$ is the set of extracted verifiable claims from the response $y$. We'll clarify this point in our revision.
>
> **“A few lines on the model used to compute support for eval (line 115) would be helpful.”**
>
> Thanks for the suggestion! We use an off-the-shelf claim verifier introduced by VERISCORE [6], which prompts an LLM (in our case, GPT4o-mini) to judge whether a verifiable claim is supported or unsupported by a corresponding set of evidence claims. Note that [6] has found substantial agreement between this claim verifier and human annotators; we also corroborate this across our PICBench tasks and baseline generations in our own human evaluation (please see our response to reviewer **Dux5**).
>
> **“Which instruction-following model is referred to in line 184?”**
>
> We take **Llama 3.1 8B Instruct** as the instruction-following model used to score responses for PIC-LM preference data creation. We choose this model because in practice,  (1) it shows strong agreement with GPT-4o in our LLM-as-a-judge comparison (see Appendix C.5), and (2) as a relatively small, open LM, it enables efficient and cost-effective data selection.
>
> **“It's curious to see the [PIC-LM] SFT model degrade in performance w.r.t the same base model in F1 for AskH, ELI5, ExpertQA & FACTS.”**
>
> We thank the reviewer for pointing this out. AskH, ELI5, ExpertQA, and FACTS are the four long-form QA tasks on PIC-Bench, and PIC-LM SFT shows a drop in PIC performance compared to the base model Llama 3.1 8B Inst. on these tasks. We believe there are two reasons: first, there is a mismatch between the token-level cross-entropy loss used for SFT and the PIC objective, which emphasizes exact adherence to a constrained set of claims. During SFT, the model may learn to generate broadly informative responses that resemble typical QA-style outputs but deviate from strict PIC constraints, leading to improved recall but reduced precision, and thus lower F1.
>
> The second factor is SFT training data quality. Table 20 (in Appendix C.4) shows that training only on filtered samples with high PIC scores improves both SFT and DPO performance. Since submission, we have experimented with raising the minimum PIC score threshold from **0.9** to **1.0** for all training examples except those in the biography domain; as shown below, training on more stringently filtered data improves PIC performance on long-form QA.
>
> | **Model**              | **AskH F1** | **ELI5 F1** | **ExpQA F1** | **FACTS Prec.** |
> |:----------------------:|:----------:|:----------:|:------------:|:----------------:|
> | (Former) PIC-LM SFT    |   81.0%     |   77.5%     |     70.0%     |      63.1%        |
> | (Former) PIC-LM        |   94.2%     |   92.2%     |   **87.0%**   |      89.0%        |
> | (New) PIC-LM SFT       |   87.2%     |   82.1%     |     77.3%     |      81.2%        |
> | (New) PIC-LM           | **96.0%**   | **94.1%**   |     85.6%     |    **89.8%**      |
>
> We hope our responses have adequately addressed your concerns. Please let us know if there are any other points we can clarify.
>
> Citations:
>
> [6] Song et al., 2023. VERISCORE: Evaluating the factuality of verifiable claims in long-form text generation.

---

> > ### Author Response · Authors · 2025-08-06
> >
> > Dear Reviewer **UZLz**,
> >
> > Thank you again for your thoughtful review. If you have any follow-up questions or concerns on our rebuttal, we would be happy to address them before the discussion period closes. For ease of review, we summarize the main updates below:
> >
> > - **W1**: We discussed the broader motivation of the PIC framework, and provided a clearer comparison to RAG and attributable generation.
> > - **W2**: We clarified that PIC does not require an auxiliary claim extractor for downstream usage, and added a RAG ablation experiment without the claim extraction step that maintains improved factuality performance.
> > - **Q1**: We added an ablation showing that PICBench evaluation is robust to prompt variations.
> > - **Q2, Q3, Q4**: We provided additional implementation details on PIC evaluation and PIC-LM training data selection.
> > - **Q5**: We provided new experiments showing improved PIC-LM performance on PICBench after stricter data quality filtering.
> >
> > We will incorporate these updates into our revised manuscript. We appreciate your time and helpful feedback!

---

> > ### Comment · Reviewer_UZLz · 2025-08-08
> >
> > I thank the authors for taking the time to provide detailed clarifications in the rebuttal. The result obtained using PIC-LM in the RAG setting on ASQA is particularly interesting and would be a good addition to the paper. This would help highlight the benefit of applying the LM trained for the PIC task to a more generic & non fine-grained setting such as RAG, directly.
> > However, for future work, I would suggest looking into experiments & ablation studies focusing on the claim extractor since the overall method heavily relies on its performance.
> >
> > I am inclined to increase my score by 1 point at this time.

---

### Official Review · Reviewer_pcWR · 2025-07-03

**Clarity:** 3
**Significance:** 2
**Originality:** 3
**Rating:** 5
**Confidence:** 4

**Summary:**

Precise Information Control (PIC) is a new generation task that eliminates intrinsic hallucinations by demanding outputs containing exactly and only a supplied list of factual claims. The authors construct PIC-Bench, eight datasets spanning biographies, QA, and news summaries in both “full” (all claims required) and “partial” (only relevant claims) modes, and show that even GPT-4 and Claude add or omit claims in over 70 % of cases. They fine-tune an 8B Llama-3 model with SFT + DPO to obtain PIC-LM, which delivers higher factual fidelity while preserving general instruction-following skills (judged by Promotheus). They also show that plugging PIC-LM into retrieval-augmented QA and self-verification pipelines yields noticeably more trustworthy answers.

**Questions:**

* Consider providing a breakdown of remaining errors (missing vs. unsupported claims, topical patterns) and discuss how those insights can guide future mitigation strategies.
* In the partial PIC setting, only a subset of provided claims should be used. It would help to know how the “relevant” subset is determined for evaluation. The paper could be more explicit on whether this is based on the original reference output or some information retrieval heuristic. For example, if multiple claims in the context could have been relevant but the model was penalized for including them (because the reference didn’t), that might unfairly penalize the model.
* How do the PIC-LM outputs do in terms of coherence and style, beyond just containing the required facts? It would be great to include a small qualitative analysis or human evaluation of output quality.
* By design, PIC penalizes any extra information not present in the input, even if that information is factually correct. This stringent criterion is appropriate for measuring hallucination, but it could be argued that in some applications a harmless additional detail (especially if true) might not be a failure. Could the authors provide more discussion on this nuance?

**Ethical Concerns:**

["NO or VERY MINOR ethics concerns only"]

**Final Justification:**

The authors have thoroughly addressed my concerns from the original review, providing several new results that will be valuable additions to the paper. Accordingly, I have decided to raise my score.

**Limitations:**

yes

**Quality:**

3

**Strengths And Weaknesses:**

Strengths

* The paper introduces a distinct and granular formulation of the hallucination problem. Instead of treating intrinsic hallucination as a binary phenomenon, PIC enforces and measures faithfulness at the level of individual factual claims. This per-claim control is a novel perspective. It goes beyond prior work by requiring exact inclusion/exclusion of specific information units, a stricter criterion than traditional factuality metrics.
* PIC-Bench covers eight tasks spanning different domains and formats. These include: generating Wikipedia-style biographies (EntityBios, PopBios), open-ended Q\&A from Reddit (ELI5, AskHistorians), expert-curated Q\&A (ExpertQA), abstractive news summarization (XSUM), and a long-document grounded generation task (FACTS).
* An 8B PIC-LM lifts full-PIC F1 from 69.1% to 89.9% and delivers strong precision improvements in downstream QA/verification pipelines.

Weaknesses

* The fine-tuned PIC-LM is specialized to follow given facts religiously. A possible weakness is how this specialization affects its general language abilities or flexibility. The paper includes an alignment evaluation (“Prometheus score”) suggesting only a minor drop in general helpfulness, but a more extensive evaluation on non-PIC tasks would strengthen confidence in PIC-LM’s general usability.
* PIC penalizes any additional (even correct) information; the paper lacks discussion or ablation on whether such rigid constraints harm user value in contexts where minor extra facts are tolerable.
* The evaluation focuses almost entirely on factual inclusion metrics (F1, precision of claims) and Prometheus scores. There is little assessment of the linguistic quality or user-perceived quality of the outputs.
* The research focuses on an 8B-parameter model for fine-tuning. While this is pragmatic, it remains unclear how well this method would scale to bigger or smaller models.

---

> ### Author Rebuttal · Authors · 2025-07-31
>
> We thank reviewer **pcWR** for their time and feedback! It is encouraging that the reviewer finds our PIC framework to bring “a novel perspective” by enforcing per-claim control, “[going] beyond prior work” and that our PIC-LM “delivers strong precision improvements” in concrete downstream use cases. We address their questions and suggestions below:
>
> **“PIC-LM is specialized to follow given facts religiously…[which may] affect its general language abilities or flexibility. [A] more extensive evaluation on non-PIC tasks would strengthen confidence in PIC-LM’s general usability.”**
>
> We did not evaluate PIC-LM on general language benchmarks as it’s not meant to be a standalone LM, but rather a highly faithful module within larger systems for grounded generation (Sect. 6). For example, using PIC-LM as a drop-in replacement in RAG or chain-of-verification pipelines boosts end-task factual accuracy (+**14%** on ASQA, +**21%** on Birthplaces, +**6.9%** on QamPARI).
>
> Although PIC-LM is a specialized model (its alignment process is solely focused on PIC), for transparency, we report new results on standard benchmarks (MMLU and ARC-C using OLMES [1]; GSM8K using LM-Harness [2]; and AlpacaEval 2 [3]):
>
> | **Benchmark**                                      | **Llama 3.1 8B Inst.** | **PIC-LM 8B** |
> |---------------------------------------------------|:----------------------:|:-------------:|
> | 5-shot MMLU Acc. (Raw Completion)                 |          43.6          |   **44.2**    |
> | ARC-Challenge Acc. (Raw Completion)               |          55.6          |   **58.7**    |
> | 5-shot GSM8K Exact Match                          |        **75.7**        |     69.3      |
> | AlpacaEval 2 Length-Controlled Win Rate           |        **20.9**        |     17.1      |
>
>
> We emphasize that PIC-LM is an initial model to tackle PIC, which still remains an open problem. For example, it is possible to instill general-purpose LMs with better PIC abilities by mixing in PIC-formatted data during post-training. We leave this direction to future work, and will include the above results in our paper with a more in-depth discussion.
>
> **“PIC penalizes any additional (even correct) information; the paper lacks discussion or ablation on whether such rigid constraints harm user value in contexts where minor extra facts are tolerable…could the authors provide more discussion on this nuance?”**
>
> We acknowledge that PIC’s strict grounding requirement isn’t appropriate for every scenario (e.g., creative storytelling, open-ended reasoning). Rather, PIC is built around user-specified control: if the user explicitly instructs the model to only ground its response on given claims, the LM should comply, even if this means forgoing extra correct details. Enforcing these constraints is necessary in fast-moving, high-stakes domains (e.g., scientific consensus on infectious diseases), where plausible but unverified information pose serious risks.  We will incorporate these points into the discussion in our revision.
>
> **“There is little assessment of the linguistic quality or user-perceived quality of [PIC evaluation] outputs…How do the PIC-LM outputs do in terms of coherence and style, beyond…required facts? …Include a small qualitative analysis or human evaluation of output quality.”**
>
> Thanks for the suggestion! We ran a new small-scale qualitative analysis by sampling 40 PIC-Bench instructions and comparing responses from Llama 3.1 8B Instruct and PIC-LM 8B along two dimensions: coherence and fluency. Following prior work [5], we define coherence as the degree to which  the ideas in the text flow together logically (i.e., whether the sentences are on-topic, and follow a clear progression of thought). We define fluency to be how grammatically correct, smooth, and natural the text is.
>
> For each output pair, we indicated a preference for Llama 3.1 8B Inst., PIC-LM 8B, or a tie along each dimension. All outputs were evaluated blindly in shuffled order, without revealing model identity, to reduce annotation bias. The results are as follows:
>
> |                         | **Prefer Llama 3.1 8B Inst.** | **Tie** | **Prefer PIC-LM 8B** |
> |-------------------------|:-----------------------------:|:-------:|:--------------------:|
> | **Coherence**           |           25%               |  27.5%  |      **47.5%**       |
> | **Fluency**             |            2.5%               |  **72.5%**|        25%           |
>
>
> These results suggested that PIC-LM 8B outputs tend to be more coherent than Llama 3.1 8B Inst. Qualitatively, we find that PIC-LM generations are more concise and factual, whereas Llama generations are more verbose. Both model generations are similarly fluent. we'll include this experiment in our revision, and provide more qualitative examples of PIC-LM generation.
> Note that PIC-LM can also be steered along stylistic dimensions while preserving claim grounding. We demonstrate this capability in a style transfer experiment in Appendix D.2.1.
>
> **“It remains unclear how well [PIC-LM training] would scale to bigger or smaller models.”**
>
> We ran new PIC-LM training experiments starting from 1B, 3B, and 70B Llama 3 Instruct models, and found strong PIC-Bench improvements in all settings. Note that due to computational constraints, we trained the 70B PIC-LM with QLoRA fine-tuning. Please see the PIC-Bench results below:
>
> | **Model Setting** | **Avg. F1** | **Avg. Perf. F1** | **Avg. Prec.** | **Avg. Perf. Prec.** |
> |-------------------------------------|:-----------:|:-----------------:|:--------------:|:---------------------:|
> | Llama 3.2 1B Inst.                  |    54.8     |       1.5         |     55.0       |         18.3          |
> | **PIC-LM 1B**                       |  **75.5**   |     **5.1**       |   **76.3**     |       **30.0**        |
> | Llama 3.2 3B Inst.                  |    54.8     |       1.6         |     63.0       |         21.4          |
> | **PIC-LM 3B**                       |  **87.0**   |    **17.8**       |   **82.4**     |       **47.7**        |
> | Llama 3.3 70B Inst.                 |    82.5     |      13.1         |     74.3       |         49.5          |
> | **PIC-LM 70B (QLoRA)**             |  **93.2**   |    **33.2**       |   **95.9**     |       **51.8**        |
>
>
> **“Consider providing a breakdown of remaining errors (missing vs. unsupported claims, topical patterns) and discuss how those insights can guide future mitigation strategies.”**
>
> Thanks for the suggestion! We manually examined **40** PIC-LM generations on PIC-Bench with imperfect scores (5 samples per PIC evaluation task). This sample has **1165** output claims in total, of which **68** response claims are marked as unsupported (affecting precision), and **63** input claims are marked as missing (affecting recall for the full PIC setting).
>
> We provide a categorical error breakdown of the 68 **unsupported** response errors below:
>
> | **Error Type** | **Explanation**                                                        | **Count (Percentage)** |
> |----------------|------------------------------------------------------------------------|-------------------------|
> | Distortion     | Claim is reworded in a way that shifts its original meaning            | 27 (39.7%)              |
> | Fabrication    | Claim comes from the LM’s parametric memory instead of input claims    | 20 (29.4%)              |
> | Conflation     | Claim is merged from two or more distinct input claims                 | 12 (17.6%)              |
> | Verifier       | Claim is supported but mis-flagged by the verifier                     | 9 (13.3%)               |
>
>  Among the 63 **missing** claims, we observe no obvious qualitative differences, but a mild head-bias: **27%** of drops occur within the first quarter of the list (vs. **22%** in the final quarter), indicating that earlier claims are slightly more likely to be omitted, and corroborating the “lost-in-the-middle” context phenomenon. We'll add these error analyses to our revision.
> For mitigation strategies, we believe it would be beneficial to explore improved data selection during training, such as incorporating hard negatives with subtle distortion and fabrication errors. Additionally, adopting techniques from long-context modeling (e.g., attention re-weighting, recurrence, or segment-aware encodings) may help address position-based errors, such as the observed head-bias in missing claims.
>
>
> **“In the partial PIC setting, only a subset of provided claims should be used. It would help to know how the “relevant” subset is determined for evaluation.”**
>
> In the partial PIC setting, it is up to the LM to select the relevant subset from the input claims. PIC does not measure the quality of the answer, but just ensures that every claim in the answer is fully grounded. PIC is reference-free in that the output isn’t compared against a given gold answer.
>
> To measure actual generation quality  (e.g., if the response adequately satisfies the task instruction), we pair PIC evaluation with Prometheus, an LLM-based judge that scores outputs along a user-defined rubric and shows high agreement with human evaluations. As our PIC evaluation framework provides well-formed claims, we find that our Prometheus scores are relatively high (see Table 4).
>
>
>
> Citations:
>
> [1] Gu et al., 2025. OLMES: A Standard for Language Model Evaluations
>
> [2] Biderman et al., 2024. Lessons from the Trenches on Reproducible Evaluation of Language Models
>
> [3] Dubois et al., 2024. Length-Controlled AlpacaEval: A Simple Way to Debias Automatic Evaluators
>
> [4] Liu et al., 2023. Lost in the Middle: How Language Models Use Long Contexts
>
> [5] Li et al., 2023. Contrastive Decoding: Open-ended Text Generation as Optimization

---

> > ### Comment · Reviewer_pcWR · 2025-08-06
> >
> > Thank you for addressing my comments in detail and providing new results! These will be valuable additions to the paper. I will update my score accordingly.

---

> > > ### Author Response · Authors · 2025-08-06
> > >
> > > We thank you for the helpful suggestions! We will revise our manuscript accordingly with the additional experiments and discussion.

---

### Official Review · Reviewer_Dux5 · 2025-07-06

**Clarity:** 3
**Significance:** 3
**Originality:** 2
**Rating:** 4
**Confidence:** 2

**Summary:**

This paper focuses on the intrinsic hallucination and introduces Precise Information Control (PIC), a new task formulation for evaluating and improving language models’ ability to generate long-form text with no intrinsic hallucinations, which is strictly grounded in provided input claims without introducing unsupported information. There are two proposed evaluation settings: full PIC and partial PIC. They also develop PIC-bench, a benchmark with 8 long-form generation tasks, and introduce PIC-LM, an 8B parameter model trained through a two-stage approach combining supervised fine-tuning with preference optimization using a novel weakly-supervised data construction method.
They demonstrate that even SOTA models have intrinsic hallucination in over 70% of outputs, while PIC-LM achieves better performance in both settings.

**Questions:**

See weaknesses.

**Ethical Concerns:**

["NO or VERY MINOR ethics concerns only"]

**Final Justification:**

My questions are answered. However I'm not an expert in this area, I cannot be confident enough to provide helpful reviews.

**Limitations:**

yes

**Quality:**

2

**Strengths And Weaknesses:**

Strengths:
- Well-presented, good motivation: I like the motivation of the paper, PIC-Bench could become standard benchmark for intrinsic hallucination research. In real world, medical, legal, and journalistic domains urgently require controllable information generation. Also, intrinsic hallucination has been understudied compared to extrinsic hallucination; this work can serve as a foundation for future intrinsic hallucination research.
- Precise framework for a more fine-grained task: this paper transforms the vague binary judgment evaluation into precise claim-level constraints, also why “verifiable claim constraints” rather than sentence and atomic is well supported. The authors define clear constraints through  rigorous set relationships, and also provide computable and optimizable objective functions.
- Well-designed Benchmark: the authors provide 8 different long-term generation tasks, and also two different settings for different real-world applications. They also provide a complete and comprehensive technical implementation details.

Weaknesses:
- I find the evaluation pipeline depends a lot on the LLM, e.g. GPT 4o, if the LLM used has some biases in claim extraction and verification accuracy, then this could result in failure to generalize to evaluation with different LLM judges.

---

> ### Author Rebuttal · Authors · 2025-07-31
>
> We thank reviewer **Dux5** for their time and feedback! We are encouraged that **Dux5** finds our paper “well-presented [with] good motivation,” that our PIC framework “can serve as a foundation for future intrinsic hallucination research” and is “precise…for a more fine-grained task,” and that our benchmark is “well-designed…[with] complete and comprehensive technical implementation details.”
>
> Responding to the reviewer’s comment below:
>
> **"[T]he evaluation pipeline depends a lot on the LLM…[which] could result in failure to generalize to evaluation with different LLM judges."**
>
> We thank the reviewer for raising this important point. We acknowledge that evaluations that rely on a specific LLM (in our case, GPT-4o mini) may be susceptible to model-specific biases that affect generalizability. We address this concern along two fronts, with new experiments:
>
> First, we explore changing the LLM backbone to GPT-4.1 and Claude 3.5 Sonnet when evaluating generations from the following baselines: Llama 3.1 8B Inst., PIC-LM 8B, and GPT-4o. We report the effect of different LLM judges on avg. PIC-Bench F1/precision. The results are as follows:
>
> |       **LLM Judge**        | **Llama 3.1 8B Inst.** |    **PIC-LM 8B**     |     **GPT-4o**     |
> |:--------------------------:|:----------------------:|:--------------------:|:------------------:|
> | GPT-4o mini (in our paper) |      69.1 / 73.6       |      89.9 / 91.2     |     90.5 / 90.2     |
> | GPT-4.1                    |      68.6 / 73.6       |      87.5 / 88.1     |     87.6 / 86.0     |
> | Claude 3.5 Sonnet          |      70.2 / 74.5       |      88.2 / 90.0     |     89.6 / 89.3     |
>
> We find that PIC scores are largely judge-agnostic, with GPT-4.1 behaving slightly stricter than the other two judges. Across judges, PIC-LM 8B consistently achieves the highest precision (partial PIC metric), while GPT-4o attains the highest F1 (full PIC metric).
>
> In practice, like most automatic LLM-based evaluation methods, we fix to one judge model for a fair comparison. Furthermore, note that employing GPT-4o mini as the LLM backbone costs roughly **7 USD** for a full PIC-Bench evaluation run; based on API costs, GPT-4.1 costs roughly 13x more, and Claude 3.5 Sonnet is 20x more.
>
> Second, our LLM judge of choice (GPT4o-mini) shows high agreement with human annotation. The LLM is invoked at two stages of our evaluation pipeline: **claim extraction** (decomposing text into verifiable claims) and **claim verification** (checking that a claim is supported by a set of verifiable claims). We conduct a pilot human evaluation experiment to assess agreement across PIC tasks and 3 baseline models (Llama 3.1 8B Instruct, PIC-LM 8B, and GPT-4o). We use 3 human annotators per component.
>
>
> ## Claim Extraction Human Evaluation
> For claim extraction evaluation, we sample 5 generations per PIC task and baseline model, leading to 120 samples in total. For each sample, we show the annotator the generated output and an extracted claim from that output.
>
> Recall that for claim extraction, the LLM is prompted to extract high-quality, verifiable claims from a source task. Here we ask the annotator to rate the extracted claim on a 3-point ordinal scale (0-2, 0 being the best) along the following dimensions: **faithfulness** (0: definitely supported, 1: partially supported, 2: not supported), **decontextualization** (0: self-contained, 1: ambiguous, 2: context-dependent), and **quality** (0: fluent, 1: minor readability issues, 2: hard to read).
>
> We report (i) LLM-human agreement (the average fraction of annotator ratings equal to the best score (0)  for the LLM-extracted claims), and (ii) exact inter-annotator agreement (the fraction of items where all raters chose the same label).
>
>
> | **Dimension / Metric**    |     **LLM–Human Agreement (%)**     |     **Exact Inter-Annotator Agreement (%)**     |
> |:-------------------------:|:-----------------------------------:|:------------------------------------------------:|
> | Faithfulness              |                97.5                 |                      94.1                        |
> | Decontextualization       |                89.6                 |                      76.5                        |
> | Quality                   |                99.7                 |                      99.2                        |
>
>
> We observe fairly strong LLM-human agreement (and substantial exact inter-annotator agreement) across all three axes, which confirms that GPT-4o mini is able to correctly extract high-quality, verifiable claims.
>
> ## Claim Verification Human Evaluation
> For claim verification evaluation, we sample 3 generations per PIC task and baseline, leading to 72 generations in total. For each sample, we provide the list of input claims, and a response claim that the LLM-based claim verifier has marked as either supported or unsupported. We ask the annotators, who are blind to model identity and the verifier’s label, to judge if the response claim is supported or unsupported by the input claims.
>
> We found high average human agreement with the LLM (**81.0%**),  and substantial inter-annotator agreement (Fleiss’ Kappa of **0.694**).
>
> Taken together, these results give confidence that our evaluation pipeline (using GPT-4o mini) is producing evaluations that are broadly consistent with human judgment. Finally, we emphasize that our PIC evaluation design is flexible and judge-agnostic: in practice, it’s easy to swap to a different LLM depending on one’s use case.

---

> > ### Comment · Reviewer_Dux5 · 2025-08-09
> >
> > Thank you so much for your response! My questions are answered and I would like to keep my score.

---

### Note · Authors · 2025-08-14

We thank all reviewers for their positive support, careful reading, and constructive feedback, which has significantly strengthened our paper! Below we summarize the key points raised during the discussion:

1. **Reliance on LLM-based claim extraction and verification**: Several reviewers (R1, R3, R4) noted that our instantiation of Precise Information Control (PIC) depends on LLM-based claim extraction and verification, which may limit generalization. In response, we (1) conducted human evaluation studies showing strong agreement with human annotators (with high inter-annotator agreement) for both steps, and (2) showed that PIC evaluation remains robust across different LLM judges and prompt variations.

2. **PIC-LM training setup and evaluation coverage**: Our two-stage post-training framework uses a cost-efficient preference-data construction method to produce PIC-LM 8B, which shows significantly stronger PIC performance. Reviewers (R2, R4) were interested in PIC-LM output quality (beyond factual inclusion), training scalability, and the rationale for using last-L-token scoring in preference data selection. We (1) qualitatively compared PIC-LM and Llama 3.1 8B Inst., and found the former to have better coherence and similar fluency, (2) showed training is effective at 1B, 3B, 8B, and 70B parameter sizes, and (3) ran an ablation showing that full-sequence scoring leads to a length bias and worse performance, so we retain last-L-token scoring as it is cheap and effective.

3. **Purpose and novelty of PIC in relation to existing paradigms**: Reviewers (R2, R3, R4) raised fundamental questions about PIC as a new task formulation. The premise of PIC is that it must ground entirely on user-supplied claims when instructed, meaning that it cannot add new verifiable claims (even if factually correct), and may override its parametric knowledge to emit factually incorrect claims if supplied. While PIC isn’t suited to every scenario, it is desirable where contextual knowledge ought to supersede the LM’s priors (e.g., in fast-moving or high-stakes domains). Though similar to RAG and attributable generation, PIC focuses on fine-grained claim-level information control over factuality and applies to a broader range of tasks. Despite having a distinct objective, PIC can complement these paradigms--we show that PIC-LM can meaningfully improve factuality in downstream RAG and chain-of-verification settings over strong 8B contextual baselines.

---

### Decision · Program_Chairs · 2025-09-17

**Decision:**

Accept (poster)

**Comment:**

The paper proposes Precise Information Control (PIC), a framework for verifiable claim-level grounding in long-form generation. It introduces PIC-Bench, a benchmark covering eight generation tasks (summarization, biography, QA, etc.), and a training recipe that produces PIC-LM, an 8B model optimized for claim-level control. PIC requires outputs to include only the provided claims (full PIC: all claims; partial PIC: relevant subset). The authors show that even frontier LMs hallucinate in >70% of outputs under PIC, while PIC-LM improves full-PIC F1 from 69.1% to 89.7% and enhances factual accuracy in downstream QA and verification pipelines.

### Strengths of the paper:

- The reviewers (pcWR, UZLz, 8i1m) praise the novelty of claim-level control, noting that PIC reframes intrinsic hallucination into a precise, verifiable task that extends beyond vague binary factuality.

- PIC-Bench is comprehensive, spanning 8 diverse tasks and two evaluation settings, with rigorous claim extraction and verification protocols. This breadth provides strong potential for PIC-Bench to become a standard benchmark.

- PIC-LM achieves clear empirical gains, improving both faithfulness and precision, and showing meaningful benefits in RAG and chain-of-verification pipelines.

- The authors conducted extensive ablations and new experiments during rebuttal, showing robustness across prompt variations, scalability to models from 1B to 70B, and consistency across different LLM judges and human evaluation.

- The paper is well-written and carefully presented, with detailed analysis of remaining errors (distortion, fabrication, conflation), making the methodology transparent and reproducible.

### Weaknesses of the paper:

- Reviewers (UZLz, Dux5) note that PIC has conceptual overlap with RAG and attribution, and that its novelty could be more sharply distinguished. The rebuttal clarified differences (claim-level granularity, strict coverage), but some still see overlap.

- The framework depends on LLM-based claim extraction and verification, which may introduce biases or errors, though human evaluation shows strong agreement.

- Some reviewers (pcWR, 8i1m) expressed concerns about rigid constraints: penalizing even correct extra facts may reduce user value in flexible settings. The authors argue this rigidity is necessary for high-stakes domains but acknowledge tradeoffs.

- Evaluation beyond factuality is limited: while PIC-LM maintains fluency, more extensive assessment of linguistic quality and general capabilities would be beneficial.

- Ethical concerns were raised about PopB-CF, a stress-test benchmark where user-provided claims contradict world knowledge. While the authors justify its inclusion as a test of context compliance, some reviewers remain cautious.

### Primary reasons for Accept (Poster)

The paper makes a principled and timely contribution by formulating intrinsic hallucination as a claim-level control problem, operationalized through PIC-Bench and PIC-LM. This approach offers a stricter, verifiable, and reproducible grounding objective with demonstrated empirical benefits. While overlaps with RAG/attribution exist, the claim-level precision and enforcement of full coverage distinguish PIC as a useful new paradigm. Despite concerns about rigidity and reliance on LLM-based evaluation, the authors provided strong rebuttal evidence (human validation, judge-agnostic results, additional ablations), convincing reviewers that the work is technically solid, novel in framing, and practically impactful. Overall, it is a valuable addition to the NeurIPS program, meriting acceptance as a poster.

### Summary of the discussion and rebuttal

The authors provided detailed and constructive responses. For R-UZLz, they clarified PIC’s differences from RAG/attribution and added a RAG ablation without claim extraction, where PIC-LM still improved factuality. For R-pcWR, they reported new non-PIC benchmark results and ran human evaluations of coherence and fluency, showing PIC-LM outputs remain coherent and concise. For R-8i1m, they addressed terminology (faithful vs. factual), explained SFT performance drops, added error analyses, and reported results with and without PopB-CF to improve fairness. For R-Dux5, they demonstrated that PIC evaluation is robust across judges (GPT-4o, GPT-4.1, Claude 3.5) and validated against human annotation with strong agreement. Overall, the rebuttal substantially strengthened the paper’s empirical and conceptual contributions, and multiple reviewers raised their scores, converging on poster acceptance.